# Stress-induced OMA1-mediated cleavage of AIFM1 suppresses cell growth by controlling mitochondrial OXPHOS activity

Mitsuhiro Nishigori [ID] [1,2], Serina Hirata[1], Hidetaka Kosako [ID] [3,4], Takeshi Ichinohe [ID] [5], Hendrik Nolte[6], Jan Riemer [ID] [7], Thomas Langer [ID] [6,8] & Takumi Koshiba [ID] [1,2][✉]

## Abstract

**Mitochondrial proteases regulate dynamic properties of organelle morphology and ensure functional plasticity at the cellular level. The metalloprotease OMA1 mediates constitutive and stress-inducible processing of its mitochondrial substrates, although only a few of its direct functional targets have been characterized. Using in vitro and in vivo multiproteomic and biochemical approaches, we here demonstrate that the membrane-anchored intermembrane space (IMS) protein AIFM1 serves as a mitochondrial stress-responsive OMA1 substrate. Under stress conditions, OMA1 cleaves AIFM1 in the IMS with slower kinetics than its conventional substrate, the dynamin-like GTPase OPA1. OMA1-mediated dislocation of cleaved AIFM1 from the mitochondrial inner membrane reduces its interaction with oxidative phosphorylation subunits, thereby decreasing respiratory activity and impairing cell growth. Furthermore, we reveal that under steady-state conditions AIFM1 broadly safeguards the mitochondrial proteome by mediating the import of proteins, particularly respiratory complex I subunits, via the TIM23 complex. Similar changes to the mitochondrial proteome occur in the lungs of virally infected mice, accompanied by stress-inducible AIFM1 processing. These findings identify OMA1 as a key integrator of mitochondrial stress and cellular energetics through AIFM1 remodeling.**

**Keywords** AIFM1; Mitochondrial Stress; OMA1; OXPHOS Activity; Proteolysis
**Subject Categories** Membranes & Trafficking; Metabolism; Organelles

## Introduction

Mitochondria are remarkably dynamic organelles in eukaryotic cells that undergo continuous cycles of homotypic fusion and fission events. Maintaining proper mitochondrial morphology is essential not only for accurate organelle inheritance but also for optimizing physiologic functions, such as signal transduction, metabolic activity, and quality control (Wai and Langer, 2016; Ng et al, 2021; Picard and Shirihai, 2022; Tábara et al, 2025). Recent studies in an in vivo model revealed that genetic ablation of individual components involved in mitochondrial dynamics impairs their organ functions and whole-body metabolism (Chen et al, 2003; Ishihara et al, 2009; Wakabayashi et al, 2009; Wai et al, 2015; Tezze et al, 2017). Abnormal mitochondrial dynamics are also associated with some human diseases or disorders, such as Charcot-Marie-Tooth disease type 2 A (Züchner et al, 2004), autosomal dominant optic atrophy (Delettre et al, 2000), neuromuscular defects (Shamseldin et al, 2012), and a lethal developmental disorder (Waterham et al, 2007). In addition to these inherited disorders, an imbalance in mitochondrial fusion and fission can suppress innate immune responses to viral infection (Castanier et al, 2010; Koshiba et al, 2011; Hanada et al, 2020; Yasukawa et al, 2020), highlighting the intimate link between mitochondrial dynamics and the host defense system.

Defects in mitochondrial fusion significantly enhance the heterogeneity and dysfunction of the mitochondrial population within the cell (Chen et al, 2005), leading to impaired stress responses and loss of cellular homeostasis (MacVicar and Langer, 2016). In mammals, three conserved guanosine triphosphatases (GTPases) of the dynamin family coordinate the fusion process: Mitofusins (Mfn1 and Mfn2), which are localized to the mitochondrial outer membrane (MOM), and optic atrophy 1 (OPA1), which is localized to the mitochondrial inner membrane (MIM). In humans, OPA1 has eight isoforms that derive from alternative splicing variants with further post-translational processing by mitochondrial proteases (Wai et al, 2015; Chan, 2020). Overlapping with the *m*-AAA protease 1 homolog (OMA1) is a

[1]Department of Chemistry, Faculty of Science, Fukuoka University, Fukuoka, Japan. [2]Division of Biology, Faculty of Science, Fukuoka University, Fukuoka, Japan. [3]Division of Cell Signaling, Institute of Advanced Medical Sciences, Tokushima University, Tokushima, Japan. [4]Laboratory of Proteomics, Institute of Photonics and Human Health Frontier, Tokushima University, Tokushima, Japan. [5]Department of Infectious Disease Control, International Research Center for Infectious Diseases, Institute of Medical Science, The University of Tokyo, Tokyo, Japan. [6]Max Planck Institute for Biology of Ageing, Cologne, Germany. [7]Redox Metabolism, Institute of Biochemistry and Cologne Excellence Cluster on Cellular Stress Responses in Aging-Associated Diseases (CECAD), University of Cologne, Cologne, Germany. [8]Cologne Excellence Cluster on Cellular Stress Responses in Aging-Associated Diseases (CECAD), University of Cologne, Cologne, Germany. [✉]E-mail: koshiba@kyudai.jp

stress-activated MIM peptidase that belongs to the M48 family of zinc metallopeptidases (Käser et al, 2003; López-Pelegrín et al, 2013) and constitutively processes OPA1 under normal conditions (Ishihara et al, 2006; Song et al, 2007; Ehses et al, 2009; Head et al, 2009; Quirós et al, 2012). General cellular stresses (e.g., depolarization, oxidization, or heat shock), however, lead to OMA1 activation and complete processing of long-form OPA1 (L-OPA1) isoforms in mitochondria (Ehses et al, 2009; Head et al, 2009; Quirós et al, 2012; Anand et al, 2014; Baker et al, 2014; Murata et al, 2020). Various studies of *OMA1* gene ablation in mice revealed the physiologic role of OMA1-regulated pathways. Knockout of *OMA1* generally produces relatively mild background-dependent phenotypes characterized by increased body weight and defective thermogenesis (Quirós et al, 2012). In certain genetic backgrounds, however, such as mouse models of mitochondrial cardiomyopathy caused by oxidative phosphorylation (OXPHOS) deficiency (Ahola et al, 2022) or by simultaneous loss of Parkin (Yamada et al, 2025), OMA1 deficiency results in more severe mitochondrial dysfunction and disrupted dynamics. In contrast, however, OMA1 depletion is protective in mouse models of neurodegeneration (Korwitz et al, 2016), ischemic kidney injury (Xiao et al, 2014), or heart failure (Wai et al, 2015; Acin-Perez et al, 2018), supporting the pro-apoptotic function of OMA1 by preventing OPA1 cleavage under stress conditions. Thus, OMA1 regulates cell survival in a cell- and tissue-specific manner, although the precise mechanisms remain unknown. Recent studies have further implicated OMA1 in another mitochondrial signaling pathway that activates the integrated stress response (ISR) via DAP3-binding cell death enhancer 1 (DELE1), a protein kinase activator (Fessler et al, 2020, 2022; Guo et al, 2020). In this pathway, stress-activated OMA1 cleaves DELE1 during import, generating a short form (S-DELE1) that is released from the mitochondria into the cytosol, where it ultimately triggers the ISR. Therefore, OMA1 also regulates the dynamic properties of mitochondrial morphology and cellular stress signaling to ensure mitochondrial integrity and cellular homeostasis. Despite its role as a central pivotal protease involved in a wide range of mitochondrial homeostatic processes, only a limited number of OMA1 substrates have been functionally characterized.

In the present study, using a proteome-wide approach combined with in vitro and in vivo biochemical analysis, we revealed that the intermembrane space (IMS) protein apoptosis-inducing factor mitochondria-associated 1 (AIFM1) is an OMA1 substrate under mitochondrial stress. AIFM1 is a mitochondrial flavoprotein initially identified as a trigger of caspase-independent apoptosis under certain conditions (Susin et al, 1999). Besides its role in cell death, AIFM1 contributes to cellular energy metabolism and redox homeostasis (Vahsen et al, 2004; Pospisilik et al, 2007; Hangen et al, 2015; Meyer et al, 2015; Salscheider et al, 2022; Rothemann et al, 2025). Here, we demonstrate at the molecular level, that OMA1-mediated processing of AIFM1 induces its release from the MIM and reduces its association with OXPHOS machinery subunits, ultimately impairing respiration and cell growth. Interestingly, stress-induced proteolysis of AIFM1 in mitochondria is also observed in the lungs of influenza virus-infected mice and in the hearts of mice with targeted deletion of Cox10 (Ahola et al, 2022), an assembly factor for cytochrome c oxidase. Together, these findings reveal a previously unrecognized physiologic role for OMA1 as a key integrator of mitochondrial stress and cellular energetics by altering the topology of one of its substrates.

# Results

## OMA1 mitochondrial interactome screening by mass spectrometry

Because OMA1 is involved in a wide range of biologic processes, from quality control to cellular stress responses (Rivera-Mejías et al, 2023), we reasoned that unidentified molecules in the mitochondria might functionally and physically associate with the peptidase to control mitochondrial integrity. To test this hypothesis, we sought to identify proteins that interact with OMA1 as candidate substrates. We first established cell lines stably expressing a carboxyl-terminal Myc-tagged murine OMA1 (termed OMA1/Myc) in *OMA1*-null mouse embryonic fibroblasts (*OMA1*$^{-/-}$ MEFs), validated the structural and functional properties of the recombinant protein (Appendix Fig. S1A–I), and screened for molecules that interact with OMA1/Myc in cells.

To identify OMA1-associated proteins, we isolated mitochondria from the OMA1/Myc-expressing cells that had undergone moderate crosslinking with formaldehyde, and immunoprecipitated (IP) the OMA1/Myc with an anti-Myc antibody. Liquid chromatography-tandem mass spectrometry (LC-MS/MS) analysis of the precipitated proteins identified 38 proteins that were enriched in the OMA1/Myc-expressing cell immunoprecipitate (Fig. 1A; Dataset EV1), including the known OMA1 substrate and interactor OPA1, the prohibitin membrane scaffold complexes PHB1 and PHB2, and components of the mitochondrial contact site and cristae organizing system (MICOS; MIC60, MIC19, MIC25, and MIC27). Other IMS proteins, HTRA2/Omi and mitochondrial adenylate kinase 2 (AK2), both associated with mitochondrial stress signaling (Chan, 2005; Lee et al, 2007), were also enriched in the OMA1 precipitates. In addition to these interactors, AIFM1 was highly enriched (>16-fold change) (Fig. 1A, red), and this was further validated using the parallel reaction monitoring (PRM) method (Fig. 1B). These results suggest that several molecules, including AIFM1, are members of the OMA1 interactome in mitochondria and are thus candidate substrates of the peptidase.

## Mitochondrial stress triggers OMA1-dependent processing of AIFM1

To elucidate unidentified substrates of OMA1 during mitochondrial stress, we next used a proteome-wide approach to identify neo-amino-terminal peptides that are formed in an OMA1-dependent manner. MEFs expressing OMA1/Myc or *OMA1* knockout (KO) cells treated with or without carbonyl cyanide-4-(trifluoromethoxy)phenylhydrazone (FCCP) were lysed, and their total protein extracts were digested with the thermostable protease LysargiNase (Tryp-N; Zhou et al, 2019), followed by proteome analysis (Fig. 1C). We observed peptide enrichment that was specific for a stress response (±FCCP) coupled with an OMA1 dependency (KO vs *OMA1/Myc* rescue, Dataset EV2) and identified 23 unique peptides derived from total mitochondrial proteins (Appendix Table S1). Overlaying this peptide enrichment profile with the previous immunoprecipitation-mass spectrometry (IP-MS) result (Fig. 1A), we identified three mitochondrial proteins (AIFM1, AK2, and TOM40) as candidate OMA1 substrates under mitochondrial depolarization (Fig. 1D).

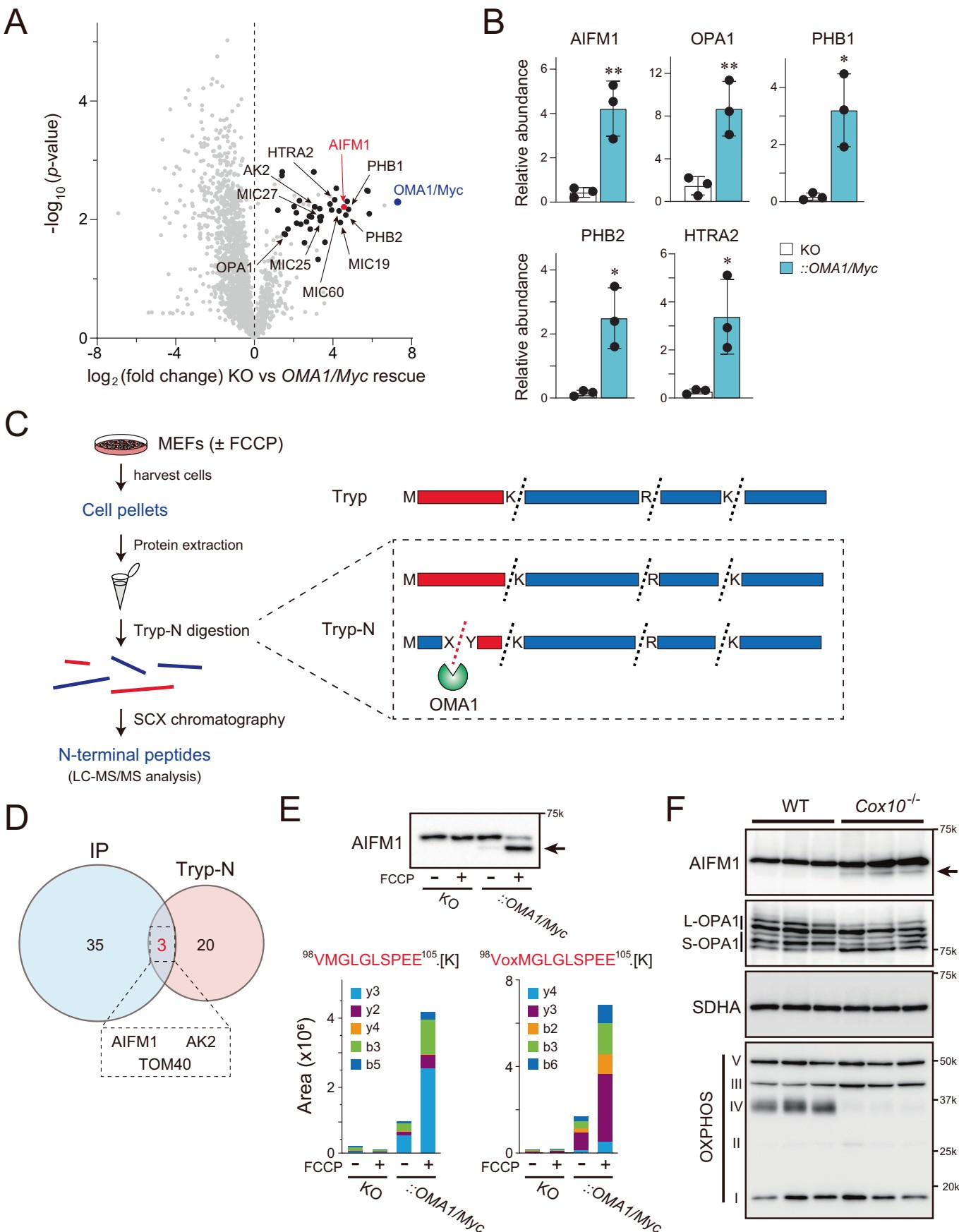

**Figure 1. AIFM1 is a stress-inducible OMA1 substrate.**

(A) Volcano plot showing OMA1/Myc interacting partners. Mitochondrial extracts from *OMA1* KO cells with or without OMA1/Myc expression were immunoprecipitated. Co-purifying proteins were identified by quantitative mass spectrometry (MS) ($n = 3$ biologic replicates). Enriched mitochondrial proteins [abundance ratio >2, $p < 0.05$ (by Student's $t$ test)] are plotted in black with AIFM1 (red) and OMA1/Myc (blue). Several known IMS or IMM proteins are also labeled. See also Dataset EV1. (B) Relative abundance of proteins (AIFM1, OPA1, PHB1, PHB2, and HTRA2) used for proteomics in (A) was measured by targeted LC-MS/MS using the PRM method. Data shown are mean values ± SD ($n = 3$ biologic replicates). *$p < 0.05$; **$p < 0.01$ (by Student's $t$ test). (C) Schematic of the N-terminal proteomic approach. Protein extracts from MEFs treated with or without FCCP were digested with Tryp-N peptidase, followed by separation of the reactants by strong cation exchange (SCX)-based chromatography, and analysis of the purified peptide fragments by LC-MS/MS. In this system, each Tryp-N-digested peptide fragment would have at least one lysine or arginine residue at the N-terminus (right dotted box), but that of the OMA1 substrates does not follow this rule (recovered in the flow-through in SCX chromatography). (D) Venn diagram showing the overlap (3 candidates; AIFM1, AK2, and TOM40) between the indicated mitochondrial molecules revealed by IP-MS (38 candidates) and the N-terminal proteomic approach (23 candidates). See also Appendix Table S1 and Dataset EV2. (E) The Tryp-N digested AIFM1 peptide (98VMGLGLSPEE105) from the *OMA1* KO or the *OMA1/Myc* rescued MEFs with (+) or without (−) FCCP treatment was quantified by targeted MS using the PRM method (the samples used in (C)). Note that the peptide was enriched only in the rescued sample and lacked K or R residues at the N-terminus, indicating that the fragment generation was OMA1-dependent. oxM, oxidized methionine. Top blot shows the immunoblot of the samples used in the study (detected by anti-AIFM1 antibody). The arrow indicates a generated AIFM1 cleavage product. (F) Loss of Cox10 activates OMA1 and leads to increased AIFM1 processing in vivo. Heart lysates from WT or Cox10$^{-/-}$ mice ($n = 3$) were analyzed by immunoblotting (indicated antibodies). The arrow indicates the AIFM1 processed band. Source data are available online for this figure.

Among these candidates, substantial AIFM1 processing in response to FCCP treatment in *OMA1/Myc*-rescued cells was confirmed by immunoblotting (Fig. 1E, arrow in top blot), consistent with data quantified by targeted MS using the PRM method (Fig. 1E, bottom graphs). Unlike in OMA1/Myc, expression of a proteolytically inactive OMA1$^{E324Q}$ mutant (Baker et al, 2014) in *OMA1* KO cells did not induce AIFM1 processing under depolarized conditions (Fig. EV1A). Hyperpolarized (oligomycin A) or oxidized ($H_2O_2$) conditions also triggered AIFM1 proteolysis in an OMA1-dependent manner (Fig. EV1B). These results demonstrate that OMA1 mediates AIFM1 processing under a wide range of stress conditions. To corroborate these in vitro findings, we analyzed AIFM1 processing in *Cox10*$^{-/-}$ mouse heart tissue. Loss of Cox10 was associated with cardiomyopathy (Diaz et al, 2005), leading to OMA1 activation and increased OPA1 processing (Ahola et al, 2022; Fig. 1F); notably, we observed at least some AIFM1 processing in cardiac tissue (Fig. 1F, arrow), but the deletion of *OMA1* in *Cox10*$^{-/-}$ mice stabilized AIFM1 as well as L-OPA1 (Fig. EV1C).

Several mitochondrial proteases process their substrates under constitutive and/or stress-induced conditions. The IMS-localized *i*-AAA protease YME1L and OMA1 share substrates, such as L-OPA1, although these proteases recognize distinct sites in the protein (Deshwal et al, 2020). In terms of redundancy, we investigated whether YME1L might also proteolyze AIFM1 under certain circumstances. By monitoring AIFM1 proteolysis in both *YME1L*$^{-/-}$ and *OMA1*$^{-/-}$*YME1L*$^{-/-}$ (DKO) cells under constitutive (DMSO) or depolarized (FCCP or valinomycin) conditions, we confirmed that the AIFM1 cleaved band was only observed in cells with OMA1 (Fig. 2A, wild-type [WT] and *YME1L*$^{-/-}$). Consistent with this finding, reintroduction of *OMA1*, but not *YME1L*, into the DKO cells was sufficient to trigger AIFM1 proteolysis (Fig. EV1D), suggesting that these two proteases are functionally distinct with respect to AIFM1 cleavage. We further investigated the actions of other mitochondrial proteases that may be involved in AIFM1 proteolysis. Knockdown of five independent IMS or MIM proteases in HeLa cells by small interfering RNA (siRNA) strongly attenuated the level of each protein (Fig. 2B). Although FCCP-induced AIFM1 cleavage was clearly inhibited in *siOMA1*-treated cells, as observed in *OMA1*$^{-/-}$ cells (Fig. 2A), knockdown of the other four proteases (YME1L, AFG3L2, HTRA2, and PARL) had no effect (Fig. 2B, top). Thus, we concluded that AIFM1 is a OMA1 substrate under mitochondrial stress conditions.

## Identification of the OMA1 cleavage site in AIFM1

As the functional domains of both OMA1 and AIFM1 are directed toward the IMS and anchored to the MIM, we next investigated how OMA1 mechanistically accesses its substrate during mitochondrial proteolysis. To address this issue, we first examined the dynamic properties of the mitochondrial membrane using fusion- or fission-incompetent cells. Interestingly, mitochondrial depolarization induced by either FCCP or valinomycin in *OPA1*- or *Mitofusin* (*Mfn1/Mfn2*)-deficient (*Mfns*-DKO) cells was sufficient to trigger the OMA1-dependent AIFM1 processing (Fig. 2C). In addition, *Drp-1*$^{-/-}$ MEFs, which lack the mitochondrial fission event (Ishihara et al, 2009; Wakabayashi et al, 2009), also exhibited normal AIFM1 processing through OMA1 function (Fig. EV1E). These results suggest that mitochondrial fusion and fission processes are likely dispensable for OMA1-mediated AIFM1 cleavage (Fig. EV1F).

Our peptide mapping based on the N-terminal proteome identified that the valine at position 98 was the N-terminal amino acid of the generated AIFM1 fragment cleaved by OMA1 (Fig. 1E; Appendix Table S1). The cleavage site was located just downstream of the transmembrane domain in AIFM1 (Fig. 2D, top structure, and Appendix Fig. S2A). Consistent with this observation, cleaved AIFM1 was released from the MIM into the IMS, where it existed as a soluble, membrane-unbound protein like HTRA2/Omi or SMAC/DIABLO (Fig. 2E). Sequence alignments between AIFM1 homologs in different species revealed conservation of the OMA1 recognition sequences, particularly the Glu96-Arg97 dipeptide just before the cleavage site (Fig. 2D, bold letters), whereas we observed no conservation of any OMA1 cleavage motif (Appendix Fig. S2B) (Ishihara et al, 2006; Guo et al, 2020; Ahola et al, 2024). We generated an AIFM1 deletion mutant lacking a tripeptide motif that includes Val98 (AIFM1$^{Δ97-99}$, Appendix Fig. S2C) and tested whether OMA1 could properly cleave the mutant protein. As expected, OMA1 failed to cleave the deletion mutant in cells (Fig. 2F; Appendix Fig. S2D). We also introduced substitutions into AIFM1 at the site corresponding to the OMA1 recognition sequence and found that the R97E variant could not be processed by OMA1 (Fig. 2F; Appendix Fig. S2D). These results demonstrate that the tripeptide motif in AIFM1 is essential for its recognition and cleavage by OMA1, and that the net charge surrounding the upstream cleavage site in AIFM1 likely facilitates substrate recognition.

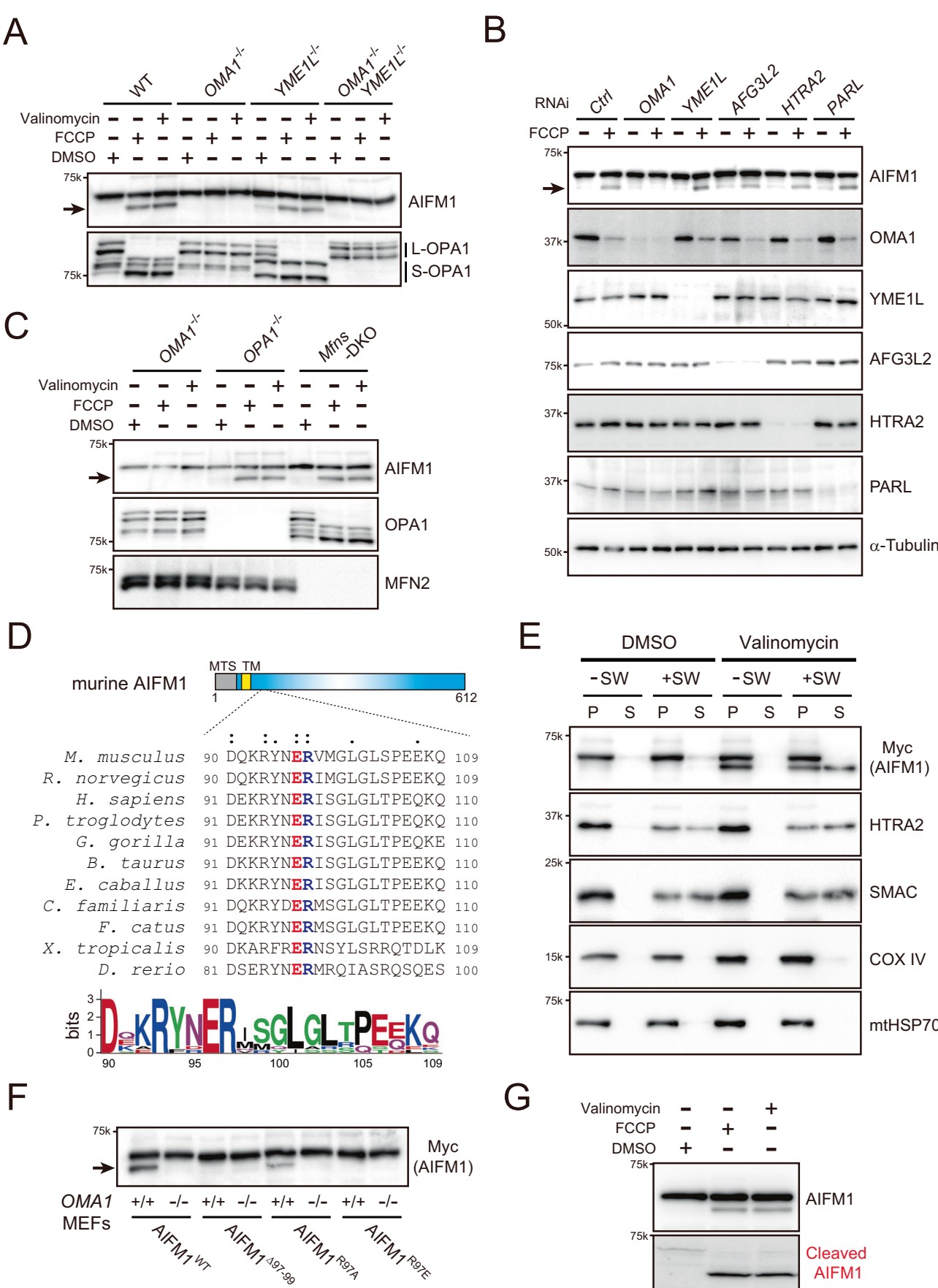

◀ **Figure 2. OMA1-dependent processing of AIFM1 in cells.**

(A) OMA1 is required for inducible AIFM1 processing. WT, $OMA1^{-/-}$, $YME1L^{-/-}$, or $OMA1^{-/-}YME1L^{-/-}$ MEFs were incubated for 3 h in the absence (DMSO) or presence of either FCCP (40 μM) or valinomycin (1 μg/mL) and analyzed by immunoblotting (indicated antibodies). The arrow indicates the AIFM1 processed band. (B) HeLa cells were transfected with the indicated siRNA and treated with or without FCCP (40 μM) for 3 h. Immunoblotting was performed as indicated and the arrow indicates the AIFM1 processed band. *Ctrl*, control siRNA. (C) Similar to (A), except that $OPA1^{-/-}$ and $Mfn1^{-/-}Mfn2^{-/-}$ (*Mfns*-DKO) MEFs were used. *Mfns*-DKO and *OPA1*-KO are incompetent for OM and IM fusion, respectively. (D) Alignment of AIFM1 homologs (*Mus musculus, Rattus norvegicus, Homo sapiens, Pan troglodytes, Genus gorilla, Bos taurus, Equus caballus, Canis familiaris, Felis catus, Xenopus tropicalis,* and *Danio rerio*) downstream of the transmembrane (TM) region. Colons (:) indicate conserved residues among all species shown in the figure, and dots (·) indicate residues with similar properties. Top, structure of murine AIFM1 showing the location of the mitochondrial target signal (MTS) and the TM segment, with amino acid positions indicated below the structure. Bottom, sequence logo of positions adjacent to the cleavage site in AIFM1 by OMA1. (E) Mitochondria isolated from WT MEFs treated with or without valinomycin (1 μg/mL) were resuspended in either isotonic (−SW) or hypotonic swelling (+SW) buffer and kept on ice for 15 min. After centrifugation, the supernatant (S) and pellet (P) were analyzed by western blotting using the indicated sub-mitochondrial markers. HTRA2 and SMAC are used as a positive control for unanchored IMS protein. COX IV and mtHSP70 are MIM and matrix proteins, respectively. (F) MEFs ($OMA1^{+/+}$ or $OMA1^{-/-}$) stably expressing a Myc-labeled version of WT or mutant (Δ97-99, R97A, or R97E) AIFM1 were treated with 40 μM FCCP for 3 h and analyzed by immunoblotting with anti-Myc antibody. The arrow indicates the AIFM1 cleaved band. (G) The cleaved AIFM1 antibody (bottom) specifically recognizes proteins with the identified cleaved end of the N-terminus. The WT MEF samples used in (A) were tested. Source data are available online for this figure.

Finally, we generated a specific antibody against the cleaved form of AIFM1 to validate our biochemical findings of the cleavage site at the endogenous level (Fig. EV2A,B). Using this antibody, we detected a cleaved AIFM1 band with an apparent molecular mass of ~56 kDa, but only under conditions in which the mitochondrial membrane potential ($\Delta\Psi_m$) was dissipated (Fig. 2G, bottom). Notably, the antibody-based probe was also highly sensitive to a range of mitochondrial stresses known to activate OMA1 (Fig. EV2C,D). In addition, our antibody revealed that μ-calpain-dependent processing of AIFM1 (Polster et al, 2005) is OMA1-independent and that the cleavage site is also distinct (Fig. EV2E).

## Differences in the rates of OMA1-mediated cleavage of AIFM1 and OPA1

When WT MEFs were treated with FCCP, activated OMA1 rapidly attacked L-OPA1 and induced substrate cleavage within minutes, leading to the accumulation of S-OPA1 isoforms; the OPA1 pattern in $OMA1^{-/-}$ cells remained unchanged (Fig. 3A, middle blots) (Baker et al, 2014). In contrast, OMA1-mediated processing of AIFM1 occurred significantly more slowly than that of OPA1, becoming apparent only approximately 15 min after membrane depolarization (Fig. 3A,B). It is conceivable that OMA1 only cleaves AIFM1, whose mitochondrial import is inhibited under depolarized conditions (Chacinska et al, 2009). Inhibition of cytosolic protein synthesis with cycloheximide (CHX), however, did not affect the kinetics of OMA1-mediated processing of either AIFM1 or OPA1 (Fig. 3C), excluding the possibility that the observed kinetic difference between these substrates was due to substrate import failure upon mitochondrial depolarization.

We next asked whether other OMA1 substrates might influence the relatively slow kinetics of AIFM1 processing by OMA1. We removed OPA1, the rapidly processed OMA1 substrate, from mitochondrial membrane and examined whether the kinetics of AIFM1 processing were altered. Strikingly, AIFM1 cleavage upon mitochondrial depolarization was accelerated in $OPA1$-deficient cells (Fig. 3D). This effect was suppressed in cells expressing either L-OPA1 (variant 1 isoform, Fig. 3D,E) or a non-cleavable OPA1 variant (OPA1V1$^{\Delta4}$, Fig. EV3A) (Ahola et al, 2024), demonstrating that S-OPA1 is dispensable for accelerated AIFM1 processing in depolarized mitochondria. In addition, AIFM1 showed an increased susceptibility to OMA1 in $OPA1^{-/-}$ cells, and the protein was degraded at even low levels of mitochondrial stress

(FCCP < 1 μM, Fig. EV3B). These observations in the $OPA1^{-/-}$ cells were not simply due to defects in the cristae morphology (Pernas and Scorrano, 2016) or loss of respiratory activity (Chen et al, 2005), as eliminating MIC60, a core component of MICOS (van der Laan et al, 2016), or knocking out both Mfn1 and Mfn2 still resulted in distinct and relatively slower AIFM1 processing kinetics compared with the $OPA1$ KO phenotype (Fig. EV3C,D). The removal of another OMA1 substrate, DELE1, also had no effect on the kinetics of OMA1-mediated AIFM1 cleavage (Fig. EV3E), suggesting that mitochondrial stress-induced substrate processing events by the activated OMA1 are independent (Fig. 3F).

Lastly, we explored the role of MIA40/CHCHD4, an AIFM1 interactor (Hangen et al, 2015; Meyer et al, 2015; Petrungaro et al, 2015) that may modulate OMA1-mediated cleavage of AIFM1. We established cell lines stably expressing an AIFM1 variant that cannot bind to MIA40 (AIFM1$^{DKP}$; Fagnani et al, 2024) in $AIFM1$ KO cell lines (the Flp-In system, Salscheider et al, 2022). We found that the AIFM1$^{DKP}$ variant markedly destabilized the protein, making it more susceptible to OMA1 attack under mitochondrial stress conditions (Appendix Fig. S3). Consequently, the protein degrades more quickly; however, if binding of MIA40 to AIFM1 induces structural changes within AIFM1 (Brosey et al, 2025; Mussulini et al, 2025), this interaction could strengthen the substrate's resistance to proteolysis.

## Membrane dislocation of AIFM1 affects its complex formation in mitochondria

To further understand the functional relevance of AIFM1 processing, we next generated cell lines that accumulate cleaved AIFM1 without inducing mitochondrial stress. We swapped the arginine 97 position in a carboxyl-terminal Myc-tagged AIFM1 with a tobacco etch virus (TEV) protease cleavage site (TCS; termed AIFM1$^{TCS}$; Fig. 4A) and expressed the variant at physiologic levels in Flp-In 293 $AIFM1$ KO cell lines. Insertion of the TCS into AIFM1 did not affect its subcellular localization (Fig. EV4A), and co-expression of TEV protease in the IMS (IMS-TEV) but not in the mitochondrial matrix (Su9-TEV) allowed specific processing of AIFM1$^{TCS}$ (Fig. 4B, arrow). Using this system, we produced a mitochondrial protein (designated AIFM1$^{TCS/TEV}$; Fig. 4C,D) with the expected molecular weight (~60k Da; including Myc tag) at high efficiency (Fig. EV4B, right lane). We also confirmed that expression of the TEV protease in the IMS had no significant off-

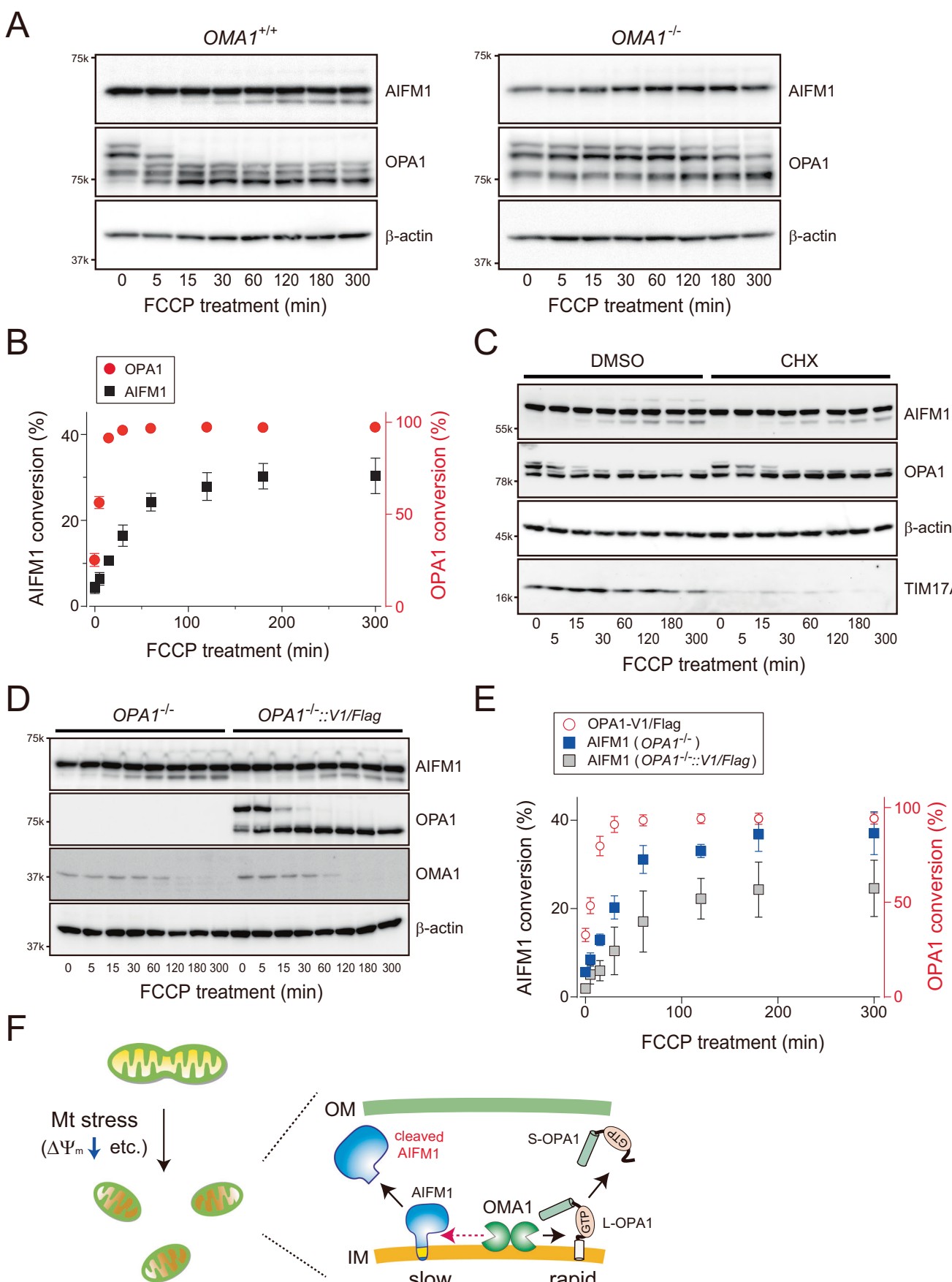

◄ **Figure 3. AIFM1 processing by OMA1 is a relatively slow reaction.**

(A) The kinetic profile of stress-induced substrate processing (OPA1 and AIFM1) is dependent on OMA1. The $OMA1^{+/+}$ (left) or $OMA1^{-/-}$ (right) MEFs incubated with FCCP (40 μM) were collected at the indicated time points (0, 5, 15, 30, 60, 120, 180, and 300 min) and analyzed by western blotting. β-Actin blots were used as loading controls for each time point. (B) OMA1-dependent substrate processing in $OMA1^{+/+}$ MEFs (A) was quantified ($n = 3$ biologic replicates) and conversions (%) were plotted (mean values ± SD). Red circle, OPA1; Black square, AIFM1. (C) Similar to (A), except that $OMA1^{+/+}$ MEFs pretreated with either DMSO or cycloheximide (CHX, 10 μg/mL) were used for the FCCP-inducible substrate processing assay. The TIM17A blot was used as a marker to verify the effect of CHX in cells. (D) Similar to (A), except that the $OPA1^{-/-}$ and $OPA1^{-/-}$ MEFs stably expressing FLAG-tagged OPA1 (variant 1; V1/FLAG) were treated with FCCP for the indicated times. (E) The OMA1-dependent AIFM1 processing in the $OPA1^{-/-}$ (blue square) or $OPA1^{-/-}$ MEFs stably expressing V1/FLAG (gray square) shown in (D) was quantified ($n = 3$ biologic replicates) and each conversion (%) was plotted (mean values ± SD). The kinetic profile of V1/FLAG (red open circle) shows the OMA1-dependent processing of the protein in the rescue cells. (F) Model of the AIFM1 processing by OMA1 in the IMS. Mitochondrial stress-induced activation of OMA1 triggers rapid processing of its substrate OPA1 followed by subsequent targeting of AIFM1 (slow). Source data are available online for this figure.

target effects on other mitochondrial proteins, such as OMA1 and OPA1, and did not activate the transcription factor ATF4, which coordinates the mitochondrial stress response in mammals (Fig. EV4C) (Quirós et al, 2017; Guo et al, 2020; Girardin et al, 2021). Having clarified the solubility of the naturally cleaved form of AIFM1 in depolarized mitochondria (Fig. 2E), we found that the AIFM1$^{TCS/TEV}$ variant was similarly released from the MIM, as assessed by a mitochondrial swelling assay (Fig. 4E). Intriguingly, the presence of membrane-dislocated AIFM1$^{TCS/TEV}$ variant did not appear to increase sensitivity to staurosporine- or Bax-induced cell death relative to the membrane-anchored form (Appendix Fig. S4A,B), suggesting that AIFM1 processing by OMA1 and apoptotic insults are not correlated (Fig. EV2E).

We found that membrane dislocation of AIFM1 affects its assembly in mitochondria. The soluble AIFM1$^{TCS/TEV}$ variant formed 150-kDa complexes, estimated to be a dimer formation (Maté et al, 2002; Ferreira et al, 2014; Brosey et al, 2025; Mussulini et al, 2025) based on clear native-polyacrylamide gel electrophoresis (CN-PAGE) (Fig. 4F, right two lanes). In contrast, both the membrane-anchored AIFM1$^{WT}$ and the non-cleavable AIFM1$^{Δ97-99}$ variant were part of larger complexes with a molecular weight >240 kDa (Hevler et al, 2021; Salscheider et al, 2022), likely due to the formation of multiple homotypic and/or heterotypic complexes in mitochondria (Fig. 4F). Notably, when the AIFM1$^{WT}$ cells were treated with FCCP (lane, AIFM1$^{WT}$ + FCCP), we observed that two major populations co-migrated, one with the original larger heterogeneous peaks and the other with a lower mass peak consistent with the predominant band seen in the AIFM1$^{TCS/TEV}$ fraction. Strikingly, our specific antibody to cleaved AIFM1 revealed that the smaller band seen in the FCCP-treated AIFM1$^{WT}$ fraction contained the cleaved AIFM1 (Fig. 4F, middle blots). Importantly, this antibody did not cross-react with the AIFM1$^{TCS/TEV}$ variant, which contains an additional glycine residue at the N-terminus due to TEV processing (Fig. 4A). Assembly of AIFM1 into a high molecular mass complex was further validated by size-exclusion chromatography at endogenous expression levels in different cell lines (Fig. 4G). Together, these results demonstrate that AIFM1 ordinarily forms larger complexes in the IMS (Hevler et al, 2021; Wang et al, 2021; Salscheider et al, 2022) that are modulated upon OMA1-mediated cleavage and the release of AIFM1 from the MIM.

## AIFM1 ensures cell proliferation by controlling OXPHOS activity

To define the AIFM1 assembly, we performed IP-MS to identify its interactome in mitochondria. Mitochondrial extracts from AIFM1$^{TCS}$ and AIFM1$^{TCS/TEV}$-expressing cells were subjected to IP,

and the digested peptides were analyzed by MS (Fig. 5A; Dataset EV3). The proteomic data revealed that most of the enriched proteins in the AIFM1$^{TCS}$ fraction were respiratory chain complex subunits, which are part of the OXPHOS machinery (Fig. 5A,B). Comparative analysis between these two variants revealed that the components of mitochondrial complex I were predominantly affected in the AIFM1$^{TCS}$ fraction (Fig. 5C,D, highlighted in red [D]), supporting the finding that AIFM1 is required for complex I biogenesis (Vahsen et al, 2004; Urbano et al, 2005; Delavallée et al, 2020; Wang et al, 2021; Salscheider et al, 2022). We verified this result by western blot analysis and confirmed that the AIFM1$^{TCS}$, but not soluble AIFM1$^{TCS/TEV}$, was efficiently co-precipitated with some complex I, III, and IV components (Fig. 5E). These results clearly indicate that the assembly of AIFM1 with such interactors (a member of the complexome) is dependent on the MIM-anchored conformation.

How does mislocalization of AIFM1 in mitochondria affect physiologic functions? The observed association of AIFM1 with OXPHOS components led us to investigate whether mislocalization of AIFM1 in mitochondria could affect its functional relevance in energy production via OXPHOS activity. To this end, we assessed complex I activity (i.e., NADH dehydrogenase activity) in cells using CN-PAGE (Wittig et al, 2007). Using this approach, we confirmed that complex I activity was significantly reduced in AIFM1 KO cells compared to WT cells (Fig. 5F), as previously reported (Delavallée et al, 2020; Salscheider et al, 2022). The attenuation of complex I activity observed in the AIFM1 KO cells was substantially restored when the AIFM1$^{TCS}$ variant was expressed, but the restoration was not sufficient in AIFM1$^{TCS/TEV}$ cells (Fig. 5F). Consistent with these results, quantification of ATP levels revealed a much higher ATP concentration in cells with AIFM1$^{TCS}$ than in those with the AIFM1$^{TCS/TEV}$ variant (Fig. 5G), and measurement of oxygen consumption rates supported respiratory recovery in AIFM1$^{TCS}$ cells (Figs. 5H and EV4D). These results indicate that AIFM1 anchoring to the MIM is correlated with control of the OXPHOS system. Most importantly, cell growth in galactose-containing medium, where cellular biogenesis would rely on mitochondrial respiratory activity (Mishra et al, 2014), was impaired in the AIFM1$^{TCS/TEV}$ cells compared to AIFM1$^{TCS}$ cells (Fig. 5I). Taken together, these data demonstrate that the membrane-associated AIFM1 interactome ensures cell proliferation by controlling OXPHOS activity. Based on these results, we propose that OMA1 cleavage-mediated AIFM1 inactivation would lead to an OXPHOS deficiency.

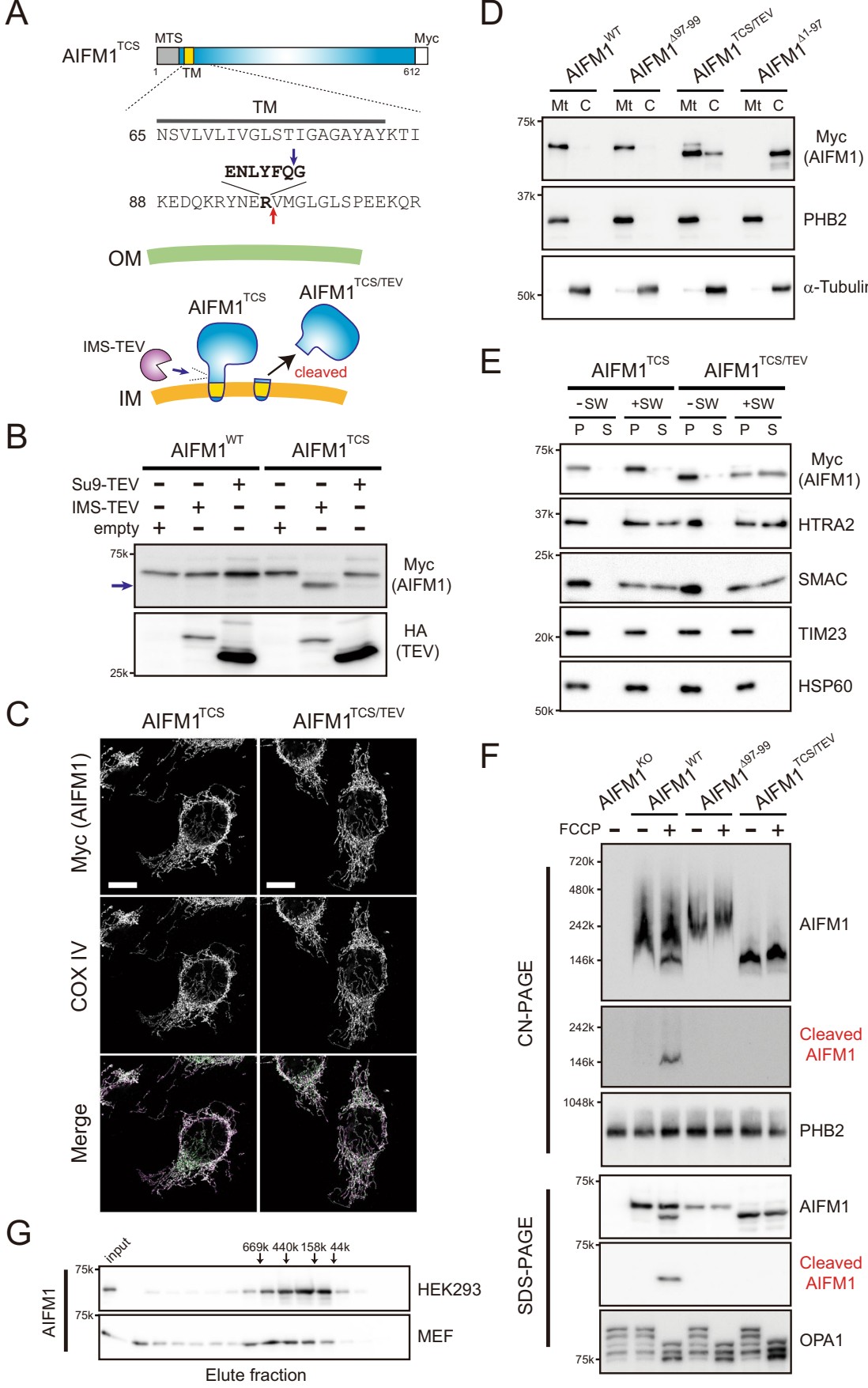

**Figure 4. Submitochondrial localization of AIFM1 is necessary to preserve its assembly states.**

(A) The sequence of AIFM1$^{TCS}$ harboring a TEV cleavage site (TCS; ENLYFQG) where the R97 residue in AIFM1 has been replaced by TCS (bold). The red arrow in the sequence indicates the original cleavage site by OMA1 and the blue arrow represents the newly generated processing site by the TEV protease. The bottom image illustrates the possible topology of AIFM1$^{TCS}$ in mitochondria and the TEV protease located in the IMS (IMS-TEV) hypothetically cleaving and releasing the variant from the membrane (designated AIFM1$^{TCS/TEV}$). (B) The Flp-In-293-AIFM1$^{WT}$/Myc or -AIFM1$^{TCS}$/Myc cells were transfected with expression plasmids of HA-tagged Su9-TEV (matrix-targeted) or IMS-TEV, and their whole cell lysates were analyzed by western blotting using the indicated antibodies. The arrow indicates the generated AIFM1$^{TCS/TEV}$. (C) Subcellular localization of AIFM1 variants. Flp-In-293-AIFM1$^{TCS}$/Myc cells without or with stable expression of IMS-TEV using a retroviral system (AIFM1$^{TCS/TEV}$) and the generated AIFM1 variants in the cells were monitored by an immunofluorescence against the Myc epitope to determine their subcellular localizations (top panel). Mitochondria in the same cells were also identified by staining with an anti-COX IV antibody (middle). We confirmed that both AIFM1 (magenta) and COX IV (green) were completely merged in mitochondria (bottom). Scale bar, 10 μm. (D) Cellular fractions from cells expressing the AIFM1 variants (WT, Δ97-99, TCS/TEV, and Δ1–97) were collected by differential centrifugation and analyzed by western blotting using the indicated subcellular markers (PHB2, mitochondria; α-Tubulin, cytosol). Mt, mitochondrial fraction; C, cytosolic fraction. (E) Similar to Fig. 2E, except that mitochondria were isolated from cells expressing the AIFM1 variants (AIFM1$^{TCS}$ and AIFM1$^{TCS/TEV}$). HTRA2 and SMAC are used as a positive control for an unanchored IMS protein. TIM23 and HSP60 are MIM and matrix proteins, respectively. (F) Depolarized (+FCCP) or normal mitochondria (−FCCP) isolated from *AIFM1* KO or cells expressing AIFM1 variants (WT, Δ97-99, and TCS/TEV) were solubilized in CN-PAGE lysis buffer containing 0.5% (w/v) digitonin, analyzed by CN-PAGE (top 3 blots), and immunoblotted using the AIFM1-, cleaved AIFM1-, and PHB2-specific antibodies. The abundance of mitochondrial proteins in each sample was also confirmed by SDS-PAGE (bottom 3 blots) in parallel. (G) Size exclusion chromatography (Superose 6 Increase column) of endogenous AIFM1 extracted from the mitochondrial fractions of HEK293 (top) or MEF (bottom) cells at pH 8.0. The positions corresponding to the elution of standard markers of molecular mass are indicated. Each eluted fraction was analyzed by western blotting with the anti-AIFM1 antibody. Source data are available online for this figure.

## AIFM1 coordinately regulates substrate import by associating with TIM23 translocase

To substantiate these findings, we performed an MS-based analysis of the mitochondrial proteome in AIFM1-variant cells at steady state. The topologic change in AIFM1 was accompanied by extensive changes in the mitochondrial proteome (Fig. 6A; Dataset EV4). Gene Ontology (GO) annotation of the mitochondrial proteome in AIFM1$^{TCS/TEV}$ cells (boxed region in Fig. 6A) revealed that mitochondrial respiration and the protein import pathway were predominantly affected (Fig. 6B), consistent with previous interactome results (Fig. 5). Therefore, we compared the enrichment profile of mitochondrial proteins found in the AIFM1$^{TCS/TEV}$ variant cells. Although membrane dislocation of AIFM1 did not grossly alter mitochondrial DNA levels (Fig. EV4E), expression of OXPHOS-related genes (Fig. EV4F; Dataset EV5), or the ΔΨ$_m$ (Fig. EV4G), 32 subunits of the OXPHOS complex were decreased in the AIFM1$^{TCS/TEV}$ variant (Fig. EV4H). These results imply that membrane-associated AIFM1 maintains the protein levels of each OXPHOS subunit at steady state. Moreover, we examined the role of the AIFM1 topology in organizing the MIA40 pathway, which controls the import and folding of a number of IMS proteins (Reinhardt et al, 2020; Edwards et al, 2021). We confirmed that MIA40 substrate protein levels were also significantly decreased in cells with the AIFM1$^{TCS/TEV}$ variant (Fig. EV4I), suggesting that spatial organization may be another key function of AIFM1 in the mitochondrial disulfide relay (Brosey et al, 2025).

To explore the possible involvement of AIFM1 in the biogenesis of OXPHOS subunits, we examined whether AIFM1 could maintain its protein import by cooperating with import machinery beyond the MIA40-dependent pathway (Salscheider et al, 2022; Peker et al, 2023). To this end, we investigated the physical interaction between AIFM1 and mitochondrial translocase complexes by IP assay. Using the AIFM1 variant cells, we found that AIFM1$^{TCS}$ interacted with subunits consisting of the translocase of the inner membrane 23 (TIM23) complex (TIM23, TIM50, and TIM17), but not with the TIM22 translocase (Fig. EV5A). By contrast, the ability of AIFM1$^{TCS/TEV}$ to bind to the TIM23

translocase was weakened, as also confirmed by the IP-MS result (Fig. EV5B). The membrane-associated AIFM1 variant also showed substance affinity for the MIM scaffold, the PHB complex (particularly for PHB1) (Fig. EV5A). Consistent with these results, we confirmed that TIM23 co-immunoprecipitated endogenous AIFM1 in HeLa cells (Fig. 6C). We believe that this interaction is not part of a transient association during protein import because inhibiting protein synthesis with CHX did not diminish the observed AIFM1−TIM23 interaction (Fig. 6C, CHX). The interaction of both proteins was further validated by a proximity ligation assay (Fig. 6D). In line with these results, size exclusion chromatography revealed that deletion of AIFM1 shifted the TIM23 translocase to a lower molecular weight fraction (Fig. EV5C), demonstrating that AIFM1 binds to the TIM23 complex. Moreover, the physical interaction between the AIFM1 and TIM23 translocase affected the susceptibility of AIFM1 to OMA1-dependent proteolytic processing. When TIM23 was depleted from cells by siRNA, OMA1-mediated AIFM1 processing was significantly increased in depolarized mitochondria (Fig. EV5D), suggesting that OMA1 accessibility to AIFM1 was increased by removal of the TIM23 complex.

Having identified that the membrane-targeted AIFM1 is associated with the TIM23 translocase in mitochondria, we revisited the mitochondrial proteome enrichment analysis in AIFM1 variant cells and sought to identify related molecules that rely on the TIM23 import pathway. LC-MS/MS identified 114 individual proteins that were downregulated in the AIFM1$^{TCS/TEV}$ cells (Fig. 6A, boxed region), and defined the majority of these (103 proteins) as substrates of the TIM23 translocase (Crameri et al, 2024) (Dataset EV4). In particular, we found that 21 of the 103 mitochondrial proteins correspond to subunits of the OXPHOS complex that are imported via the TIM23 pathway (Fig. 6E, orange). Indeed, our in vitro import assay revealed that loss of AIFM1 or its membrane anchoring moderately impaired protein import of NDUFV2 or NDUFAB1, the 24-kDa or 16-kDa subunits of complex I, into mitochondria compared with AIFM1$^{TCS}$-containing mitochondria, but not the TIM23-independent substrate NDUFB7 (Fig. 6F,G) (Crameri et al, 2024). We conclude that AIFM1 anchoring to the MIM results in a functional interaction

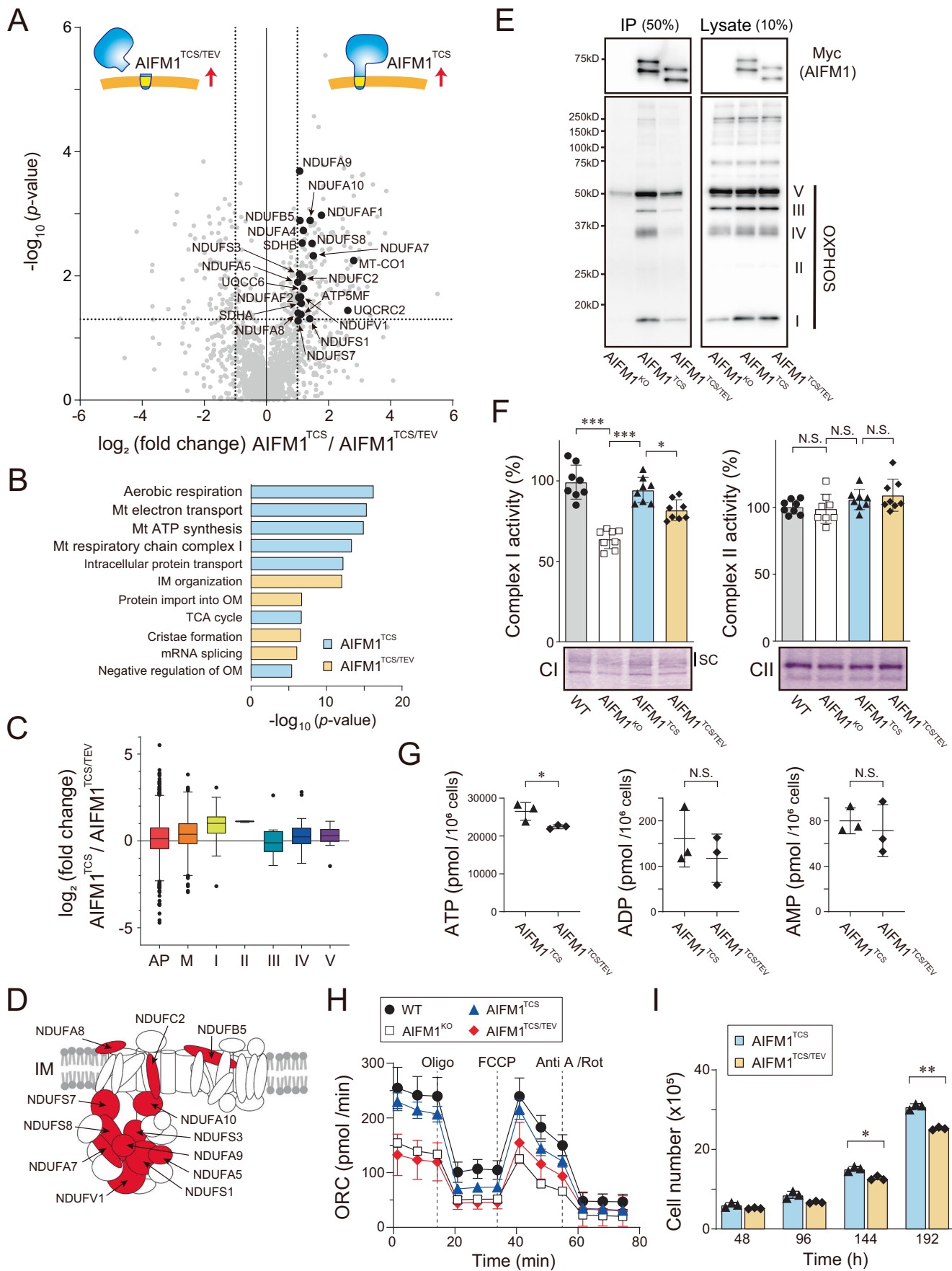

◄

**Figure 5. Functional interaction between AIFM1 and OXPHOS machinery.**

(A) Proteomic analysis to evaluate the interactome of AIFM1 dependent on membrane anchoring. Mitochondrial extracts from either formalin-fixed AIFM1$^{TCS}$ or AIFM1$^{TCS/TEV}$-expressing cells were immunoprecipitated. Co-purifying proteins were identified by quantitative MS ($n = 3$ biologic replicates) followed by statistical testing (Student's $t$ test). Several subunits of OXPHOS (black) enriched in the AIFM1$^{TCS}$ fraction are indicated. See also Dataset EV3. (B) GO enrichment analysis of IP-MS results from (A) showing the 11 most significantly altered biologic processes in AIFM1$^{TCS}$ (light blue) and AIFM1$^{TCS/TEV}$ (yellow) (by modified Fisher's exact test). (C) Box and whisker plot showing log$_2$ fold-change distributions of proteins quantified by LC-MS/MS in (A) that were identified in specific OXPHOS complexes I-V. AP = all detected proteins (1397 proteins), M = MitoCarta3.0 (418 proteins), CI = complex I (39 proteins), CII = complex II (3 proteins), CIII = complex III (13 proteins), CIV = complex IV (34 proteins), and CV = complex V (12 proteins). The box borders indicate the 25% and 75% quantiles, and outliers are indicated by a black plot (greater distance than 1.5 times the interquartile range). (D) Illustration of the mitochondrial respiratory complex I, highlighting the positions of the 12 subunit positions with a high affinity for AIFM1, which are shown in red. (E) Interaction of OXPHOS components with AIFM1 variant. The *AIFM1* KO or AIFM1 variants (AIFM1$^{TCS}$ and AIFM1$^{TCS/TEV}$) expressing cells were moderately fixed with 0.2% formaldehyde and their lysates were immunoprecipitated with anti-Myc antibody followed by western blotting analysis with an OXPHOS monoclonal antibody cocktail. The IP and lysate were loaded at 50% and 10% of the input samples, respectively. Complex I, NDUFB8; Complex II, SDHB; Complex III, UQCRC2; Complex IV, MTCO1; Complex V, ATP5A. (F) In-gel catalytic activity assay of digitonin-solubilized mitochondrial complex I and II in CN-PAGE (bottom image). In this assay, mitochondria were isolated from Flp-In-293 cells (WT, AIFM1 KO, AIFM1$^{TCS}$, and AIFM1$^{TCS/TEV}$). SC, supercomplexes. Graph, Each value represents the mean ± SD ($n = 8$ biologic replicates) and statistical analysis was performed by one-way ANOVA followed by Tukey's test. ***$p < 0.001$ and *$p < 0.05$. N.S., not significant. (G) ATP, ADP, and AMP levels (pmol/10$^6$ cells) in AIFM1$^{TCS}$ and AIFM1$^{TCS/TEV}$ cells were analyzed by LC-MS/MS. Data shown are mean ± SD ($n = 3$ biologic replicates). *$p < 0.05$. N.S., not significant (by Student's $t$ test). (H) The oxygen consumption rate (OCR) of *AIFM1* KO or AIFM1 variants (AIFM1$^{TCS}$ and AIFM1$^{TCS/TEV}$)-expressing cells is shown along with the Flp-In-293 cells (WT). Graphs are mean ± SD of nine independent biologic experiments. Dashed lines indicate injections of oligomycin (2 μM), FCCP (0.5 μM), and antimycin A/rotenone (0.5 μM each). (I) Growth rate of the AIFM1 variant cells in a galactose-containing medium. Each cell was seeded at $4 \times 10^5$ cells per well (in a six-well plate) in a customized galactose-containing medium to switch the cellular biogenesis to rely on mitochondrial respiratory activity, and the number of cells was counted for up to 8 days after seeding. Data shown are mean ± SD ($n = 3$ biologic replicates). **$p < 0.01$ and *$p < 0.05$ (by two-way ANOVA followed by Bonferroni's multiple comparisons test). Source data are available online for this figure.

with the TIM23 translocase, which coordinately regulates protein import into mitochondria.

## Physiologic impact of AIFM1 cleavage in vivo

To evaluate whether our in vitro observations were physiologically relevant, we finally investigated the impact of AIFM1 cleavage in vivo. When mice were challenged with a mouse-adapted influenza A virus PR8 strain (H1N1 subtype; a negative-sense, single-stranded RNA virus of the *Orthomyxoviridae* family), a higher pathogenicity was observed in the lung. Hyperemia and infiltration of inflammatory cells around the bronchi with desquamation of the bronchial epithelium in the entire lung tissue were observed in mice infected with PR8 (Fig. EV6A, right panels, arrowheads). Analysis of the lung tissue by transmission electron microscopy revealed an altered mitochondrial ultrastructure with disrupted cristae morphology from the PR8-infected mice (Fig. 7A–F). Notably, we found increased AIFM1 processing in the PR8-infected lungs, as well as a slight reduction in L-OPA1 isoforms coupled with OMA1 activation (Fig. 7G). We made a similar observation of stress-induced OMA1-mediated cleavage of AIFM1 in viral-infected MEFs, showing that their actions are accompanied by OMA1 enzymatic activity (Fig. EV6B).

To further explore the role of AIFM1 topology in OXPHOS subunit biogenesis in vivo, we examined the fidelity of the entire proteome in the tissue (Dataset EV6). We found that the lungs from PR8-infected mice had significantly higher levels of proteins from immune response pathways (Fig. EV6C), whereas the levels of mitochondrial proteins corresponding to OXPHOS complex subunits and the MIA40 pathway were much lower in the isolated tissues than in uninfected mice (Figs. 7H and EV6D). In line with reports of bioenergetic changes in influenza A virus-infected lungs (Nolan et al, 2021; Ren et al, 2021), our results also show that the glycolytic pathway is upregulated in PR8-infected mouse lung tissues (Fig. EV6E). Consequently, high lactate levels were observed in the lungs relative to those in uninfected mice (Fig. 7I). Taken together, these results underscore the physiologic importance of

mitochondrial stress sensing and cellular energetics, which alter the topology of AIFM1.

## Discussion

OMA1 is a stress-sensitive mitochondrial metalloprotease that belongs to the membrane-integrated peptidase M48 superfamily (López-Pelegrín et al, 2013). It plays diverse roles in cellular processes throughout the organism, although its catalytic molecular mechanisms especially how it recognizes and processes substrates remain poorly understood. In the present study, we used in vitro and in vivo multiproteomic and biochemical approaches to determine the interactome of the OMA1 peptidase in mitochondria and identified AIFM1 as an OMA1-targeted substrate under various stress conditions. OMA1-mediated cleavage of AIFM1 leads to membrane dislocation of the substrate and downregulates the import of OXPHOS-related proteins into mitochondria, ultimately impairing mitochondrial bioenergetics.

OMA1-mediated AIFM1 processing is strictly triggered by mitochondrial stress. Notably, OMA1 substrates do not share a conserved cleavage motif (Appendix Fig. S2B) (Ishihara et al, 2006; Guo et al, 2020; Ahola et al, 2024), and the cleavage site in AIFM1 targeted by OMA1 does not match those targeted by μ-calpain or cathepsins during apoptosis (Fig. EV2E) (Susin et al, 1999; Polster et al, 2005; Yuste et al, 2005). Therefore, the previously reported cleavage of AIFM1 during apoptosis and the OMA1-mediated pathway we observed here are mechanistically distinct. However, it is conceivable that the apoptosis-related attenuation of ΔΨ$_m$ (Ly et al, 2003) might activate OMA1 and consequently leading to AIFM1 processing. Additionally, it should be noted that AK2, a mediator of intrinsic apoptosis (Lee et al, 2007), interacts with AIFM1 and is important for metabolic adaptations (Rothemann et al, 2025; Schildhauer et al, 2025). It is interesting to compare our results with the finding that AK2 cleavage by OMA1 also removes the IAP-binding motif in AK2 (Appendix Table S1), thereby destabilizing the protein; this destabilization may be required for AK2's role in apoptosis.

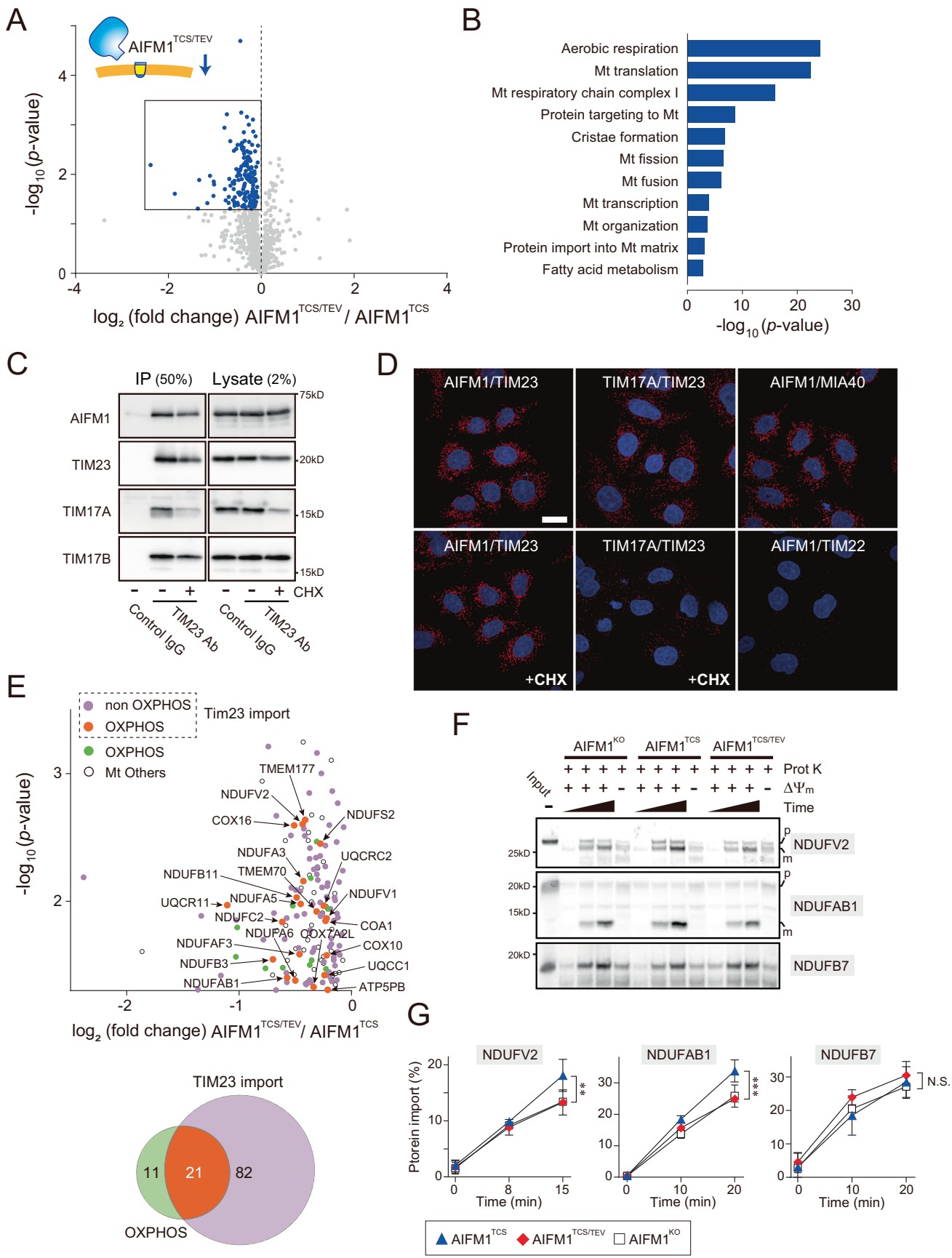

**Figure 6. AIFM1 cooperates with TIM23 translocase and ensures substrate import.**

(A) Volcano plot of mitochondrial protein (MitoCarta3.0) changes between AIFM1$^{TCS}$ and AIFM1$^{TCS/TEV}$ cells. Blue plot area in the volcano plot shows decreased proteins [$n = 4$ biologic replicates, $p < 0.05$ (by Student's $t$ test)] in AIFM1$^{TCS/TEV}$, and several mitochondrial proteins in the boxed area are annotated in (E). See also Dataset EV4. (B) GO enrichment analysis of the mitochondrial proteome from the boxed area in (A), showing significantly downregulated biologic processes in AIFM1$^{TCS/TEV}$ (by modified Fisher's exact test). (C) Interaction of endogenous AIFM1 with TIM23 translocase. Lysates of HeLa cells treated with or without CHX (10 μg/mL) were immunoprecipitated with either an anti-TIM23 monoclonal antibody or a control IgG (anti-GFP), followed by western blot analysis with an anti-AIFM1 polyclonal antibody (top). The IP and lysate samples were loaded at 50% and 2% of the input samples, respectively. As a control, endogenous TIM23 was to co-immunoprecipitate with TIM17A and TIM17B, two components of the TIM23 translocase. Note that the TIM17A blot was also used as a marker to verify the effect of CHX in cells. (D) A proximity ligation assay (PLA) confirmed the endogenous interaction between AIFM1 and TIM23 in HeLa cells. PLA was performed to demonstrate the interaction between AIFM1 and TIM23 in HeLa cells using pairs of antibodies (AIFM1, polyclonal; TIM23, monoclonal), along with positive (TIM17A–TIM23 and AIFM1–MIA40) or negative (AIFM1–TIM22) control conditions, as indicated. The PLA spots appear red, and the nuclear DNA in the cells appears blue when counterstained with DAPI. Treating the cells with CHX eliminates the possibility of transient associations between newly synthesized substrates and the TIM23 translocase. Note that TIM17A was also used as a marker to verify the effect of CHX (protein degradation) in cells. Scale bar, 20 μm. (E) Enlarged box area in (A). A total of 123 mitochondrial proteins are plotted, 103 of which are TIM23 substrates, including the 21 OXPHOS components (orange). See also Venn diagram in the bottom and Dataset EV4. (F) In organello import assay with TIM23 substrates (NDUFV2 and NDUFAB1). In vitro radiolabeled NDUFV2 or NDUFAB1 ($^{35}$S-labeled) were incubated with mitochondria isolated from the *AIFM1* KO cells or cells expressing AIFM1 variant (AIFM1$^{TCS}$ and AIFM1$^{TCS/TEV}$). Non-imported proteins were removed by treatment with proteinase K (Prot K), and imported proteins at different time points were analyzed by SDS-PAGE followed by autoradiography. As a control, the import reaction was also performed on mitochondria treated with uncouplers to dissipate the $\Delta\Psi_m$. Radioactive NDUFB7 was used as a control for TIM23-independent substrate in this assay. NDUFV2 and NDUFAB1 import is affected by the loss or membrane dislocation of AIFM1, but NDUFB7 import is less affected. *p*, precursor form; *m*, mature form. (G) Signals in (F) were quantified using ImageJ and the amount of each imported protein at different time points was plotted. Data shown are mean ± SD ($n = 3$ biologic replicates). **$p < 0.01$, ***$p < 0.001$, and N.S., not significant, respectively (by two-way ANOVA followed by Sidak's multiple comparisons test). Each statistic compares AIFM1$^{TCS}$ (blue) vs AIFM1$^{TCS/TEV}$ (red) and AIFM1$^{TCS}$ (blue) vs. AIFM1$^{KO}$ (white), respectively. Source data are available online for this figure.

Our biochemical data show that the OMA1-mediated AIFM1-processing kinetics are significantly slower than those of the conventional substrate, OPA1 (Fig. 3B). Like AIFM1 proteolysis, the kinetic profile of OMA1-induced DELE1 cleavage is also slower (Fessler et al, 2020), although there is no apparent interaction between these two substrates. Because elimination of OPA1, but not DELE1 (Fig. EV3E), would accelerate OMA1-mediated processing of AIFM1 (Fig. 3E), we propose a model that the OMA1 peptidase has substrate selectivity and that OPA1 has a much higher affinity than other substrates (and/or differential abundance), which is also supported by the fact that OPA1 is constitutively processed at steady state (Ishihara et al, 2006; Song et al, 2007; Ehses et al, 2009; Head et al, 2009). This may explain why AIFM1 cleavage is observed in depolarized *OPA1*-depleted cells at early time points, increasing its susceptibility to OMA1 attack (Fig. EV3B). Additionally, the physical interaction between AIFM1 and MIA40 might also influence AIFM1's susceptibility to OMA1-dependent proteolytic processing by stabilizing the dimeric structure of AIFM1 in mitochondria and increasing folding stability (Mussulini et al, 2025) with less flexibility around the OMA1 cleavage site. Although the catalytic reaction of OMA1 in AIFM1 processing appears to act in *cis* (i.e., two molecules located on the same membrane), we cannot exclude the possibility that each molecule exists on a different membrane (*trans*) and proteolysis occurs when the membranes come closer together (Fig. EV1F), allowing the molecules to persist for an extended period of time. Together, this unusual catalytic mechanism of OMA1 regulates a rapid change in mitochondrial dynamics when stress stimuli are induced, while subsequently leading to an alternative stress response program in cells, ensuring functional plasticity at the cellular level.

While examining the role of AIFM1 involvement in the biogenesis, we discovered that AIFM1 functionally associates with the TIM23 complex, which coordinately regulates the translocase-dependent import of substrates into the matrix (Fig. 6). The function of the TIM23 complex is critically dependent on ATP hydrolysis and the $\Delta\Psi_m$ (Neupert and Brunner, 2002; Chacinska et al, 2009; Mokranjac and Neupert, 2010; van der Laan et al, 2010), so direct investigations of the mechanistic action of AIFM1

involved in the TIM23-dependent import pathway under the $\Delta\Psi_m$-less condition have been challenging. Our platform, which generated the AIFM1 cleaved protein without inducing any mitochondrial stress (e.g., depolarization) overcomes these limitations and allowed us to reveal the regulatory role of AIFM1 involved in TIM23-mediated mitochondrial protein import. Our results suggest that the release of AIFM1 from the MIM attenuates some interactions between AIFM1 and the TIM23 complex, and that this dissociation ultimately downregulates the mitochondrial proteome and impairs cell growth. Indeed, the membrane dislocation of AIFM1 decreased protein levels of 32 OXPHOS-related components at steady state, with >60% of them being TIM23-dependent pathways (Fig. 6E). Notably, the stress-induced change in the AIFM1 topology in mitochondria and the subsequent alteration of the OXPHOS-related mitochondrial proteome were also observed in the lungs of virus-infected mice. These results are consistent with previous observations implicating AIFM1 in OXPHOS biogenesis (Vahsen et al, 2004; Urbano et al, 2005; Pospisilik et al, 2007; Delavallée et al, 2020; Wang et al, 2021; Salscheider et al, 2022; Peker et al, 2023) and are supported by the finding that membrane-anchored AIFM1 allosterically activates MIA40, which is also involved in regulating OXPHOS metabolism (Brosey et al, 2025) (Figs. EV4I and EV6D). These results clarify that AIFM1 is an essential import platform for mitochondrial disulfide relay components through the MIA40 pathway (Reinhardt et al, 2020; Salscheider et al, 2022; Peker et al, 2023; Brosey et al, 2025) and for OXPHOS components translocated into the matrix.

AIFM1 mutations are reported in humans with autosomal recessive mitochondrial disorders (Bano and Prehn, 2018; Wischhof et al, 2022), although their precise physiologic functions and the biochemical properties of these mutations remain elusive. While these *AIFM1* mutation-associated disorders present with variable clinical features, many *AIFM1* mutant alleles are generally associated with respiratory defects (Wischhof et al, 2022). It is interesting to consider our results with previous findings that two AIFM1 variants (deletion of residue Arg$^{201}$; R201 del, and substitution of residue Glu 493 to Val; E493V), both of which fold nearly identically to the WT protein, exhibit higher sensitivity

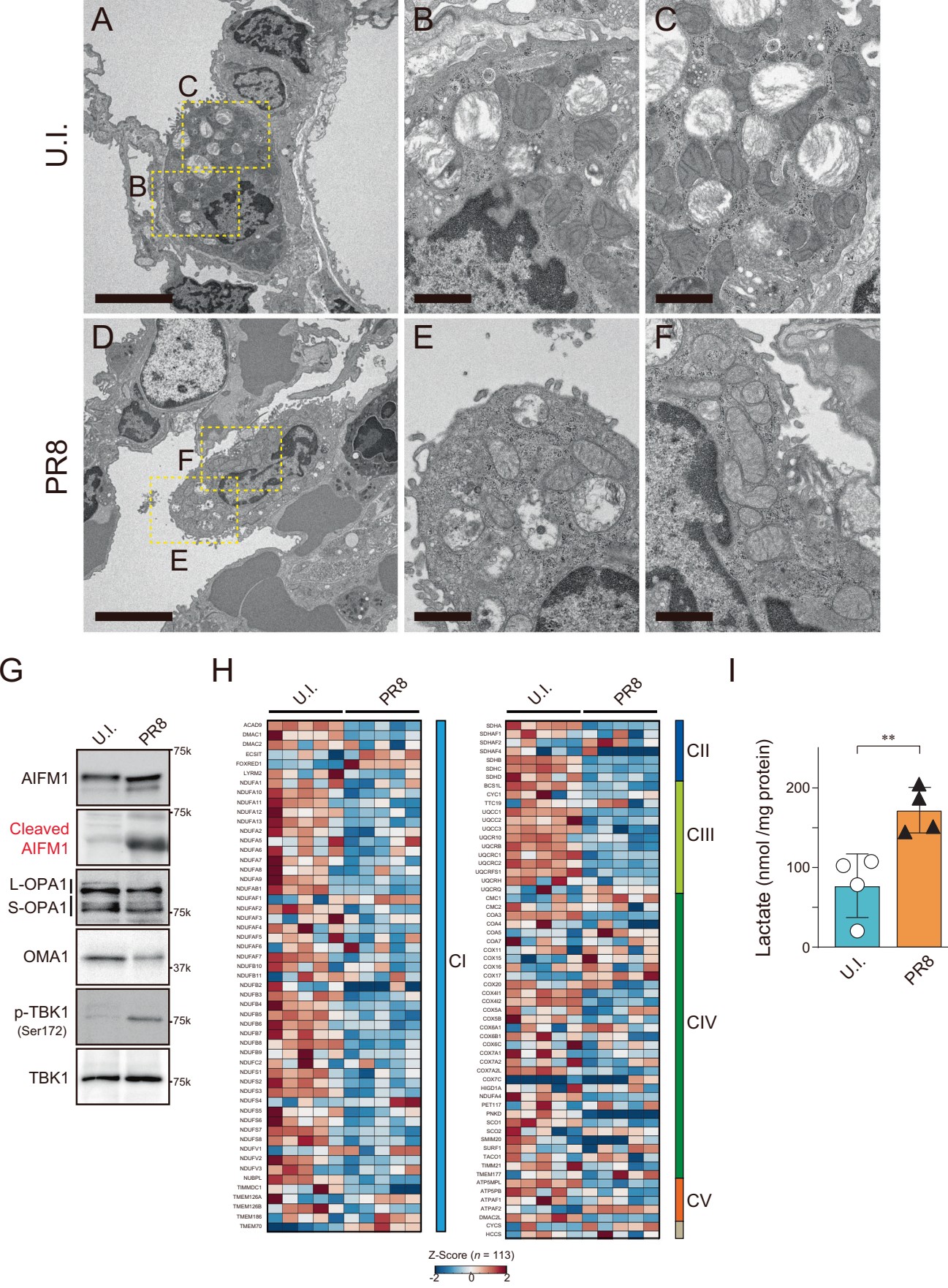

**Figure 7. AIFM1 processing impairs the biogenesis of the mitochondrial proteome in vivo.**

(A–F) Representative transmission electron microscopy (TEM) images of lung tissues from 6-week-old uninfected (A) and PR8-infected (D) mice. The rectangles in panels (A) and (D) indicate regions magnified in (B, C), or (E, F), respectively. Note the images of the mouse lung tissue showing collapsed cristae morphology in PR8-infected lung (see E and F). Scale bars: 5 μm in (A, D); 1 μm in (B, C, E, F). (G) Viral infection leads to increased AIFM1 processing in mouse lung tissue. Lung lysates from uninfected (U.I.) or PR8-infected (PR8) mice were analyzed by immunoblotting using the indicated antibodies. We assessed phosphorylation of the Ser172 residue on TBK1 as a marker of viral infection in the mice. (H) Alterations in the mitochondrial proteome (representing subunits of the OXPHOS complexes, Z-scores) in lung tissues from uninfected (U.I.) and PR8-infected (PR8) mice ($n = 5$) were determined by mass spectrometry. Heatmap (Z-scores): minimum ($-2$), blue; maximum (2), red. See also Dataset EV6. (I) Quantification of lactate in mouse lung tissue. Changes in lactate levels (nmol/mg protein) in lung tissue homogenates from uninfected (U.I.) or PR8-infected (PR8) mice were measured using a lactate assay kit. Data are shown as mean ± SD ($n = 4$), and **$p < 0.01$ (by Student's $t$ test). Source data are available online for this figure.

to tryptic digestion in vitro (Ghezzi et al, 2010; Rinaldi et al, 2012), suggesting that subtle conformational changes due to the removal or substitution of residues in AIFM1 destabilize the protein's stability, making it more susceptible to proteolytic attack. Indeed, western blot analysis of three alleles in patient fibroblasts reveals a reduction in the abundance of AIFM1 (Ardissone et al, 2015; Diodato et al, 2016; Moss et al, 2021). Although it remains to be directly demonstrated that disease-associated AIFM1 mutations or deletion increase protein destabilization, the present observations raise the possibility that some mitochondrial proteases are involved in the degradation process, leading to AIFM1 inactivation in mitochondrial bioenergetics, and that OMA1 may fulfill this role. Accordingly, patients with *AIFM1* mutations who experience viral infections may exhibit accelerated AIFM1 processing via the activated OMA1 pathway, which ultimately impairs respiration. Further investigation into the degradation process of AIFM1 variants mediated by OMA1 and/or other mitochondrial proteases and its contribution to human disease, including infection, is warranted.

# Methods

## Reagents

Carbonyl cyanide *m*-chlorophenyl hydrazone (CCCP, Cat# c2759), antimycin A (Cat# A8674), rotenone (Cat# R8875), hygromycin B (Cat# H3274), puromycin (Cat# P8833), hydrogen peroxide (Cat# H1009), anti-*c*-Myc agarose beads (Cat# A7470), and Protein A Sepharose beads (Cat# P3391) were purchased from Sigma-Aldrich (St. Louis, MO). FCCP (Cat# HY-100410) and oligomycin A (Cat# sc-201551) were supplied from either MedChemExpress (Monmouth Junction, NJ) or Santa Cruz Biotechnology (Dallas, TX). Valinomycin (Cat# V1644) and cycloheximide (Cat# 037-20991) were obtained from Thermo Fisher Scientific (Waltham, MA) and Wako Pure Chemical Industries (Tokyo, Japan), respectively. All other reagents used in the study were of biochemical research grade and are listed in the Reagents and Tools Table.

## Cell culture and cell treatments

All cell lines used in the study are listed in the Reagents and Tools Table. HeLa cells were used for siRNA-mediated knockdown experiments and the Platinum-A retroviral packaging cell line (Cell Biolabs, San Diego, CA) was used for retrovirus production. Unless otherwise indicated, cells were maintained in Dulbecco's modified

**Reagents and tools table**

| Reagent/resource | Reference or source | Identifier or catalog number |
|---|---|---|
| **Experimental models** | | |
| **Cell line** | **Inserted gene/ Tag (C-terminal)** | **Resource** |
| HEK293 (*H. sapiens*) | — | Koshiba et al, 2011 |
| Flp-In T-REx-293 (*H. sapiens*) (Flp-In™ T-Rex™ 293) | — | Thermo Fisher, R78007 |
| Flp-In T-REx-293-mock | pcDNA5/FRT (empty) | This work |
| Flp-In T-REx-293 *AIFM1* KO | Deletion of AIFM1 | Salscheider et al, 2022 |
| Flp-In T-REx-293 *AIFM1* KO-mock | pcDNA5/FRT (empty) | This work |
| Flp-In T-REx-293 *AIFM1* KO-*AIFM1*$^{WT}$ | Murine AIFM1 (A.A.1-612)/ Myc | This work |
| Flp-In T-REx-293 *AIFM1* KO-*AIFM1*$^{Δ97-99}$ | Murine AIFM1 (lacking A.A.97-99)/ Myc | This work |
| Flp-In T-REx-293 *AIFM1* KO-*AIFM1*$^{TCS}$ | Murine AIFM1 (R97►TEV site)/ Myc | This work |
| Flp-In T-REx-293 *AIFM1* KO-*AIFM1*$^{TCS/TEV}$ | Murine AIFM1 (R97►TEV site)/ Myc IMS-TEV protease/ HA | This work |
| Flp-In T-REx-293 *AIFM1* KO-*AIFM1*$^{DKP}$ | Murine AIFM1 (T503►D/V504►K/G505►P)/ Myc | This work |
| HeLa (*H. sapiens*) | — | Yoshizumi et al, 2014 |
| Platinum-A retroviral packaging cell line (Amphotropic) | Viral gag, pol, and env | Cell Biolabs, RV-102 |
| MEF wild type (*M. musculus*) | — | Quirós et al, 2012 |

| Reagent/resource | Reference or source | Identifier or catalog number |
|---|---|---|
| MEF-*AIFM1*$^{WT}$ | Murine AIFM1 (A.A.1-612)/ Myc | This work |
| MEF-*AIFM1*$^{Δ97-99}$ | Murine AIFM1 (lacking A.A.97-99)/ Myc | This work |
| MEF-*AIFM1*$^{R97A}$ | Murine AIFM1 (R97►A)/ Myc | This work |
| MEF-*AIFM1*$^{R97E}$ | Murine AIFM1 (R97►E)/ Myc | This work |
| MEF *OMA1* KO | Deletion of OMA1 | Quirós et al, 2012 |
| MEF *OMA1* KO-*OMA1*$^{WT}$ | Murine OMA1 (A.A.1-521)/ Myc | This work |
| MEF *OMA1* KO-*OMA1*$^{E324Q}$ | Murine OMA1 (E324►Q)/ Myc | This work |
| MEF *OMA1* KO-*AIFM1*$^{WT}$ | Murine AIFM1 (A.A.1-612)/ Myc | This work |
| MEF *OMA1* KO-*AIFM1*$^{Δ97-99}$ | Murine AIFM1 (lacking A.A.97-99)/ Myc | This work |
| MEF *OMA1* KO-*AIFM1*$^{R97A}$ | Murine AIFM1 (R97►A)/ Myc | This work |
| MEF *OMA1* KO-*AIFM1*$^{R97E}$ | Murine AIFM1 (R97►E)/ Myc | This work |
| MEF *OMA1* KO-*OMA1*$^{WT}$/*DELE1* | Murine OMA1 (A.A.1-521)/ Myc<br>Murine DELE1 (A.A.1-510)/ HA | This work |
| MEF *OMA1* KO-*OMA1*$^{E324Q}$/*DELE1* | Murine OMA1 (E324►Q)/ Myc<br>Murine DELE1 (A.A. 1-510)/ HA | This work |
| MEF *YME1L* KO | Deletion of YME1L | Anand et al, 2014 |
| MEF *OMA1* and *YME1L* DKO | Deletion of both OMA1 and YME1L | Anand et al, 2014 |
| MEF *OMA1* and *YME1L* DKO-*OMA1*$^{WT}$ | Murine OMA1 (A.A.1-521)/ Myc | This work |
| MEF *OMA1* and YME1L DKO-*YME1L* | Murine YME1L (A.A.1-715)/ HA | This work |
| MEF *OPA1* KO | Deletion of OPA1 | Song et al, 2007 |
| MEF *OPA1* KO-*OPA1*$^{V1}$ | Rat OPA1 variant 1 (A.A.1-960)/ FLAG | Yoshizumi et al, 2017 |
| MEF *OPA1V1* | Deletion of exons 4b and 5b in OPA1 locus | Ahola et al, 2024 |
| MEF *OPA1V1*$^{Δ4}$ | Deletion of A.A.193-197 (FRAT) in OPA1 variant 1 | Ahola et al, 2024 |
| MEF *DELE1* KO | Deletion of DELE1 | Ahola et al, 2022 |
| MEF *Mfn1* and *Mfn2* DKO | Deletion of both Mfn1 and Mfn2 | Koshiba et al, 2004 |
| MEF *Drp1* KO | Deletion of Drp1 | Ishihara et al, 2009 |
| MEF *MIC60 shRNA* | Silence of MIC60 | This work |
| **Organisms/strains** | | |
| Mouse: WT: C57BL/6 J (female; six-week-old) | Japan SLC, Inc. | C57BL/6JJmsSlc |
| *Escherichia coli* BL21(DE3) | New England Biolabs | C2527H |
| Sendai virus (Strain: Cantell) | ATCC | VR-907 |
| Influenza A virus strain A/Puerto Rico/8/1934 (PR8) | — | Nagai et al, 2023 |
| Encephalomyocarditis virus (EMCV) | — | Koshiba et al, 2011 |
| **Recombinant DNAs** | **Inserted gene/ Tag (C-terminal)** | **Resource** |
| pcDNA5/FRT | — | Thermo Fisher, V601020 |
| pcDNA5/FRT-*AIFM1*$^{WT}$ | mAIFM1 (A.A.1-612)/ Myc or HA | This work |
| pcDNA5/FRT-*AIFM1*$^{Δ97-99}$ | mAIFM1 (ΔA.A.97-99)/ Myc or HA | This work |
| pcDNA5/FRT-*AIFM1*$^{Δ116-118}$ | mAIFM1 (ΔA.A.116-118)/ Myc or HA | This work |
| pcDNA5/FRT-*AIFM1*$^{TCS}$ | mAIFM1 (R97►TEV site)/ Myc or HA | This work |
| pcDNA5/FRT-*AIFM1*$^{E96A}$ | mAIFM1 (E96►A)/ Myc or HA | This work |
| pcDNA5/FRT-*AIFM1*$^{R97A}$ | mAIFM1 (R97►A)/ Myc or HA | This work |
| pcDNA5/FRT-*AIFM1*$^{R97E}$ | mAIFM1 (R97►E)/ Myc or HA | This work |
| pcDNA5/FRT-*AIFM1*$^{V98A}$ | mAIFM1 (V98►A)/ Myc or HA | This work |
| pcDNA5/FRT-*AIFM1*$^{E96A/R97A/V98A}$ | mAIFM1 (E96/R97/V98►A)/ Myc or HA | This work |
| pcDNA5/FRT-*AIFM1*$^{T503D/V504K/G505P}$ | mAIFM1 (T503►D/V504►K/G505►P)/ Myc | This work |
| pOG44 | Flp recombinase | Thermo Fisher, V600520 |

| Reagent/resource | Reference or source | Identifier or catalog number |
|---|---|---|
| pcDNA3.1(-) | — | Thermo Fisher, V79520 |
| pcDNA3.1(-)/*OMA1*<sup>WT</sup> | mOMA1 (A.A.1-521)/ Myc or HA | This work |
| pcDNA3.1(-)/*OMA1*<sup>E324Q</sup> | mOMA1 (E324▸Q)/ Myc or HA | This work |
| pcDNA3.1(-)/*IMS-TEV* | TEV protease fused with rAIFM1 (A.A.1-95)/ HA | This work |
| pcDNA3.1(+)/*Su9-TEV* | TEV protease fused with ATPase subunit 9 (A.A.1-69)/ HA | Sekine et al, 2019 |
| pcDNA3.1(-)/BAX | HA/ hBAX (A.A.1-192) | This work |
| pcDNA3.1(-)/Bcl-xL | Myc/ hBcl-xL (A.A.1-233) | This work |
| pET28a/*OMA1* | Human OMA1 (A.A.1-524)/ intact | This work |
| pMXs-puro | — | Cell Biolabs, RTV-012 |
| pMXs-puro/*OMA1*<sup>WT</sup> | mOMA1 (A.A.1-521)/ Myc | This work |
| pMXs-puro/*OMA1*<sup>E324Q</sup> | mOMA1 (E324▸Q)/ Myc | This work |
| pMXs-puro/*YME1L* | mYME1L (A.A.1-715)/ HA | This work |
| pMXs-puro/*DELE1* | mDELE1 (A.A.1-510)/ HA | This work |
| pMXs-puro/*IMS-TEV* | TEV protease fused with AIFM1 (A.A.1-95)/ HA | This work |
| pMXs-puro/*AIFM1*<sup>Δ1−97</sup> | mAIFM1 (ΔA.A.1–97)/ Myc | This work |
| pMXs-puro/*OPA1*<sup>V1</sup> | Rat OPA1 variant 1 (A.A.1-960)/ FLAG | Yoshizumi et al, 2017 |
| pMK-RQ/*NDUFV2* | hNDUFV2 (A.A. 1-249)/ — | This work |
| pMK-RQ/*NDUFAB1* | hNDUFB11 (A.A. 1-156)/ — | This work |
| pGEM4/*NDUFB7* | hNDUFB7 (A.A. 1-137)/ — | This work |
| **Antibodies** | | |
| Anti-AFG3L2, rabbit polyclonal | Proteintech | 14631-1-AP |
| Anti-AIFM1 (E-1), mouse monoclonal | Santa Cruz Biotechnology | sc-13116 |
| Anti-AIFM1, rabbit polyclonal | Proteintech | 17984-1-AP |
| Anti-AIFM1 (cleaved site), rabbit polyclonal (mouse specific) | This study | N/A |
| Anti-ATF4 (D4B8), rabbit monoclonal | Cell Signaling Technology | Cat# 11815 |
| Anti-β-actin (C4), mouse monoclonal | Santa Cruz Biotechnology | sc-47778 |
| Anti-Caspase-3, rabbit polyclonal | Cell Signaling Technology | Cat# 9662 |
| Anti-CHCHD4/MIA40, rabbit polyclonal | Proteintech | 21090-1-AP |
| Anti-COX IV, rabbit polyclonal | Cell Signaling Technology | Cat# 4844 |
| Anti-COX IV (3E11), rabbit monoclonal | Cell Signaling Technology | Cat# 4850 |
| Anti-SMAC/DIABLO, rabbit polyclonal | Proteintech | 10434-1-AP |
| Anti-DNA, mouse monoclonal | Millipore | CBL186 |
| Anti-DRP1, mouse monoclonal | BD Transduction Laboratories | Cat# 611113 |
| Anti-HA (HA.11), mouse monoclonal | Covance | MMS-101P |
| Anti-HSP60 (D307), rabbit polyclonal | Cell Signaling Technology | Cat# 4870 |
| Anti-HTRA2/Omi, rabbit polyclonal | R&D Systems | AF1458 |
| Anti-MAVS, rabbit polyclonal | Koshiba et al, 2011 | N/A |
| Anti-MFN2 (XX-1), mouse monoclonal | Santa Cruz Biotechnology | sc-100560 |
| Anti-Mitofilin/MIC60, mouse monoclonal | Abcam | ab110329 |
| Anti-mtHSP70 (JG1), mouse monoclonal | Thermo Fisher Scientific | Cat# MA3-028 |
| Anti-Myc (9E10), mouse monoclonal | Covance | MMS-150P |
| Anti-Myc (9B11), mouse monoclonal | Cell Signaling Technology | Cat# 2276 |
| Anti-Myc (My3), mouse monoclonal | MBL Life Science | M192-3 |
| Anti-Myc (A-14), rabbit polyclonal | Santa Cruz Biotechnology | sc-789 |

| Reagent/resource | Reference or source | Identifier or catalog number |
|---|---|---|
| Anti-OMA1 (P-19), goat polyclonal | Santa Cruz Biotechnology | sc-168844 |
| Anti-OMA1 (H-11), mouse monoclonal | Santa Cruz Biotechnology | sc-515788 |
| Anti-OPA1, mouse monoclonal | BD Transduction Laboratories | Cat# 612606 |
| Anti-OXPHOS (cocktail), mouse monoclonal | Abcam | ab110413 |
| Anti-PARL, rabbit polyclonal | Proteintech | 26679-1-AP |
| Anti-PARP, rabbit polyclonal | Cell Signaling Technology | Cat# 9542 |
| Anti-PARP (cleaved form; Asp214) (D64E10), rabbit monoclonal | Cell Signaling Technology | Cat# 5625 |
| Anti-PHB1, rabbit monoclonal | Abcam | ab75766 |
| Anti-PHB2 (H-80), rabbit polyclonal | Santa Cruz Biotechnology | sc-67045 |
| Anti-ROMO1, rabbit polyclonal | Proteintech | 24200-1-AP |
| Anti-SDHA, rabbit polyclonal | Proteintech | 14865-1-AP |
| Anti-TBK1/NAK, rabbit monoclonal | Abcam | ab40676 |
| Anti-Phospho-TBK1/NAK (Ser172) (D52C2), rabbit monoclonal | Cell Signaling Technology | Cat# 5483 |
| Anti-TIM17A, rabbit polyclonal | Ishihara et al, 1998 | N/A |
| Anti-TIM17A (C1C3), rabbit polyclonal | GeneTex | GTX108280 |
| Anti-TIM17B, rabbit polyclonal | Novus Biologicals | NBP1-84036 |
| Anti-TIM22, rabbit monoclonal | Abcam | ab167423 |
| Anti-TIM23, mouse monoclonal | BD Transduction Laboratories | Cat# 611223 |
| Anti-TIM29, rabbit polyclonal | Proteintech | 25652-1-AP |
| Anti-TIM44, rabbit polyclonal | Proteintech | 13859-1-AP |
| Anti-TIM50, rabbit polyclonal | Proteintech | 22229-1-AP |
| Anti-α-tubulin, rabbit polyclonal | MBL Life Science | PM054 |
| Anti-YME1L, rabbit polyclonal | Proteintech | 11510-1-AP |
| Alexa Fluor 488 goat anti-mouse IgG | Invitrogen | A-11001 |
| Alexa Fluor 488 goat anti-rabbit IgG | Invitrogen | A-11008 |
| Alexa Fluor 568 goat anti-mouse IgG | Invitrogen | A-11004 |
| Alexa Fluor 568 goat anti-rabbit IgG | Invitrogen | A-11011 |
| Cy-3-conjugated sheep anti-mouse IgG | Jackson ImmunoResearch | Cat# 515-165-003 |
| Goat Anti-Mouse IgG (H + L)-HRP | Bio-Rad | Cat# 1706516 |
| Goat Anti-Rabbit IgG (H + L)-HRP | Bio-Rad | Cat# 1706515 |
| Donkey Anti-Goat IgG-HRP | Santa Cruz Biotechnology | sc-2020 |
| **Oligonucleotides** | **Sequence** | **Restriction site** |
| TK430: Reverse for *hBcl-XL* with stop codon | tttGATATCtcatttccgactgaagag | *Eco*RV |
| TK614: Forward for *hBAX* | aaGCGGCCGCcatggacgggtccggggagcagccc | *Not*I |
| TK615: Reverse for *hBAX* with stop codon | tttGATATCtcagcccatcctcttccagatgg | *Eco*RV |
| TK718: Forward for *hBcl-XL* | aaGCGGCCGCcatgtctcagagcaaccggg | *Not*I |
| TK921: Forward for *hOMA1* | aaagctagcaCCATGGggagcttcatctgtggattgc | *Nco*I |
| TK890: Reverse for *hOMA1* with stop codon | tttGATATCtcaactgcccgttcttttctcaac | *Eco*RV |
| TK950: Forward for *mOMA1* with Kozak sequence | aaaGCGGCCGCaccatggggagcctcctttatggactgc | *Not*I |
| TK889: Reverse for *mOMA1* | tttGGTACCgcctgcagttctcttctctagg | *Kpn*I |
| **TK951: forward for** mOMA1 *(E324►Q)* | ctgggccatCagatcgcacacgc | — |
| **TK952: reverse for** mOMA1 *(E324►Q)* | gcgtgtgcgatctGatggcccag | — |
| TK956: forward for *mYME1L* with Kozak sequence | aaGCTAGCaccatggggttctccctgtcgagcactgtg | *Nhe*I |
| TK957: reverse for *mYME1L* | tttGGTACCtctcacttccaatttctttccctc | *Kpn*I |
| TK1378: forward for *mDELE1* with Kozak sequence | aaaGCGGCCGCaccatggggtggcgcctgacagggatcctggggcga | *Not*I |

| Reagent/resource | Reference or source | Identifier or catalog number |
|---|---|---|
| TK1379: reverse for *mDELE1* | aaaGGTACCgccgaaacctagtcttacgaggctcctttcc | *Kpn*I |
| TK1038: forward for *mAIFM1* with Kozak sequence | tttGCGGCCGCaccATGgggttccggtgtggaggcctggcggg | *Not*I |
| TK1039: reverse for *mAIFM1* | aaaGGTACCatcttcatgaatgttgaagag | *Kpn*I |
| TK1316: forward for *mAIFM1 (A.A.100)* | ggattaggactgtccccagaagag | — |
| TK1317: reverse for *mAIFM1 (A.A.96)* | ttcattgtatcttttttggtcttc | — |
| TK1318: forward for *mAIFM1 (A.A.119)* | ggaggctcagttcctcagatcagg | — |
| TK1319: reverse for *mAIFM1 (A.A.115)* | ggaggcaatggctcttctctgtttc | — |
| TK1331: forward for *mAIFM1 (R97►A)* | gatacaatgaaGCagtgatgggatt | — |
| TK1332: reverse for *mAIFM1 (R97►A)* | aatcccatcactGCttcattgtatc | — |
| TK1333: forward for *mAIFM1 (V98►A)* | tacaatgaaagagCgatgggattagg | — |
| TK1334: reverse for *mAIFM1 (V98►A)* | cctaatcccatcGctctttcattgta | — |
| TK1335: forward for *mAIFM1 (E96►A)* | aaaagatacaatgCaagagtgatggg | — |
| TK1336: reverse for *mAIFM1 (E96►A)* | cccatcactcttGcattgtatctttt | — |
| TK1337: forward for *mAIFM1 (R97/V98/M99►A)* | GCCGCAGCGggattaggactgtccccagaagag | — |
| TK1341: forward for *mAIFM1 (R97►E)* | GAagtgatgggattaggactgtcc | — |
| TK1384: forward for *mAIFM1 (R97►ENLYFQG;* TEV site*)* | GAGAATCTCTACTTCCAAGGTgtgatgggattaggactgtcccc | — |
| TK1393: forward for *TEV* protease | tttGGTACCtccagcttgtttaagggaccacg | *Kpn*I |
| TK1477: forward for *mAIFM1 (T503►D/V504►K/G505►P)* | GACAAACCtgtttttgcaaaagcaactgc | — |
| TK1478: reverse for *mAIFM1 (T503►D/V504►K/G505►P)* | gggcaaactactatccaccagacc | — |
| TK2001: forward for *pGEM4* | tccggtctccctatagtgag | — |
| TK2002: reverse for *pGEM4* | ccgtgtattctatagtgtcaccta | — |
| TK2003: forward for *hNDUFB7* | ttaggtgacactatagaatacacgggccaccatgggggcccacctggtc | — |
| TK2004: reverse for *hNDUFB7* | tacgactcactatagggagaccggactacagggccaccttggggtcc | — |
| mtDNA qPCR: forward for *human ND5* | tcgaaaccgcaaacatatca | — |
| mtDNA qPCR: reverse for *human ND5* | caggcgtttaatggggttta | — |
| Nuclear DNA qPCR: forward for *human ACTB* | ctgtggcatccacgaaacta | — |
| Nuclear DNA qPCR: reverse for *human ACTB* | agtacttgcgctcaggagga | — |
| **siRNAs** | **Sequence (sense)** | **Source/identifier** |
| *human OMA1* | 5′-gaaugaccucugacuuuaauucautt | Qiagen/SI02779343 |
| *human YME1L* | 5′-guaucgacaucaucaagautt | Ambion/s21077 |
| *human AFG3L2* | 5′-ggaaggacuuugucaauaautt | Ambion/s21518 |
| *human HTRA2* | 5′-gcaccugccguggucuauautt | Ambion/s653 |
| *human PARL* | 5′-ggcaugaaauaaggacuaautt | Ambion/s30890 |
| *human TIM23* | 5′-acaggtggtcttcgagggata | Qiagen/1027417 |
| AllStars Negative Control | non-silencing | Qiagen/1027281 |
| **Chemicals, enzymes and other reagents** | | |
| CCCP | Sigma-Aldrich | c2759 |
| FCCP | MedChemExpress | HY-100410 |
| Oligomycin A | Santa Cruz Biotechnology | sc-201551 |
| Antimycin A | Sigma-Aldrich | A8674 |
| Rotenone | Sigma-Aldrich | R8875 |
| Valinomycin | Thermo Fisher Scientific | Cat# V1644 |
| Cycloheximide | Wako Pure Chemical Industries | Cat# 037-20991 |
| Hydrogen peroxide | Sigma-Aldrich | H1009 |
| Staurosporine | Sigma-Aldrich | S5921 |

| Reagent/resource | Reference or source | Identifier or catalog number |
|---|---|---|
| Actinomycin D | Wako Pure Chemical Industries | Cat# 010-21263 |
| Caspase inhibitor Z-VAD-FMK | MBL Life Science | #4800-520 |
| Native human Calpain 1 protein (Active) | Abcam | ab91019 |
| Calpain Inhibitor I | Nacalai Tesque | Cat# 07036-24 |
| LDH Assay Kit (Cytotoxicity) | Abcam | ab65393 |
| Digitonin | Wako Pure Chemical Industries | Cat# 043-21371 |
| Nonidet(R) P40 Substitute | Nacalai Tesque | Cat# 18558-54 |
| Lipofectamine 2000 | Thermo Fisher Scientific | Cat# 11668019 |
| Lipofectamine RNAiMAX | Thermo Fisher Scientific | Cat# 13778150 |
| Protease inhibitor cocktail | Roche | Cat# 11836153001 |
| PMSF | Nacalai Tesque | Cat# 06297-02 |
| Tris(2-carboxyethyl)phosphine hydrochloride (TCEP) solution | Sigma-Aldrich | 646547 |
| 2-Chloroacetamide (CAA) | Sigma-Aldrich | C0267 |
| Trypsin, Proteomics grade | Sigma-Aldrich | T6567 |
| LysargiNase (Tryp-N) | Merck Millipore | EMS0008 |
| Lysyl endopeptidase (Lys-C) | Wako Pure Chemical Industries | Cat# 129-02541 |
| GL-Tip SDB | GL Sciences Inc. | Cat# 7820-11200 |
| GL-Tip SCX | GL Sciences Inc. | Cat# 7510-11203 |
| Water, Optima LC/MS grade | Thermo Fisher Scientific | Cat# AAB-W6-1 |
| TRIzol | Thermo Fisher Scientific | Cat# 15596026 |
| M-MLV reverse transcriptase | Wako Pure Chemical Industries | Cat# 187-01281 |
| PowerSYBR Green PCR Master Mix | Thermo Fisher Scientific | Cat# 4368708 |
| Q5 Site-Directed Mutagenesis Kit | New England Biolabs | E0554S |
| Q5 High-Fidelity 2X Master Mix | New England Biolabs | M0492 |
| NEBuilder HiFi DNA Assembly Master Mix | New England Biolabs | E2621 |
| Wizard Plus SV Minipreps DNA Purification Systems | Promega | Cat# A1330 |
| DNeasy Blood and Tissue Kit | Qiagen | Cat# 69504 |
| D-MEM, high glucose | Thermo Fisher Scientific | Cat# 11965092 |
| D-MEM, no glucose, no glutamine | Thermo Fisher Scientific | Cat# A14430 |
| Fetal bovine serum (FBS) | Thermo Fisher Scientific | Cat# 10437028 |
| Penicillin-Streptomycin | Thermo Fisher Scientific | Cat# 15140122 |
| GlutaMAX | Thermo Fisher Scientific | Cat# 35050061 |
| D-Galactose | SERVA Electrophoresis GmbH | Cat# 22020.02 |
| Puromycin | Sigma-Aldrich | P8833 |
| Blasticidin | InvivoGen | Cat# ant-bl-05 |
| Hygromycin B | Sigma-Aldrich | H3274 |
| Anti-*c*-Myc Agarose beads | Sigma-Aldrich | A7470 |
| Protein A Sepharose beads | Sigma-Aldrich | P3391 |
| SureBeads Protein G (Magnetic) | Bio-Rad | Cat# 1614023 |
| 16% Formaldehyde (w/v) | Thermo Fisher Scientific | Cat# 28906 |
| 4%-Paraformaldehyde Phosphate Buffer Solution | Nacalai tesque | Cat# 09154-14 |
| Poly-D-Lysin solution | Thermo Fisher Scientific | Cat# A38904 |
| Duolink In Situ Red Starter Kit Mouse/Rabbit | Merck Millipore | Cat# DUO92191 |
| ProLong™ Gold Antifade Mountant | Thermo Fisher Scientific | Cat# P36934 |
| Chemi-Lumi One Super | Nacalai tesque | Cat# 02230 |

| Reagent/resource | Reference or source | Identifier or catalog number |
|---|---|---|
| Precision Plus Protein Dual Color Standards | Bio-Rad | Cat# 1610374 |
| TnT SP6 Quick Coupled Transcription/Translation System | Promega | Cat# L2080 |
| Tetramethylrhodamine, Methyl Ester (TMRM) | Thermo Fisher Scientific | Cat# T668 |
| Seahorse XF Cell Mito Stress Test kit | Agilent Technologies | Cat# 103015-100 |
| Synthetic peptide (mAIFM1 A.A.98 ~ ) $NH_2$-VMGLGLSPEEC-COOH | Eurofins Genomics | This work |
| Synthetic peptide (N-term acetylated) Acetyl-VMGLGLSPEEC-COOH | Eurofins Genomics | This work |
| Superose 6 Increase 10/300 GL | Cytiva | Cat# 29-0915-96 |
| HMW Native Marker Kit | Cytiva | Cat# 17-0445-01 |
| New Hematoxylin type C | Muto Pure Chemicals | Cat# 30152 |
| New Eosin type M | Muto Pure Chemicals | Cat# 32081 |
| *Murine MIC60 shRNA* lentiviral particles | Santa Cruz Biotechnology | sc-75792-V |
| Lactate Assay Kit-WST | DOJINDO Laboratories | Cat# 343-09281 |
| **Software** | | |
| Proteome Discoverer version 2.4 | Thermo Fisher Scientific | |
| DIA-NN | https://github.com/vdemichev/DiaNN | Demichev et al, 2020 |
| GraphPad Prism 8J | https://www.graphpad.com | GraphPad Software |
| ImageJ | https://imagej.nih.gov/ij/index.html | Schneider et al, 2012 |
| Instant Clue | http://www.instantclue.uni-koeln.de | Nolte et al, 2018 |
| DAVID Functional Annotation | https://davidbioinformatics.nih.gov/home.jsp | Huang et al, 2009 |
| **Other** | | |
| Nikon C2+ confocal microscope | Nikon Instruments Inc. | |
| WSE-6200H LuminoGraphII | ATTO Corporation | |
| Mica WideFocal | Leica Microsystems | |
| Countess II automated cell counter | Thermo Fisher Scientific | |
| Typhoon FLA 9500 biomolecular imager | Cytiva | |
| Seahorse XFe96 Analyzer | Agilent Technologies | |
| 3730xl DNA Analyzer | Applied Biosystems Inc. | |
| AKTA pure 25 | Cytiva | |
| Multimode Plate Reader EnVision 2105 | PerkinElmer | |
| QuantStudio 5 Real-Time PCR System | Thermo Fisher Scientific | |
| Orbitrap Fusion mass spectrometer | Thermo Fisher Scientific | |
| timsTOF HT mass spectrometer | Bruker | |
| **Deposited data** | | |
| LFQ of anti-Myc IP-MS of OMA1-Myc-expressing MEFs | This work (related to Dataset EV1) | ProteomeXchange: PXD063014 |
| LFQ of neo-amino terminal peptides using Tryp-N | This work (related to Dataset EV2) | ProteomeXchange: PXD063017 |
| LFQ of anti-Myc IP-MS of AIFM1[TCS] or AIFM1[TCS/TEV]-expressing cells | This work (related to Dataset EV3) | ProteomeXchange: PXD063094 |
| Mitochondrial proteome of AIFM1[TCS] or AIFM1[TCS/TEV]-expressing cells | This work (related to Dataset EV4) | ProteomeXchange: PXD063602 |
| Gene expression profile of AIFM1[TCS] or AIFM1[TCS/TEV]-expressing cells | This work (related to Dataset EV5) | GEO: GSE296062 |
| DIA-PASEF analysis of lung tissues from influenza-infected and uninfected mice | This work (related to Dataset EV6) | ProteomeXchange: PXD069450 |

Eagle medium (D-MEM, high glucose) supplemented with 1% GlutaMAX, penicillin (100 U/mL)-streptomycin (100 µg/mL), and 10% fetal bovine serum (FBS; Thermo Fisher Scientific) at 37 °C under 5% $CO_2$ conditions. Mitochondrial stress was induced by treating cells with CCCP (20 or 40 µM), FCCP (20 µM), valinomycin (1 µg/mL), oligomycin A (1 µM), antimycin A (1 µM), $H_2O_2$ (1 mM), or rotenone (5 µM) as indicated. In some experiments, CHX (10 µg/mL) was added to the culture medium to inhibit protein synthesis. Cells were regularly seeded at equal densities before an experiment and checked for mycoplasma contamination.

## Plasmid construction and mutagenesis

Total mRNAs from HEK293 and MEFs were isolated using TRIzol reagent (Thermo Fisher Scientific) and reverse transcribed using M-MLV reverse transcriptase (Wako Pure Chemical Industries). Polymerase chain reaction assays were performed using Q5 High-Fidelity DNA polymerase (New England Biolabs, Ipswich, MA). The following primers (see Reagents and Tools Table for sequences) were used to generate the complete open reading frames of human OMA1: TK921/TK890; murine OMA1: TK950/TK889; mYME1L: TK956/TK957; mDELE1: TK1378/TK1379; mAIFM1: TK1038/TK1039. Plasmids encoding epitope-tagged proteins were constructed by ligation of each cDNA into either NotI or NheI at the 5' end and KpnI at the 3' end of a digested pcDNA5/FRT vector (Thermo Fisher Scientific) encoding either a C-terminal 3× Myc or 3× hemagglutinin (HA) tag. To generate a plasmid encoding TEV protease localized to the mitochondrial IMS (IMS-TEV), a DNA fragment encoding the IMS targeting sequence of rat AIFM1 (A.A.1-95) was cloned in front of the sequence encoding the TEV protease. The substitution was introduced into each plasmid by site-directed mutagenesis according to the manufacturer's protocol (New England Biolabs). To generate retroviral expression constructs, each cDNA was subcloned into the retroviral vector pMXs-puro (Cell Biolabs). All constructs used in the study were confirmed by DNA sequencing (Applied Biosystems 3730xl Genetic Analyzer, Waltham, MA).

## Generation of stable expression cell lines

Flp-In T-REx-293 AIFM1 knockout HEK293 cells (Salscheider et al, 2022) were cotransfected with either pcDNA5/FRT-AIFM1$^{WT}$ or AIFM1 variants, along with pOG44 Flp recombinase expression vector (Thermo Fisher Scientific). After transfection, cells were selected with 100 µg/mL hygromycin B for 7−10 days, single colonies were selected by plating, and expression of the gene of interest at steady state levels was checked by western blotting. Viral gene delivery using a retrovirus system was also used to generate stable expression cell lines. Briefly, each constructed pMXs-puro plasmid (see Reagents and Tools Table) was transfected into the platinum packaging cell lines using Lipofectamine 2000 reagent (Thermo Fisher Scientific). The retroviral supernatants were harvested 48 h post-transfection and used to infect MEF or other cell lines. After retroviral infection, cells were selected with 1.5 µg/mL puromycin for 1 week, and single colonies were selected based on expression confirming by immunoblotting. All antibodies used in the study are listed in the Reagents and Tools Table.

## Protein expression in Escherichia coli

Recombinant human OMA1 was expressed in E. coli BL21(DE3) cells (New England Biolabs). The inoculated culture was grown to log phase at 37 °C, and overproduction of the recombinant protein was induced by the addition of isopropyl β-D-1-thiogalactopyranoside (Nacalai Tesque, Kyoto, Japan) to a final concentration of 1 mM. After 3 h of induction, cells were harvested by centrifugation (6000 × g for 15 min), and the pellets were stored frozen (−20 °C) until used for the western blotting.

## Mitochondrial isolation and proteolysis

Mitochondrial isolation, proteolysis, and membrane association assays were performed as previously described (Yoshizumi et al, 2014) with slight modifications. Briefly, cultured cells were washed once with 1× phosphate-buffered saline (PBS, pH 7.4), scraped from the culture plate, and lysed in ice-cold homogenization buffer containing 20 mM HEPES (pH 7.5), 70 mM sucrose, and 220 mM mannitol by 30 strokes in a Dounce homogenizer on ice. The homogenate was then centrifuged at 800 × g for 5 min (4 °C) to precipitate the nuclei, and the resulting supernatant was further centrifuged at 10,000 × g for 10 min (4 °C) to precipitate the crude mitochondrial fraction.

For the proteinase K resistance assay, the isolated mitochondrial pellet was resuspended in either the homogenization buffer or a hypotonic buffer (20 mM HEPES, pH 7.5) and kept on ice for 30 min to induce swelling. Both samples were then treated with proteinase K (50 µg/mL) for 15 min on ice, and the reactants were subjected to western blot analysis with the indicated antibodies. For the membrane association assay, mitochondrial pellets with or without swelling treatment were washed once with the homogenization buffer and centrifuged at 15,000 × g for 15 min (4 °C) to separate supernatant or precipitate fractions. The supernatant was subjected to TCA precipitation, and the precipitated fraction was resuspended in a sodium dodecyl sulfate (SDS) sample buffer (50 mM Tris-HCl [pH 6.8], 2% SDS, 0.1% [w/v] bromophenol blue, 10% [v/v] glycerol, and 5% β-mercaptoethanol) and analyzed by western blotting.

## Generation of the N-terminal specific antibody against AIFM1

To generate a rabbit polyclonal antibody against the N-terminal region of the murine AIFM1 cleavage site (A.A. 98 ~ ), antiserum raised against the synthetic peptide NH$_2$-$^{98}$VMGLGLSPEEC-COOH (Eurofins Genomics, Louisville, KY) was affinity-purified using the same synthetic peptide column. The purified fraction was then passed through a column with the N-terminally acetylated peptide sequence (CH$_3$CO-NH$_2$-$^{98}$VMGLGLSPEEC) to remove immunoglobulin G (IgG), which recognizes peptides mimicking the uncleaved protein. Finally, the flow-through fraction was affinity-purified on Protein A Sepharose (Sigma-Aldrich).

## Western blotting

Immunoblotting was performed as previously described (Koshiba and Chan, 2003) with minor modifications. Briefly, cells were collected and washed with 1× PBS (pH 7.4) before lysis with ice-cold lysis buffer (50 mM Tris-HCl buffer [pH 7.4] containing 150 mM NaCl, 1 mM

EDTA, 1% [w/v] NP-40, 10% [w/v] glycerol, and protease inhibitor cocktail; Roche, Basel, Switzerland). The extracts were then mixed with the SDS sample buffer and samples were separated by SDS-PAGE. Precision Plus Protein Dual Color Standards (Bio-Rad, Hercules, CA) were used to estimate the molecular weight of each protein of interest on SDS gels. Immunoblotting of the gels was performed on Immobilon-P polyvinylidene difluoride (PVDF) membrane (Merck Millipore, Burlington, MA) for 60 min at 25 V using transfer buffer (23 mM Tris, 180 mM glycine, 19% methanol). Immunoblotted membranes were then blocked in 5% nonfat dry milk in Tris-buffered saline (TBS) containing 0.1% Tween-20, followed by incubation with the indicated primary antibodies, and proteins were detected with horseradish peroxidase-conjugated secondary antibodies using a WSE-6200H LuminoGraphII Image Analyzer (ATTO Corporation, Tokyo, Japan). All antibodies used in the study are listed in the Reagents and Tools Table.

## RNA interference

For RNA interference knockdown experiments, siRNAs listed in the Reagents and Tools Table were used. HeLa cells were each transfected twice with 10 nM siRNA (final concentration) at 48 h intervals using Lipofectamine RNAiMAX reagent (Thermo Fisher Scientific) according to the manufacturer's protocols. At 96 h after the first treatment, the siRNA-treated cells were used for some functional assays. The AllStars Negative Control siRNA (Qiagen, Hilden, Germany) was used as a control.

## Immunofluorescence

Cells were plated on coverslips in 12-well plates ($5 \times 10^4$ cells/well). For monitoring the Flp-In T-REx-293 cell line, the coverslips were pre-coated with 0.05 mg/mL poly-D-lysin solution (Thermo Fisher Scientific) before seeding the cells. The next day, the cells were fixed with 4% paraformaldehyde phosphate buffer solution (Nacalai Tesque) for 10 min at 37 °C, permeabilized with 0.2% Triton X-100 in 1× PBS (pH 7.4), and blocked with 5% FBS. Epitope-tagged proteins (HA- or Myc-) were detected with their specific primary and AlexaFluor488 secondary antibodies (Invitrogen, Waltham, MA), and mitochondria were stained with either anti-COX IV (3E11) or mtHSP70 (JG1) primary antibodies followed by the AlexaFluor568 or Cy3-conjugated secondary antibodies, respectively. Cells were imaged using a C2+ confocal (Nikon Instruments Inc., Melville, NY) or Mica WideFocal (Leica Microsystems, Wetzlar, Germany) microscope.

## Immunoprecipitation

For the immunoprecipitation experiment, ~95% confluence cells stably expressing Myc-tagged AIFM1 constructs were washed once with 1× PBS (pH 7.4), lysed with lysis buffer containing 50 mM Tris-HCl (pH 7.4), 150 mM NaCl, 10% (w/v) glycerol, 1 mM EDTA, and 0.5% (w/v) digitonin, protease inhibitor cocktail, and the clarified supernatants were incubated with 1 µg of the anti-c-Myc monoclonal antibody 9E10 (Covance), followed by incubation overnight at 4 °C with 20 µL Protein A-Sepharose beads (Sigma-Aldrich). The next day, the beads were washed four times with 1× PBS (pH 7.4), the precipitated proteins were eluted from the beads by adding SDS sample buffer and heating to 96 °C for 3 min, and the immunoprecipitates were resolved by SDS-PAGE and

immunoblotted with the indicated antibodies. For immunoprecipitation of the OXPHOS complex, samples were heated to 50 °C for 10 min to avoid protein degradation.

## Clear native-PAGE analysis

CN-PAGE was performed as previously described (Ban et al, 2025) with slight modifications. Briefly, isolated mitochondria from each cell were solubilized in 20 mM Bis-Tris (pH 7.0) buffer containing 2 mM NaCl, 500 mM aminocaproic acid, 10% (w/v) glycerol, 1 mM EDTA, 0.5% (w/v) digitonin, and protease inhibitor cocktail for 15 min on ice. The clarified supernatant was then supplemented with 0.5% (w/v) CBB-G250 and separated on a 3%-12% gradient gel. Anode buffer (50 mM Bis-Tris, pH 7.0) and cathode buffer (50 mM Tricine, 15 mM Bis-Tris [pH 7.0], 0.05% sodium deoxycholate, and 0.02% n-dodecyl-D-maltoside) were used for electrophoresis. After electrophoresis, the separated proteins in the gel were denatured at 60 °C for 15 min in Tris-HCl (pH 6.8) buffer containing 1% SDS and 2% β-mercaptoethanol, transferred to a PVDF membrane, followed by western blotting analysis.

## Size exclusion chromatography

Size exclusion chromatography of each mitochondrial extract from cultured cells was performed on a Superose 6 Increase 10/300 GL column (Cytiva), as previously described (Koshiba et al, 2011), with slight modifications. Briefly, mitochondria extracted from the cells were solubilized in lysis buffer (50 mM Tris-HCl [pH 8.0], 150 mM NaCl, and 1% digitonin) and centrifuged at $15,000 \times g$ for 15 min at 4 °C to obtain the soluble fraction. The supernatant of mitochondrial extracts was loaded onto a column equilibrated with 50 mM Tris-HCl (pH 8.0) containing 150 mM NaCl, and 0.01% digitonin. Chromatography was performed at a flow rate of 0.5 mL/min at 4 °C. The eluted fractions were resolved using 7%, 12%, or 15% SDS-PAGE, followed by western blot analysis using antibodies against the indicated proteins. The molecular weight protein standards used in the study were thyroglobulin (669 kDa), ferritin (440 kDa), aldolase (158 kDa), and ovalbumin (44 kDa) (HMW Native Marker Kit, Cytiva).

## LC-MS/MS analysis (IP-MS & Neo-amino terminal MS)

### IP-MS

Control parental OMA1 KO MEFs and rescued cells stably expressing WT OMA1-Myc (cultured in 10-cm dishes) were incubated with 0.2% (w/v) formaldehyde (Thermo Fisher Scientific) for 15 min at 37 °C, followed by quenching with 100 mM glycine-NaOH (pH 7.5) for 10 min at room temperature. After washing once with 1× PBS (pH 7.4), the cells were scraped, and mitochondria were isolated as described above. The mitochondrial fractions were then lysed in 1 mL of lysis buffer containing 50 mM Tris-HCl (pH 7.4), 150 mM NaCl, 10% (w/v) glycerol, 1 mM EDTA, 0.5% (w/v) digitonin, and protease inhibitor cocktail. After centrifugation, the clarified supernatants were incubated with an antibody against Myc (My3; MBL Life Science, Tokyo, Japan) for 2 h at 4 °C. The reactants were then incubated with magnetic SureBeads Protein G (Bio-Rad) overnight at 4 °C, and the next day the beads were washed three times with 1× PBS (pH 7.4) and twice with 50 mM ammonium bicarbonate. Proteins on the beads were

digested by adding 200 ng trypsin/Lys-C mix (Promega) for 16 h at 37 °C. The digests were reduced, alkylated, acidified, and desalted using GL-Tip SDB (GL Sciences Inc., Tokyo, Japan), and the eluates were evaporated and dissolved in 0.1% trifluoroacetic acid (TFA) and 3% acetonitrile.

LC-MS/MS analysis of the resulting peptides was performed on an EASY-nLC 1200 UHPLC connected to an Orbitrap Fusion mass spectrometer through a nanoelectrospray ion source (Thermo Fisher Scientific). Peptides were separated on a 75-μm inner diameter × 150 mm C18 reverse phase column (Nikkyo Technos, Tokyo, Japan) with a linear gradient of 4%-32% acetonitrile for 0−100 min followed by an increase to 80% acetonitrile for 100−110 min. The mass spectrometer was operated in a data-dependent acquisition mode with a maximum duty cycle of 3. MS1 spectra were measured with a resolution of 120,000, an automatic gain control (AGC) target of 4e5, and a mass range of 375 to 1500 $m/z$. Higher-energy collisional dissociation-MS/MS spectra were acquired in the linear ion trap with an AGC target of 1e4, an isolation window of 1.6 $m/z$, a maximum injection time of 35 ms, and a normalized collision energy of 30. Dynamic exclusion was set to 20 s. Raw data were analyzed directly against the Swiss-Prot database restricted to *Mus musculus* using Proteome Discoverer version 2.4 (Thermo Fisher Scientific) for identification and label-free precursor ion quantification. Search parameters were as follows: (a) trypsin as an enzyme with up to two missed cleavages; (b) precursor mass tolerance of 10 ppm; (c) fragment mass tolerance of 0.6 Da; and (d) cysteine carbamidomethylation as a fixed modification; and (e) protein N-terminal acetylation and methionine oxidation as variable modifications. Peptides were filtered with a false discovery rate of 1% using the Percolator node. Normalization was performed so that the total sum of abundance values for each sample was equal across all peptides.

### Neo-amino-terminal MS

To determine candidate OMA1 substrates during mitochondrial stress, cells were treated with or without 40 μM FCCP for 2 h, then washed once with 1× PBS (pH 7.4), and cell pellets were collected. The pellets were then lysed in a guanidine buffer (6 M guanidine-HCl, 100 mM HEPES-NaOH [pH 7.5], 10 mM Tris-(2-carboxyethyl) phosphine [TCEP, Sigma-Aldrich], and 40 mM 2-chloroacetamide [CAA, Sigma-Aldrich]). After heating and sonication, 30 μg of protein was purified by methanol-chloroform precipitation and resuspended in 20 μL of PTS buffer (100 mM Tris-HCl [pH 8.0], 12 mM sodium deoxycholate, and 12 mM sodium lauroylsarcosinate). The protein solution was diluted 10-fold with 10 mM CaCl$_2$ and digested with 600 ng of Tryp-N (LysargiNase, Merck Millipore) overnight at 37 °C. After acidification with 0.5% TFA (final conc.), an equal volume of ethyl acetate was added to each sample, followed by centrifugation at 15,700 × $g$ for 2 min to separate the ethyl acetate layer. The aqueous layer was collected and desalted using GL-Tip SDB, and the eluates were evaporated and dissolved in 50 μL of 2.5% formic acid and 30% acetonitrile. Enrichment of protein N-terminal peptides was performed using GL-Tip SCX (GL Sciences Inc.) based on a previous report (Chang et al, 2021). Flow-through fractions were evaporated and dissolved in 0.1% TFA and 3% acetonitrile. LC-MS/MS analysis of the resulting peptides was performed as described above. Raw data were analyzed directly against the Swiss-Prot database restricted to *Mus musculus* using Proteome Discoverer version 2.4 for identification and label-free precursor ion quantification. Search parameters were as

follows: (a) Tryp-N as a semi-specific enzyme with up to two missed cleavages; (b) precursor mass tolerance of 10 ppm; (c) fragment mass tolerance of 0.6 Da; (d) cysteine carbamidomethylation as a fixed modification; and (e) protein N-terminal acetylation and methionine oxidation as variable modifications.

### Activities of respiratory chain complexes

Mitochondrial fractions from each Flp-In T-REx-293 cell line were solubilized in 20 mM Bis-Tris (pH 7.0) buffer containing 2 mM NaCl, 500 mM aminocaproic acid, 10% (w/v) glycerol, 1 mM EDTA, 0.5% (w/v) digitonin, and protease inhibitor cocktail for 15 min on ice, and the extracts were centrifuged at 15,000× $g$ for 10 min at 4 °C. The clarified supernatant was then applied to CN-PAGE, followed by evaluation of in-gel activities by incubating the gel in either Complex I (NADH dehydrogenase) activity substrate (5 mM Tris-HCl [pH 7.4], 0.2 mM NADH, and 0.25% [w/v] nitro blue tetrazolium [NBT] chloride) or Complex II (succinate dehydrogenase) activity substrate (5 mM Tris-HCl [pH 7.4], 20 mM succinate, 0.2 mM phenazine methosulfate, and 0.25% [w/ v] NBT). The activities of Complexes I and II were quantified using ImageJ software (Schneider et al, 2012).

### Metabolite measurement

For metabolite extraction, $4 \times 10^6$ cells were treated with methanol containing internal standards (H3304-1002, Human Metabolome Technologies, Inc. [HMT], Tsuruoka, Japan), including AMP, ADP, and ATP. The extract was then centrifuged at 2300 × $g$ for 5 min at 4 °C. The resulting supernatant was filtered using a 5-kDa cutoff centrifugal filter at 9100 × $g$ and 4 °C to remove macromolecules. The filtrate was then dried and reconstituted in Milli-Q water for metabolomic analysis. Anion analysis was performed by HMT using capillary electrophoresis- (CE-)MS/MS, as previously described (Ohashi et al, 2008; Ooga et al, 2011). CE-MS/ MS analysis was performed using an Agilent CE capillary electrophoresis system equipped with an Agilent 6460 Triple Quadrupole LC/MS (Agilent Technologies, Inc., CA). Peak areas of individual metabolites were calculated using MasterHands software (Keio University, Tsuruoka, Japan) (Sugimoto et al, 2010) and MassHunter Quantitative Analysis (Agilent). Peak areas of AMP, ADP, and ATP were normalized to their respective internal standards, and concentrations were determined using standard curves.

### Cell viability assay

Cell viability was monitored by measuring lactate dehydrogenase (LDH) activity in the culture supernatant using an LDH-cytotoxicity Assay Kit (Abcam, Cambridge, UK) according to the manufacturer's protocol.

### Measurement of cellular respiration

The oxygen consumption rate was measured in a Seahorse Extracellular Flux Analyzer XFe96 (Agilent) in cells grown in D-MEM-GlutaMAX containing 25 mM glucose medium. In each well, $5 \times 10^4$ cells were plated and incubated at 37 °C for 1 h prior to measurement. ATP production was assessed after the addition of

oligomycin (2 µM), maximal respiration after FCCP (0.5 µM), and non-mitochondrial respiration after the addition of rotenone and antimycin A (each 0.5 µM).

## Measurement of ΔΨ$_m$

To measure the ΔΨ$_m$, $6 \times 10^4$ cells were seeded on 96-well plates at 37 °C for overnight. The next day, the medium was removed and replaced with fresh medium without (DMSO) or with 20 µM FCCP and incubated for another 1 h. The medium was then replaced with a medium containing 200 nM TMRM (Thermo Fisher Scientific) and incubated for 30 min. Cells were washed once with ice-cold 1× PBS (pH 7.4) and replaced with fresh PBS buffer containing 2% FBS. Fluorescence was measured using an Envision 2105 multi-mode plate reader (PerkinElmer) at excitation/emission wavelengths of 544 nm/579 nm. Each signal intensity was normalized to the average intensity of DMSO-treated WT cells.

## Quantification of mtDNA

Total genomic DNA from cultured cells was extracted and purified using the DNeasy Blood and Tissue Kit (Qiagen) and diluted to a concentration of 25 ng/µL. To quantify the amount of mitochondrial DNA (mtDNA) per nuclear DNA (nDNA), quantitative real-time polymerase chain reaction (qPCR) was performed using PowerSYBR Green PCR Master Mix (Thermo Fisher Scientific) and QuantStudio 5 Real-Time PCR System (Thermo Fisher Scientific). Quantification of relative copy number differences was performed based on the difference in the threshold amplification between mtDNA and nDNA [ΔΔC(t) method]. The following primer sets were used for mtDNA: forward (5'-tcgaaaccgcaaacatatca) and reverse (5'-caggcgtttaatggggttta) targeting human ND5; and for nDNA: forward (5'-ctgtggcatccac-gaaacta) and reverse (5'-agtacttgcgctcaggagga) targeting human ACTB.

## Whole cell proteomics

### Protein digestion

Protein (15 µg) were subjected to tryptic digestion and the samples were reduced (10 mM TCEP) with alkylated (20 mM CAA) in the dark for 45 min at 45 °C. Samples were then subjected to an SP3-based digestion (Hughes et al, 2014). Washed the SP3 beads (Sera-Mag magnetic carboxylate modified particles, Sigma-Aldrich) were mixed equally, and 3 µL of bead slurry was added to each sample. Acetonitrile was added to a final concentration of 50% and washed twice using 70% ethanol (200 µL) on an in-house manufactured magnet. After an additional acetonitrile wash (200 µL), 5 µL digestion solution (10 mM HEPES [pH 8.5] containing 0.5 µg Trypsin [Sigma-Aldrich] and 0.5 µg Lys-C [Wako Pure Chemical Industries]) was added to each sample and incubated overnight at 37 °C. Peptides were desalted on a magnet using $2 \times 200$ µL acetonitrile. Peptides were eluted in 10 µL 5% DMSO in LC-MS water (Sigma-Aldrich) in an ultrasonic bath for 10 min. Formic acid and acetonitrile were added to a final concentration of 2.5% and 2%, respectively. Samples were stored at −20 °C before being subjected to LC-MS/MS analysis.

### Liquid chromatography and mass spectrometry

LC-MS/MS instrumentation consisted of an Easy-nLC 1200 (Thermo Fisher Scientific) coupled via a nano-electrospray

ionization source to an Exploris 480 mass spectrometer (Thermo Fisher Scientific). For peptide separation, an Aurora Frontier column (60 cm × 75 µm C18 UHPLC, IonOpticks, Victoria, Australia) was used. A binary buffer system (A: 0.1% formic acid and B: 0.1% formic acid in 80% acetonitrile) was used with a gradient at a flow rate of 185 nL/min: buffer B was linearly increased from 4% to 28% over 100 min, followed by an increase to 40% over the next 10 min. The buffer B concentration was further increased to 50% over 4 min, then to 65% over the next 3 min. The column was then washed with 95% buffer B for an additional 3 min. The RF Lens amplitude was set to 45%, the capillary temperature was 275 °C, and the polarity was set to positive. MS1 profile spectra were acquired at a resolution of 120,000 (at 200 $m/z$) over a mass range of 450−850 $m/z$ and an AGC target of $1 \times 10^6$.

For MS/MS independent spectra acquisition, 34 equally spaced windows were acquired at an isolation $m/z$ range of 7 Th, and the isolation windows overlapped by 1 Th. The fixed first mass was 200 $m/z$. The isolation center range covered a mass range of 500–740 $m/z$. Fragmentation spectra were acquired at a resolution of 30,000 at 200 $m/z$ using a maximal injection time setting of 'auto' and stepped normalized collision energies (NCE) of 24, 28, and 30. The default charge state was set to 3. The AGC target was set to 3e6 (900% - Exploris 480). MS2 spectra were acquired in centroid mode. FAIMS was enabled using an inner electrode temperature of 100 °C and an outer electrode temperature of 90 °C. The compensation voltage was set to −45 V.

### Data analysis

Data were analyzed in direct data-independent acquisition mode using Spectronaut (version 18.5.231110.55695, Bruderer et al, 2015) with the Uniprot *Homo sapiens* (Human) reference proteome (UP000005640; reviewed only, 20,400 protein sequences). Trypsin/P was selected as the cleavage enzyme with a specific digestion type. The minimum peptide length was set to seven amino acids, allowing up to two missed cleavages. The false discovery rate (FDR) for peptide spectrum matches, peptides, and protein groups were controlled at 0.01. Mass tolerances were applied using the default dynamic settings (Dynamic, 1). The directDIA + (Deep) workflow was selected, and cross-run normalization was enabled. The protein group file was exported, and LFQ intensities (calculated using the MaxLFQ algorithm; Cox et al, 2014) were log2 transformed. Statistically significant differences in protein abundance were determined using a two-sided $t$ test followed with permutation-based FDR correction ($s_0 = 0.1$, number permutations=500, FDR < 0.05) in Instant Clue (Nolte et al, 2018).

## Cell growth assay

For the cell growth assay, we used our customized oxidative medium (glucose-free D-MEM [Thermo Fisher Scientific] supplemented with 10% FBS, 2% GlutaMAX, 1% penicillin [100 U/mL]-streptomycin [100 µg/mL], and 10 mM galactose as a carbon source) to cultivate cell cultures. The day before the assay, $4 \times 10^5$ cells were seeded on 6-well dishes and incubated overnight at 37 °C. Every 2 days (total of 8 days), cells were counted in triplicate using the Countess II automated cell counter (Thermo Fisher Scientific).

## In organello import assay

The In organello import assay was performed as previously described with minor modifications (Salscheider et al, 2022). Radiolabeled precursor proteins were synthesized by a mammalian cell-free protein synthesis system using the rabbit reticulocyte lysate-based TNT SP6 Quick Coupled Transcription/Translation System (Promega) containing [$^{35}$S]-methionine according to the manufacturer's protocol. Mitochondria (40 μg) isolated from cells as described above were subjected to each import reaction. The protein import assay was initiated by mixing the radiolabeled precursor protein with the mitochondrial fraction at 30 °C in the presence or absence of 40 μM FCCP/10 μM oligomycin A. The import reaction was stopped after 8 or 15 min (10 or 20 min) by placing the reactants on ice. The samples were then treated with proteinase K (10 μg/mL) for 15 min to degrade non-imported precursor protein. After protease inactivation with 1 mM phenyl-methyl sulfonyl fluoride (PMSF), mitochondria were washed twice with homogenization buffer and analyzed by SDS-PAGE. The radiolabeled proteins were detected by autoradiography using Typhoon FLA 9500 (Cytiva, Marlborough, MA).

## RNA sequencing

RNA samples used for sequencing analysis were quantified using an ND-1000 spectrophotometer (NanoDrop Technologies, Wilmington, DE) and their quality was confirmed using a Tapestation (Agilent). Sequencing libraries were prepared from 200 ng total RNA using the MGIEasy rRNA Depletion Kit and the MGIEasy RNA Directional Library Prep Set (MGI Tech Co., Ltd.) according to the manufacturer's protocol. The libraries were sequenced on the DNBSEQ-G400 FAST Sequencer (MGI Tech Co., Ltd.) using the 150-nt paired-end strategy. To analyze the sequencing data, all sequencing reads were trimmed for low-quality bases and adapters using Trimmomatic (v.0.38) (Bolger et al, 2014). Raw counts for each gene were estimated in each sample using RSEM version 1.3.0 (Li and Dewey, 2011) and Bowtie 2 (Langmead and Salzberg, 2012). We used the edgeR program (Robinson et al, 2010) to detect differentially expressed genes (DEGs). Normalized counts per million (CPM) values and fold-changes (logFC) between samples were obtained from the raw gene-level counts. We defined differentially expressed genes as those exhibiting a fold-change of ≥2, either upregulated or downregulated.

## Viral infection

### Mice

Six-week-old female C57BL/6JJmsSlc mice were purchased from Japan SLC, Inc. (Shizuoka, Japan). All mouse experiments were conducted in accordance with the University of Tokyo's Regulations for Animal Care and Use and approved by the Animal Experiment Committee of the Institute of Medical Science, the University of Tokyo (PA22-33).

### Influenza virus infection in vivo

A mouse-adapted influenza A virus (IAV) strain, A/Puerto Rico/8/1934 (PR8), was grown in the allantoic cavities of 10-day-old fertile chicken eggs for 2 days at 35 °C. Viral titers were quantified using a standard plaque assay with MDCK cells, and the viral stock was stored at -80 °C. For the IAV/PR8 infection, the mice were infected via intranasal administration of a 50-μL suspension of the virus (1000 pfu of influenza virus in PBS) under isoflurane anesthesia (Nagai et al, 2023). For histopathologic analysis, dissected lung tissues were fixed in 10% formalin and paraffin-embedded. Tissues were sectioned at 5-μm-thick paraffin and stained with hematoxylin and eosin (Muto Pure Chemicals Co., Ltd., Tokyo, Japan). The lactate level in mouse lung tissue homogenates was measured using the Lactate Assay Kit-WST (DOJINDO Laboratories, Kumamoto, Japan), following the manufacturer's instructions.

For electron microscopy analysis, the dissected lung tissues from uninfected or infected-mice were fixed overnight at 4 °C in 100 mM cacodylate buffer (pH 7.4) containing 2% paraformaldehyde and 2% glutaraldehyde. The tissues were then washed three times for 30 min each in 100 mM cacodylate buffer and postfixed for 2 h at 4 °C in 2% osmium tetroxide (OsO$_4$) in cacodylate buffer. The samples were dehydrated in a graded ethanol series (50%, 70%, 90%, and 100%), embedded in Quetol 812 resin (Nisshin EM Co., Japan), and polymerized for 48 h at 60 °C. The polymerized resins were cut into ultra-thin sections at 70 nm with a LEICA UTC ultramicrotome (Ultracut UCT; Leica, Vienna, Austria) and mounted on copper grids. The grids were then stained with 2% uranyl acetate and lead solution (Sigma-Aldrich). Images were collected using a JEM-1400Plus transmission electron microscope (JEOL Ltd., Tokyo, Japan) operating at 100 kV and equipped with an EM-14830RUBY2 CCD camera (EM-14830RUBY2, JEOL Ltd.).

### Whole proteomics in lung tissue

Lung tissues from uninfected and infected mice ($n = 5$ per group) were homogenized in 0.5 mL of guanidine buffer. After heating and sonication, the lysates were centrifuged at $20,000 \times g$ for 15 min at 4 °C. The resulting supernatants were collected, and proteins (20 μg each) were purified using the SP3 method (Hughes et al, 2019) and digested with 200 ng trypsin/Lys-C mixture at 37 °C overnight. The digests were acidified, centrifuged, and desalted using a GL-Tip SDB. The eluates were evaporated to dryness and reconstituted in 0.1% TFA and 3% ACN. LC-MS/MS analysis of the resulting peptides was performed using a timsTOF HT system (equipped with a CaptiveSpray 2 ion source) coupled to a nanoElute 2 (Bruker, Billerica, MA). A 150-mm C18 reversed-phase column with an inner diameter of 75 μm was used. Mobile phase A consisted of ultrapure water containing 0.1% formic acid, and mobile phase B consisted of ACN containing 0.1% formic acid. The gradient was initiated at 5% solvent B, increased to 20% at 40 min, and increased further to 35% at 60 min, followed by a rapid increase to 95% at 61 min, which was maintained until 75 min. Data were acquired in DIA-PASEF mode with an MS1 $m/z$ range of 100–1700 and an ion mobility ($1/K_0$) range of 0.6–1.6. The MS2 polygon was manually defined based on the region where peptides were predominantly detected, using the following four vertices: Point 1 ($m/z = 300$, $1/K_0 = 0.64$), Point 2 ($m/z = 300$, $1/K_0 = 0.8$), Point 3 ($m/z = 770$, $1/K_0 = 0.97$), and Point 4 ($m/z = 770$, $1/K_0 = 1.15$). An 8-Da isolation window with a 0.5-Da overlap was applied within this polygon, resulting in 62 windows per cycle and a total cycle time of 3.36 s. DIA-MS data were searched using DIA-NN (version 1.9) against a mouse in silico spectral library. The library was constructed from the UniProt mouse protein sequence database using the following parameters: trypsin as the digestion enzyme, one missed cleavage allowed, peptide lengths ranging of

7−45 amino acids, precursor charges of 2−4, and a fragment ion $m/z$ range of 200–1800. The following features were enabled: library-free search, deep learning–based retention time and ion mobility predictions, N-terminal methionine cleavage, and cysteine carbamidomethylation. MS1 and MS2 mass accuracies were automatically optimized; both neural network classifiers were run in single-pass mode, and quantification was performed using the high-precision QuantUMS strategy.

### Viral infection in vitro

*OMA1* KO MEFs that stably expressing either WT or E324Q OMA1-Myc in 24-well plates ($1.5 \times 10^5$ cells) were either infected with 120 µL of the Sendai virus (4 or 10 hemagglutinin [HA] units/mL) or encephalomyocarditis virus (at a multiplicity of infection of 0.1, 0.5, or 1) for 1 h at 37 °C. Then, 400 µL of D-MEM medium was added, and the cells were incubated for an additional 15 h before performing western blots.

## Quantification and statistical analysis

Statistical analysis was performed using GraphPad Prism 8.4. We considered different cell populations to be biologic replicates; aliquots or repeated measurements of a cell population were considered technical replicates. We obtained at least three independent reproducible results for most key experiments, although we did not perform explicit power calculations. Data are presented as the mean ± standard deviation (SD), and statistical significance between two groups was assessed using an unpaired, two-tailed Student's $t$ test. Comparisons among three or more groups were performed using one-way analysis of variance (ANOVA) followed by the Tukey post hoc multiple comparisons test. For analyses involving two independent variables, two-way ANOVA was conducted, followed by the appropriate post hoc multiple comparisons test as indicated in the figure legends. A $p$ value of less than 0.05 was considered statistically significant.

## Data availability

The IP-MS, neo-amino-terminal peptides, and whole proteomics in lung tissue data have been deposited to the ProteomeXchange Consortium via the jPOST partner repository with the dataset identifiers PXD063014, PXD063017, PXD063094, and PXD069450. The mitochondrial proteome data were deposited to the PRoteomics IDEntifications (PRIDE) database with the dataset identifier PXD063602. The RNA sequencing data were deposited to the Gene Expression Omnibus (GEO) with the dataset identifier GSE296062.

The source data of this paper are collected in the following database record: biostudies:S-SCDT-10_1038-S44318-026-00734-y.

## Peer review information

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

## Acknowledgements

We thank David Chan (Caltech) and Naotada Ishihara (Osaka University, Japan) for generously providing the $OPA1^{-/-}$ and $Drp-1^{-/-}$ MEFs, respectively. Carlos López-Otín (University of Oviedo) kindly provided the $OMA1^{-/-}$ MEFs, and Shiori Sekine (University of Pittsburgh) gifted the plasmid pcDNA3.1(+)/Su9-TEV. We also thank Takashi Tatsuta (Max Planck Institute) for his valuable input on the study, Kohei Nishino (Tokushima University) for his help with the proteomic analysis, and Robin Alexander Rothemann (University of Cologne) for generating the AIFM1 cell line. We also thank Hatsumi Abe (The University of Tokyo) for supporting the viral infection experiments, Seitaro Koga and Steffen Hermans-Henschke for providing critical suggestions on the western blotting analysis, and Takahiro Yoshinaka for his early work on this study. This work was supported by the JSPS KAKENHI Grants (No. 25K14949 to M.N., 24K22014 to TI, and 21K19389, 23K26789, and 25H02336 to TK), the Deutsche Forschungsgemeinschaft (DFG, German Research Foundation) - SFB 1218 – Projektnummer 269925409 (to TL and JR), the Medical Research Center Initiative for High Depth Omics (to HK), the JSPS Program for Forming Japan's Peak Research Universities (J-PEAKS, JPJS00420240022 to HK), the Japan Agency for Medical Research and Development (AMED, JP233fa627001 to TI), the Joint Usage and Joint Research Programs, the Institute of Advanced Medical Sciences, Tokushima University (to TK), and the Grant from International Joint Usage/Research Center, the Institute of Medical Science, the University of Tokyo (to TK).

## Author contributions

**Mitsuhiro Nishigori**: Data curation; Formal analysis; Validation; Investigation; Visualization; Methodology. **Serina Hirata**: Data curation; Formal analysis; Investigation; Methodology. **Hidetaka Kosako**: Data curation; Formal analysis; Methodology. **Takeshi Ichinohe**: Data curation; Methodology. **Hendrik Nolte**: Data curation; Formal analysis; Methodology. **Jan Riemer**: Data curation; Methodology. **Thomas Langer**: Conceptualization; Supervision; Funding acquisition; Project administration; Writing—review and editing. **Takumi Koshiba**: Conceptualization; Data curation; Supervision; Funding acquisition; Investigation; Writing—original draft; Project administration; Writing—review and editing.

Source data underlying figure panels in this paper may have individual authorship assigned. Where available, figure panel/source data authorship is listed in the following database record: biostudies:S-SCDT-10_1038-S44318-026-00734-y.

## Disclosure and competing interests statement

The authors declare no competing interests.

# Expanded View Figures

**Figure EV1.  Stress-inducible AIFM1 processing by OMA1.**

(A) $OMA1^{-/-}$ MEFs stably expressing OMA1/Myc or OMA1$^{E324Q}$/Myc were treated for 3 h with either FCCP (40 µM) or valinomycin (1 µg/mL) and analyzed by immunoblotting with the indicated antibodies. OMA1$^{E324Q}$/Myc accumulated stably in cells upon the loss of $\Delta\Psi_m$ (bottom two right lanes), whereas OMA1/Myc was rapidly degraded by autocatalytic proteolysis. (B) Stress-induced substrates (AIFM1 and OPA1) processed by OMA1. $OMA1^{+/+}$ (WT) and $OMA1^{-/-}$ MEFs were incubated for 3 h with FCCP, $H_2O_2$, rotenone, antimycin A (Anti A), oligomycin A (Oligo A), or Anti A/Oligo A and analyzed by western blotting. (C) AIFM1 processing levels observed in $Cox10^{-/-}$ hearts (arrow) were notably reduced in $Cox10^{-/-}OMA1^{-/-}$ hearts ($n = 2$). (D) $OMA1^{-/-}YME1L^{-/-}$ (DKO) MEFs stably expressing either OMA1/Myc or YME1L/HA were incubated for 3 h in the absence (DMSO) or presence of either FCCP (40 µM) or valinomycin (1 µg/mL) and analyzed by immunoblotting (indicated antibodies). (E) OMA1-dependent processing of AIFM1 in $Drp1$ KO cells. The WT or $Drp1$ KO MEFs were treated for 3 h with either FCCP (40 µM) or valinomycin (1 µg/mL) and analyzed by immunoblotting with the indicated antibodies. (F) Illustration shows a hypothetical model of OMA1-mediated AIFM1 processing on the same membrane (left, in a dotted box, acting in *cis*) or on different membranes (right, acting in *trans*). *Mfns*-DKO and *OPA1*-KO are incompetent for OM and IM fusion, respectively. *Drp1*-KO is incompetent for mitochondrial fission.

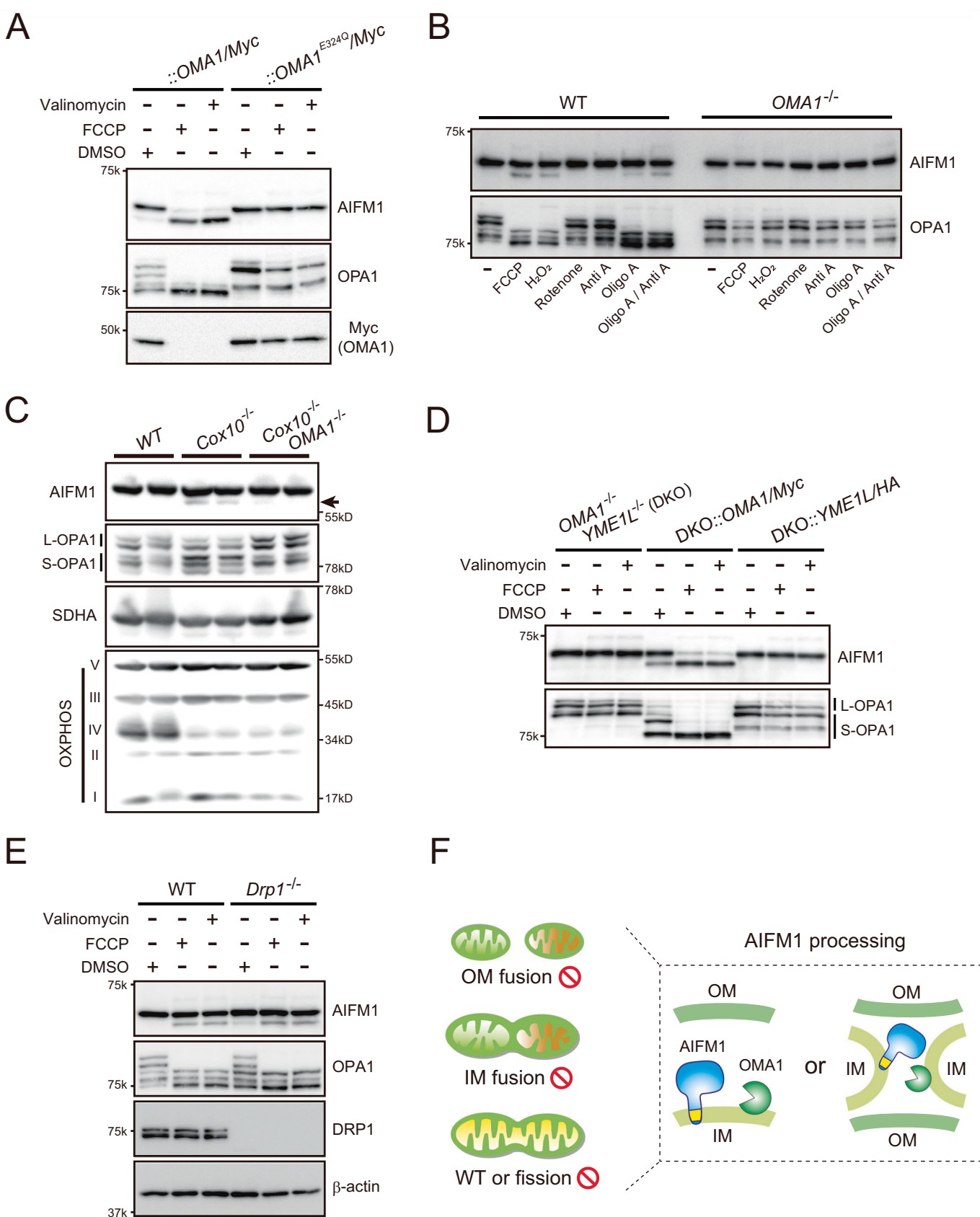

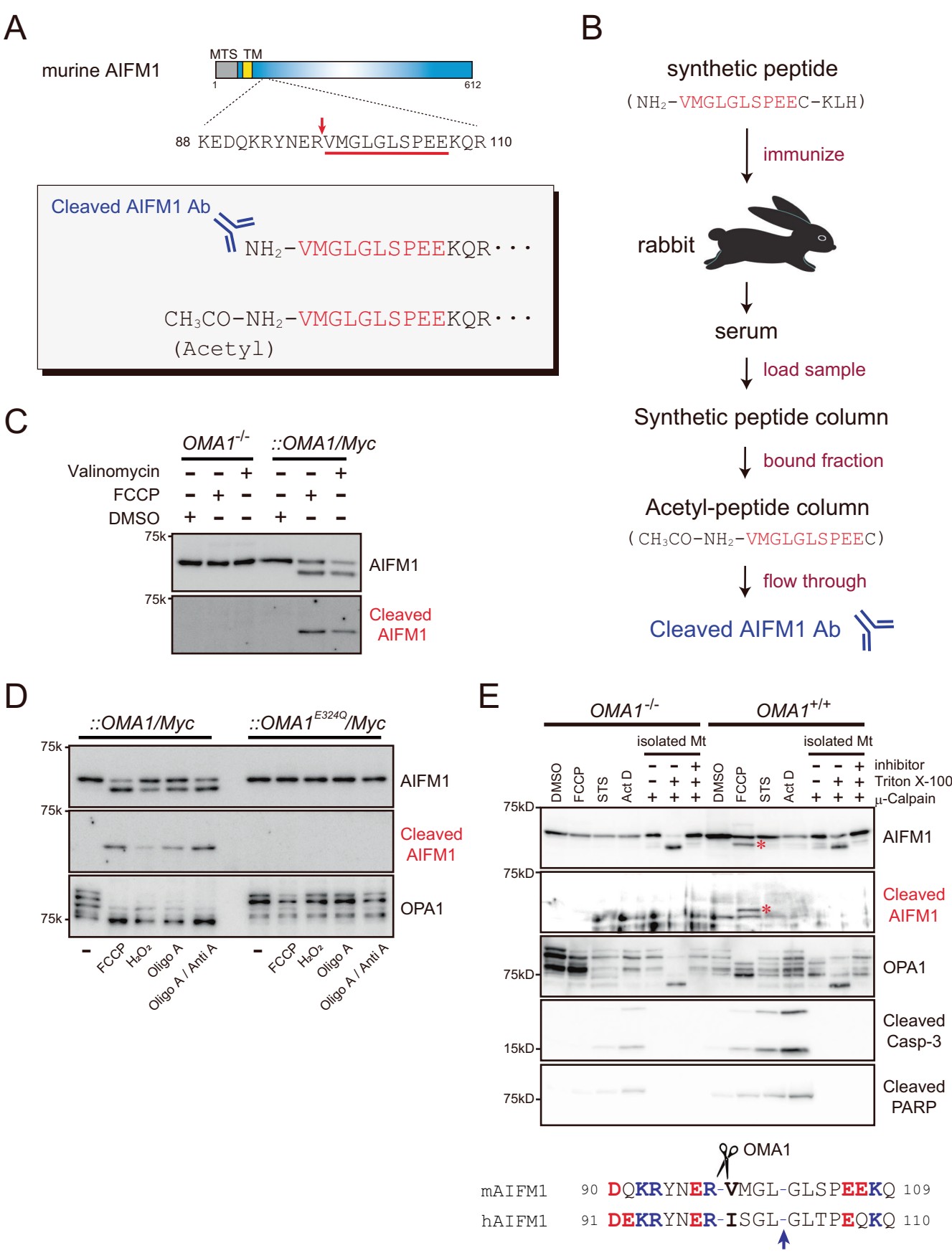

◀ **Figure EV2.   Scheme for generating an N-terminal specific antibody against AIFM1.**

(A) Position of the AIFM1 cleavage site (red arrow) targeted by OMA1. The sequence portion underlined in red was chemically synthesized for immunized rabbits to generate polyclonal antibodies against the region (lower inset, top). The same N-terminally acetylated peptide (lower inset, bottom) was also synthesized for use in affinity purification. (B) Flowchart of the generation of the custom antibody against the N-terminal portion of AIFM1. (C, D) $OMA1^{-/-}$ or $OMA1^{-/-}$ MEFs stably expressing OMA1/Myc were treated for 3 h with either FCCP (40 μM) or valinomycin (1 μg/mL) and analyzed by immunoblotting with the custom cleaved AIFM1 antibody (bottom). The antibody specificity was confirmed by comparing the same immunoblot with that detected by the normal AIFM1 antibody (top). In (D), $OMA1^{-/-}$ MEFs stably expressing OMA1/Myc or $OMA1^{E324Q}$/Myc were treated for 3 h with the indicated chemicals and analyzed by immunoblotting with the indicated antibodies. (E) $OMA1^{+/+}$ and $OMA1^{-/-}$ MEFs were treated with FCCP (20 μM), staurosporine (STS, 1 μM), or actinomycin (Act D, 20 μM) for 12 h to induce cell death. The cells were then analyzed by immunoblotting with the indicated antibodies. In addition to the experiment involving cells treated with proapoptotic drugs, mitochondria were isolated from $OMA1^{+/+}$ and $OMA1^{-/-}$ MEFs, and then incubated for 30 min at 25 °C with μ-calpain, with or without Triton X-100 (0.1%) and a calpain I inhibitor. AIFM1 processing was not observed under these conditions during apoptotic stimuli, whereas μ-calpain-mediated AIFM1 cleavage was OMA1-independent, and the cleavage sites were distinct (see red asterisk). The bottom sequences indicate the AIFM1 cleavage sites by OMA1 and by μ-calpain, as previously reported (Polster et al, 2005).

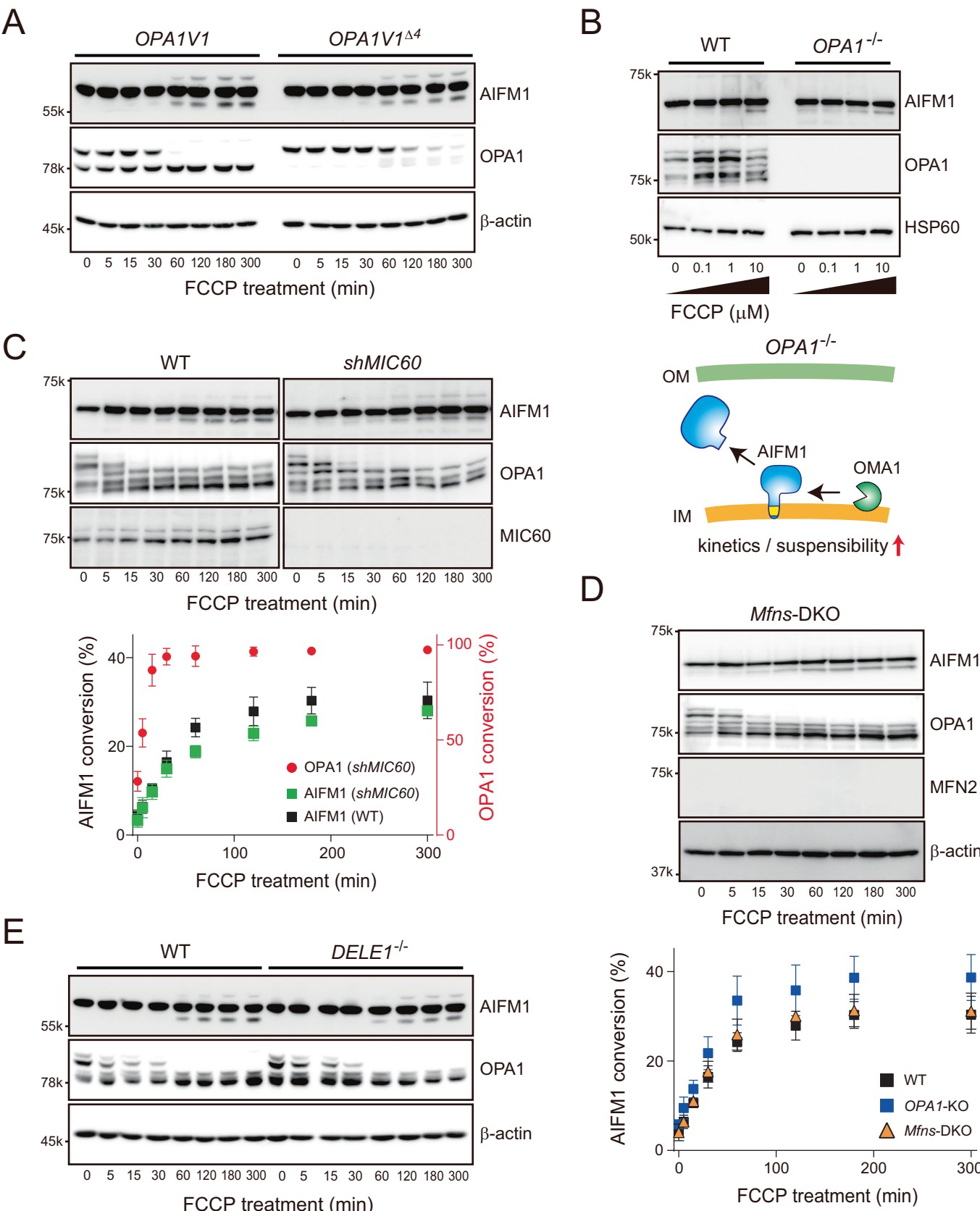

◄

**Figure EV3.  Role of OPA1 in OMA1-dependent AIFM1 processing.**

(A) MEFs only stably expressing either the OPA1 V1 isoform (*OPA1V1*) or its non-cleavable version (*OPA1V1^Δ4^*) (Ahola et al, 2024) incubated with FCCP (40 μM) were collected at the indicated time points (0, 5, 15, 30, 60, 120, 180, and 300 min) and analyzed by western blot. β-Actin blots were used as loading controls for each time point. (B) WT or *OPA1^−/−^* MEFs were treated for 3 h with different concentrations of FCCP (0, 0.1, 1 and 10 μM) and analyzed by immunoblotting with the indicated antibodies. Bottom illustration, Model of OMA1-mediated AIFM1 processing in the absence of OPA1. The loss of OPA1 increases susceptibility to and accelerates the attack of AIFM1 by OMA1. (C) MEFs without or with expression of shRNA against *MIC60* incubated with FCCP (40 μM) were collected at the indicated time points (0, 5, 15, 30, 60, 120, 180, and 300 min) and analyzed by western blot. Bottom graph, OMA1-dependent substrate processing in *shMIC60* MEFs (red, OPA1; green, AIFM1) was quantified (n = 3 biologic replicates) and their conversions (%) were plotted (mean values ± SD). AIFM1 processing in WT MEFs (control) is shown in black. (D) Similar to (C), except that the *Mfns*-DKO MEFs were treated with FCCP (40 μM) for the indicated times. The bottom graph shows the quantitative data (mean values ± SD) obtained from the immunoblots. (E) Similar to (A), except that the WT or *DELE1^−/−^* MEFs were treated with FCCP (40 μM) for the indicated times.

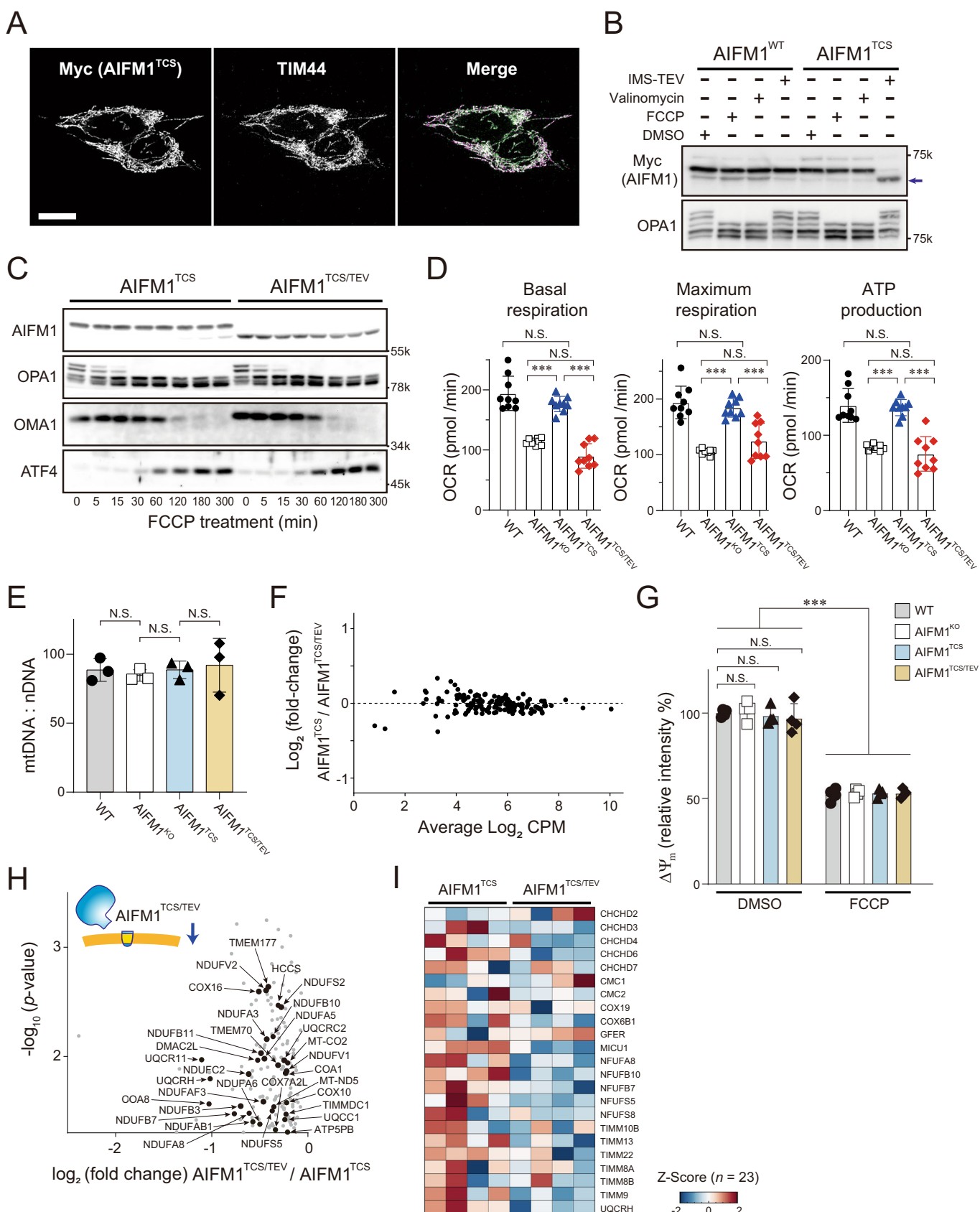

◀ **Figure EV4.  Characterization of AIFM1$^{TCS}$ and AIFM1$^{TCS/TEV}$.**

(A) Subcellular localization of AIFM1$^{TCS}$. Flp-In-293-AIFM1$^{TCS}$/Myc cells were monitored by immunofluorescence against the Myc epitope to determine its subcellular localization (left). Mitochondria in the same cells were also identified by staining with an anti-TIM44 antibody (middle). We confirmed that both AIFM1$^{TCS}$ (magenta) and TIM44 (green) were completely merged in the mitochondria (right). Scale bar, 10 μm. (B) The Flp-In-293-AIFM1$^{WT}$/Myc or -AIFM1$^{TCS}$/Myc cells were treated for 3 h with either FCCP (40 μM) or valinomycin (1 μg/mL) and analyzed by immunoblotting with the indicated antibodies. AIFM1$^{TCS}$/Myc cells were resistant to the FCCP/valinomycin treatment but able to convert to the cleaved form in the presence of IMS-TEV (right lane). Arrow, cleaved AIFM1 product. (C) AIFM1$^{TCS}$- or AIFM1$^{TCS/TEV}$-expressing cells incubated with FCCP (40 μM) were collected at the indicated time points (0, 5, 15, 30, 60, 120, 180, and 300 min) and analyzed by western blot. In this experiment, we confirmed that OPA1 processing as well as OMA1 and ATF4 activations were similar in both cell types, indicating that the TEV protease targeted to the IMS had no significant off-target effect. (D) Basal respiration, maximal respiration, and ATP production were calculated from the OCR experiment of *AIFM1* KO cells (white), cells expressing AIFM1 variants (AIFM1$^{TCS}$ [blue] and AIFM1$^{TCS/TEV}$ [red]), and Flp-In-293 WT cells (black). Data shown are mean ± SD ($n = 9$ biologic replicates). ***$p < 0.001$ and N.S., not significant (by one-way ANOVA followed by Tukey's multiple comparisons test). The exact $p$ values are summarized in Appendix Table S2. See also Fig. 5H. (E) Analysis of mtDNA copy number per nuclear DNA (nDNA) in WT, *AIFM1* KO, and AIFM1 variant Flp-In-293 cells. Data shown are mean ± SD ($n = 3$ biologic replicates). N.S. not significant (by one-way ANOVA followed by Tukey's multiple comparisons test). The exact $p$ values are summarized in Appendix Table S2. (F) Representative MA plot of expressed OXPHOS-related genes (MitoCarta3.0) in AIFM1$^{TCS}$- or AIFM1$^{TCS/TEV}$-expressing cells from transcriptome data. The x-axis represents the $\log_2$ transform of counts per million (CPM), and the y-axis represents the $\log_2$ transform of the fold-change of each gene in AIFM1 variant cells ($n = 1$). A total of 151 of OXPHOS-related genes from the transcriptome data were used in this graph. Expression changes greater than or less than twofold are considered significant, but we identified no genes meeting these criteria. See also Dataset EV5. (G) Comparison of the $\Delta\Psi_m$ in WT, *AIFM1* KO, and AIFM1 variant Flp-In-293 cells treated without (DMSO) or with FCCP. Fluorescence values obtained by measuring TMRM were normalized to the average intensity of DMSO-treated WT cells. Data shown are mean ± SD ($n = 3$ biologic replicates). ***$p < 0.001$ and N.S., not significant (by two-way ANOVA followed by Tukey's multiple comparisons test). The exact $p$ values are summarized in Appendix Table S2. (H) Enlarged box area in Fig. 6A. A total of 123 mitochondrial proteins (MitoCarta3.0) are plotted and 32 of the OXPHOS subunits decreased in AIFM1$^{TCS/TEV}$ cells are labeled. See also Dataset EV4. (I) Alterations in MIA40-pathway in AIFM1$^{TCS}$- and AIFM1$^{TCS/TEV}$-expressing cells. The mitochondrial proteome was sorted by MIA40 substrates (Reinhardt et al, 2020). Heatmap (Z-scores, $n = 23$): minimum ($-2$), blue; maximum (2), red. See also Dataset EV4.

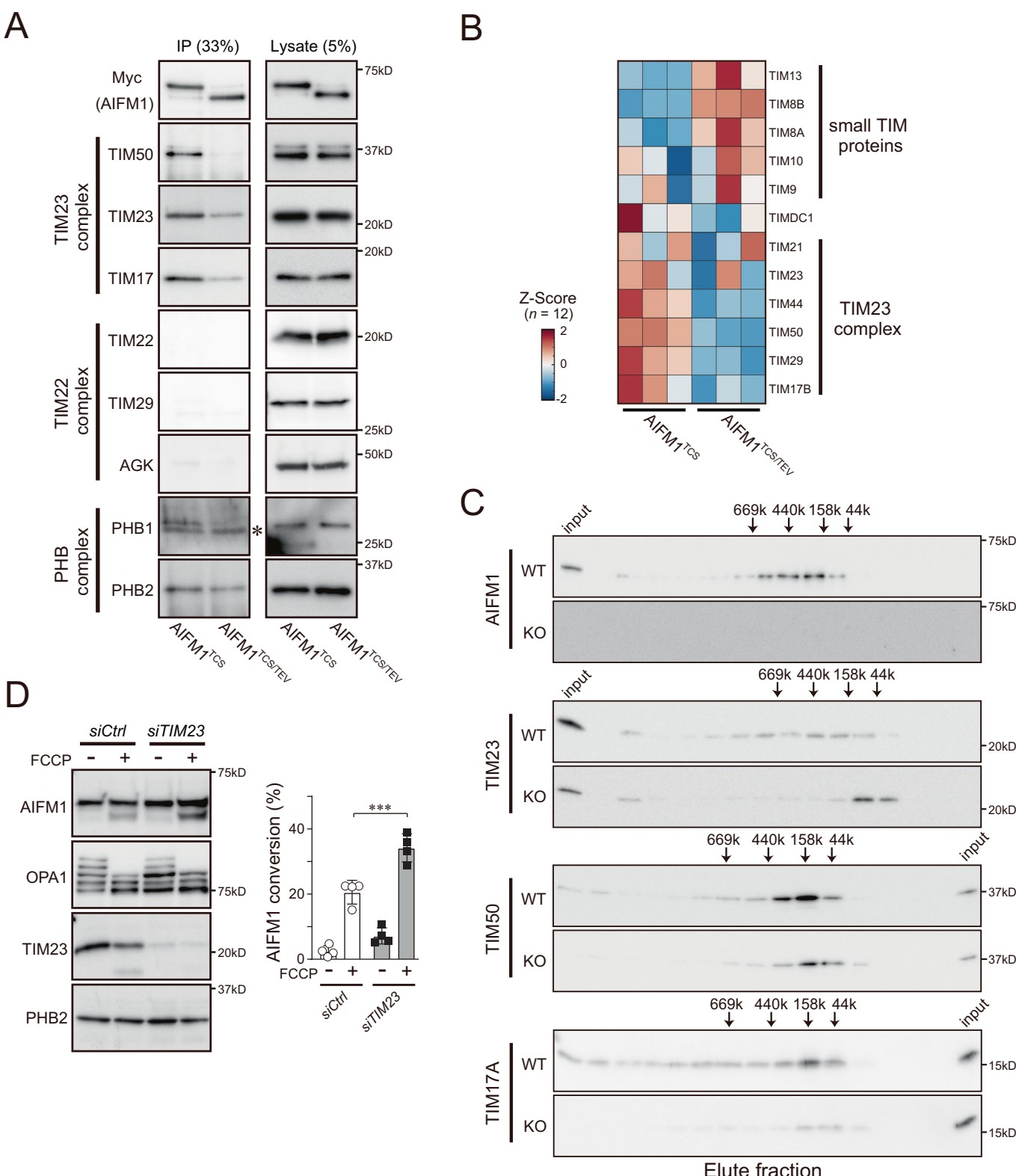

◀ **Figure EV5.  AIFM1 functions as an assembly factor of the TIM23 translocase.**

(A) Mitochondrial extracts from the AIFM1 variant (AIFM1$^{TCS}$ and AIFM1$^{TCS/TEV}$) cells were immunoprecipitated with anti-Myc antibody and subjected to immunoblotting using the indicated antibodies. The IP and lysate samples were loaded at 33% and 5% of the input samples, respectively. Asterisk indicates non-specific band. (B) Heatmap (Z-scores) showing relative enrichment of selected proteins as identified by IP-MS ($n = 3$) in Fig. 5A. Components of the TIM23 complex were enriched in the AIFM1$^{TCS}$ precipitates, whereas small TIM proteins showed a higher affinity for the membrane-dislocated AIFM1 variant, AIFM1$^{TCS/TEV}$. Heatmap (Z-scores, $n = 12$): minimum ($-2$), blue; maximum (2), red. See also Dataset EV3. (C) Gel filtration elution profile of endogenous AIFM1 and each subunit of TIM23 complex extracted from the mitochondrial fraction of Flp-In-293 WT cells and *AIFM1* KO cells. The positions corresponding to the elution of standard markers molecular mass and input of samples are indicated, and fractions were analyzed by western blotting with indicated antibodies. (D) HeLa cells transfected with siRNA against *TIM23* were treated for 3 h with or without FCCP (40 μM) and analyzed by immunoblotting as indicated. The graph on the right shows the quantification of AIFM1 bands from the immunoblot analyzed by densitometry. Data shown are mean ± SD ($n = 4$ biologic replicates), and ***$p < 0.001$ (by one-way ANOVA followed by Tukey's multiple comparisons test). The exact $p$ values are summarized in Appendix Table S2.

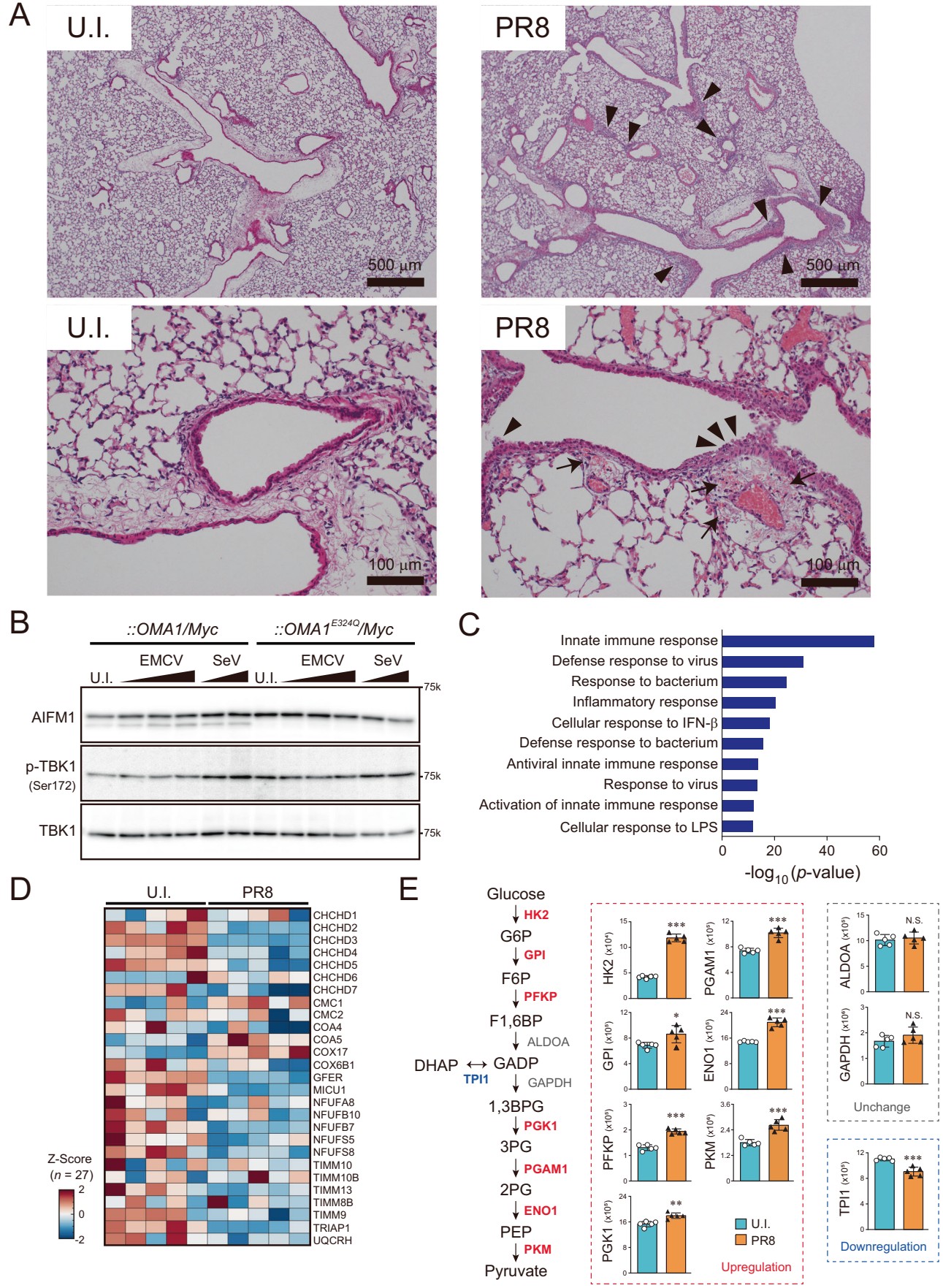

◀

**Figure EV6.  Functional relevance of AIFM1 processing in lungs.**

(**A**) On day 7 post-infection (1000 pfu), the lungs were obtained from each PR8-infected or uninfected (U.I.) mouse, sectioned, and analyzed for histopathology following hematoxylin and eosin staining. The lower two panels show higher magnification images. The arrowheads in the PR8-infected lungs indicate detachment of the bronchial epithelium, and the arrows show perivascular hemorrhage. Scale bars, 500 μm (top) and 100 μm (bottom), respectively. (**B**) $OMA1^{-/-}$ MEFs that stably express OMA1/Myc or OMA1$^{E324Q}$/Myc were either infected with SeV (4 or 10 HAU/mL) or EMCV (MOI of 0.1, 0.5, or 1) for 16 h, and the cellular lysates were analyzed by immunoblotting with the indicated antibodies. U.I. uninfected. (**C**) GO enrichment analysis of the whole proteome from the top 10 upregulating biologic processes in PR8-infected lungs (by modified Fisher's exact test). The exact $p$ values are summarized in Appendix Table S2. See also Dataset EV6. (**D**) Alterations in MIA40-pathway in uninfected (U.I.) and PR8-infected lungs. The mitochondrial proteome was sorted by MIA40 substrates (Reinhardt et al, 2020). Heatmap (Z-scores, $n = 27$): minimum ($-2$), blue; maximum (2), red. See also Dataset EV6. (**E**) PR8 infection induces glycolysis in lungs. Quantitative DIA intensities of glycolytic enzymes from the whole proteome analysis were plotted. The left panel shows a schematic overview of glucose metabolism and altered glycolytic enzymes (red, upregulated; blue, downregulated; gray, unchanged). Data shown are mean ± SD ($n = 5$). *$p < 0.05$, **$p < 0.01$, ***$p < 0.001$, and N.S., not significant (by Student's $t$ test). The exact $p$ values are summarized in Appendix Table S2. See also Dataset EV6.

