## [Peer Review File · The EMBO Journal]

Stress-induced OMA1-mediated cleavage of AIFM1 suppresses cell growth by controlling mitochondrial OXPHOS activity

Mitsuhiro Nishigori, Serina Hirata, Hidetaka Kosako, Takeshi Ichinohe, Hendrik Nolte, Jan Riemer, Thomas Langer, and Takumi Koshiba

Corresponding author: Takumi Koshiba (koshiba@kyudai.jp)

Review Timeline:

Submission Date:	19th May 25
Editorial Decision:	29th Jun 25
Revision Received:	17th Nov 25
Editorial Decision:	30th Jan 26
Revision Received:	4th Feb 26
Accepted:	13th Feb 26

Editor: Daniel Klimmeck

Transaction Report:

Dear Dr Koshiba,

Thank you again for the submission of your manuscript (EMBOJ-2025-121411) to The EMBO Journal, as well as for your patience with our feedback at this time. As mentioned earlier, your study was assessed by three reviewers with expertise in mitochondrial stress signaling and cell death, whose comments are enclosed below.

As you will see from the experts' reports, the referees acknowledge the analysis and potential interest and value of your findings. However, they also express important issues that will need to be conclusively addressed before they can support publication at the EMBO Journal. In more detail, referee #2 points to major conceptual gaps remaining and requests more insights into the exact context relevant to the uncovered OMA1-AIFM1 connection (ref#2, pts.1,2,3). This expert also raises issues with the completeness of support for functional cooperation of TIM23 with AIFM1 (ref#2, pt.5; see also ref#1, pt.1). Further, reviewer #3 requests consideration of complementary stress inducers to corroborate your results and states that mechanistic insights into the affected protein import control remain too limited. Further, the reviewers raise a number of issues related to the presentation of the findings, additional controls and improved methods annotation required, statistics applied and overall discussion of related literature, that would need to be conclusively addressed to achieve the level of robustness and clarity needed for The EMBO Journal.

Given the overall interest stated and broader angle of your findings, we are able to invite you to revise your manuscript experimentally to address the referees' comments. However, please note that the extent of revisions requested appear threshold in our view for the amount of complementary work we typically invite for our venue; also, I need to stress that we do require strong support from the referees on a revised version of the study in order to move on to publication of the work.

I would appreciate if you could contact me during the next weeks for exchange e.g. a video call to discuss your perspective on the comments and potential plan for revisions.

Please feel free to contact me if you have any questions or need further input on the referee comments.

When submitting your revised manuscript, please carefully review the instructions below.

Please feel free to approach me any time should you have additional questions related to this.

Thank you for the opportunity to consider your work for publication.

I look forward to your revision.

Kind regards,

Daniel Klimmeck

Daniel Klimmeck, PhD
Senior Editor
The EMBO Journal

Instruction for the preparation of your revised manuscript:

2) individual production quality figure files as .eps, .tif, .jpg (one file per figure).

3) a .docx formatted letter INCLUDING the reviewers' reports and your detailed point-by-point response to their comments. As part of the EMBO Press transparent editorial process, the point-by-point response is part of the Review Process File (RPF), which will be published alongside your paper.

4) a complete author checklist, which you can download from our author guidelines ([https://wol-prod-cdn.literatumonline.com/pb-assets/embo-site/Author Checklist%20-%20EMBO%20J-1561436015657.xlsx](https://wol-prod-cdn.literatumonline.com/pb-assets/embo-site/Author%20Checklist%20-%20EMBO%20J-1561436015657.xlsx)). Please insert information in the checklist that is also reflected in the manuscript. The completed author checklist will also be part of the RPF.

6) It is mandatory to include a 'Data Availability' section after the Materials and Methods. Before submitting your revision, primary datasets produced in this study need to be deposited in an appropriate public database, and the accession numbers and database listed under 'Data Availability'. Please remember to provide a reviewer password if the datasets are not yet public (see <https://www.embopress.org/page/journal/14602075/authorguide#datadeposition>).

7) Our journal encourages inclusion of *data citations in the reference list* to directly cite datasets that were re-used and obtained from public databases. Data citations in the article text are distinct from normal bibliographical citations and should directly link to the database records from which the data can be accessed. In the main text, data citations are formatted as follows: "Data ref: Smith et al, 2001" or "Data ref: NCBI Sequence Read Archive PRJNA342805, 2017". In the Reference list, data citations must be labeled with "[DATASET]". A data reference must provide the database name, accession number/identifiers and a resolvable link to the landing page from which the data can be accessed at the end of the reference. Further instructions are available at .

Numerical data can be provided as individual .xls or .csv files (including a tab describing the data). For 'blots' or microscopy, uncropped images should be submitted (using a zip archive or a single pdf per main figure if multiple images need to be supplied for one panel).

Additional information on source data and instruction on how to label the files are available

9) We replaced Supplementary Information with Expanded View (EV) Figures and Tables that are collapsible/expandable online (see examples in <https://www.embopress.org/doi/10.15252/emboj.201695874>). A maximum of 5 EV Figures can be typeset. EV Figures should be cited as 'Figure EV1, Figure EV2' etc. in the text and their respective legends should be included in the main text after the legends of regular figures.

11) For data quantification: please specify the name of the statistical test used to generate error bars and P values, the number (n) of independent experiments (specify technical or biological replicates) underlying each data point and the test used to calculate p-values in each figure legend. The figure legends should contain a basic description of n, P and the test applied. Graphs must include a description of the bars and the error bars (s.d., s.e.m.).

We realize that it is difficult to revise to a specific deadline. In the interest of protecting the conceptual advance provided by the work, we recommend a revision within 3 months (27th Sep 2025). Please use the link below to submit your revision:

Referee #1:

This exciting work identifies and characterizes a new substrate of the mitochondrial metalloprotease OMA1. Using chemical crosslinking followed by immunoprecipitation of OMA1 and mass spectrometry, the authors identified the inner membrane protein AIFM1 as an OMA1-binding protein. They also employed a proteomic approach that detects neo-amino-terminal peptides in response to FCCP treatment-which activates OMA1-and found that AIFM1 is a potential substrate. Previous studies have shown that AIFM1 is involved in apoptosis, protein import into the intermembrane space, and oxidative phosphorylation.

The authors demonstrated that AIFM1 is cleaved in an OMA1-dependent manner when cells are treated with FCCP or when Cox10 is knocked out in mouse hearts, both of which activate OMA1. They further mapped the cleavage site in AIFM1 and generated both a processing-defective mutant and a constitutively cleaved mutant. Using these mutants, the authors showed that cells expressing only the cleaved form of AIFM1 exhibit reduced complex I activity, impaired mitochondrial respiration, decreased cellular ATP levels, and diminished cell proliferation.

Through proteomic comparison of the interactomes between cleavage-resistant and cleaved forms, followed by confirmation via immunoprecipitation, the authors found that many electron transport chain components specifically interact with uncleaved AIFM1. These interactions are consistent with previous studies showing that AIFM1 functions as a chaperone for these complexes.

The authors also found that cleavage-resistant AIFM1 preferentially interacts with components of the TIM23 translocase, including TIM23, TIM17, and TIM50. The functional importance of this interaction was demonstrated through in vitro import assays, which showed partial impairment of TIM23 substrate import. Based on these findings, the authors propose that AIFM1 plays a vital role in protein import mediated by the TIM23 translocase, and that this function is inhibited by OMA1-mediated cleavage under stress conditions.

Overall, this study reveals a novel mechanism by which OMA1 regulates protein import and mitochondrial bioenergetics in response to stress through the cleavage of its newly identified substrate, AIFM1. The authors effectively integrate multiple cutting-edge proteomic approaches with comprehensive biochemical analyses. This work significantly and uniquely advances our understanding of mitochondrial stress responses.

My specific comments are:

1. AIFM1 may function as a stable subunit, an assembly factor, or a regulator of the import activity of the TIM23 translocase. The authors should test whether AIFM1 is a component of the TIM23 complex using BN-PAGE. Is the integrity of the TIM23 translocase compromised in AIFM1 KO cells or AIFM1 KO cells expressing AIFM1(TCS/TEV)?
2. Related to the above point, does FCCP treatment affect the integrity of the TIM23 complex in an OMA1-dependent manner?
3. Previous studies have reported that AIFM1 is cleaved near the OMA1 cleavage site during apoptosis. Are the functional consequences of these cleavages the same or different?

Referee #2:

General summary and opinion about the principle significance of the study, its questions and findings

The authors identify OMA1 as a protease, which is involved in the cleavage of AIFM1. The manuscript demonstrates the nice set of data which identifies the AIFM1 recognition and cleavage site by OMA1 and the stress conditions that activate this cleavage. The data are convincing.

The most exciting finding of this paper is the identification of OMA1 as a protease processing AIFM1. Based on a the large body of literature, this cleavage is needed for a canonical function related to the programmed cell death. A putative protease responsible for this action has been unknown. Is this OMA1? This notion is however not sufficiently addressed. This is the major weakness and the large gap left open.

Another aspect that requires strengthening are conclusions made on the basis of the experiments with the use of an artificial construct of AIFM1 released from the membrane. None of the functional conclusion can be drawn from such a system exclusively and without taking under consideration and addressing experimentally the following remarks below.

Major concerns essential to be addressed to support the conclusions

- 1) The authors seem to avoid the most important question, still unanswered despite the large body of the literature. The AIFM1 to be translocated and induce the cell death must be proteolytically released from its TM anchored to the inner membrane. A putative protease responsible for this action has been unknown. Is it OMA1 that contributes to this process? The important and pending question which must be addressed whether this function of OMA1 is prerequisite for execution of the cell death and its lack contribute to cell death resistance.
- 2) Another important and pending question is the interaction with MIA40, described originally in Salscheider et al., 2022, and structurally explored further by Brosey et al 2025 and via AlphaFold by Mussulini et al. 2025. How this interaction with CHCHD4/MIA40 modulates the cleavage of AIFM1 by OMA1?
- 3) Salscheider et al and Brosey et al report the role of AIFM1 in organizing the MIA40 pathway. How the cleavage by OMA1 influences this function?
- 4) Mussulini et al. reports an interesting functional role for interaction of AIFM1 with MIA40 in regulating the cell death resistance likely by inhibiting the AIFM1 release from mitochondria. Does the cleavage by OMA1 influences this function?
- 5) The part concerning TIM23 is based on the physical interaction with the TIM23 complex. This interaction is not sufficient to support the functional claims, which remind fully unjustified. In the mitochondrial membranes many various complexes are close to each other. Furthermore, the interaction can be the remnant of the interaction with OMA1, prohibitins or other players. There is no evidence that would justified the functional consequences of such interaction. Without further experimental development this part should be minimized. The conclusion of this part, mostly based on the artificial construct of AIFM1 should be minimal. The above mentioned papers should be cited.

Minor concerns that should be addressed

The authors focus the introduction on the mechanisms involved in morphology of mitochondria. However the paper is about OMA and AIFM1 and the introduction should fine-tuned in one of the directions, or both. The intro seems to deal with another topic that the experiments.

There is a typo in the abstract: AIFM1 is...the membrane-anchored inter membrane space (IMS) protein, not "inner membrane space".

Referee #3:

Nishigori et al. used co-IP with myc-tagged Oma1 to identify binding partners. Interestingly, they pulled down AIFM1, along with Opa1, prohibitins and a series of MICOS proteins. A MassSpec assay for Oma1-dependent changes in the N-termini of peptides further suggested that AIFM1 is a substrate for Oma1. Cleavage was induced by FCCP as well as COX10 deletion. The authors also confirmed that other mitochondrial proteases were not involved, and they found that the cleavage of AIFM1 is much slower than that of Opa1. The authors identified the Oma1 cleavage site in AIFM1, mutated it, and replaced it with a TEV cleavage site to study the consequences of cleavage events without stress-inducing conditions. The authors then demonstrate that uncleaved AIFM1 is part of a larger protein complex. Other proteins in this complex include ETC complex I proteins and inner membrane carrier proteins. They also found that the uncleaved form is needed for respiration, consistent with a role in promoting complex I function. The authors then demonstrated that the levels of a large number of other proteins imported by TIM23 are also influenced by the cleavage state of AIF. In addition, AIFM1 co-IPs components of the TIM23 complex, consistent with earlier studies showing that it is an integral part of the import machinery.

The identification of AIF as a substrate for Oma1 and the observation of slow cleavage kinetics are potentially significant findings. However, the discussion of their functional consequences is unclear. For instance, the heading "AIFM1 ensures cell proliferation by controlling OXPHOS activity" suggests a direct control of OXPHOS, but the data are more consistent with a general role as an oxidoreductase that aids protein import during mitochondrial biogenesis, as also noted toward the end of the discussion. Stress-induced cleavage by Oma1 would, therefore, hinder import in a response that is considerably slower than Opa1 cleavage, but aligns with the kinetics of Dele1 cleavage by Oma1.

Some comments:

FCCP is probably not the best inducer for studying the physiological relevance of AIFM1 cleavage, because it will also block protein import during biogenesis, which seems to be one of the main functions of AIF.

It is unclear why AIF, which is reported to assist the disulfide relay, including MIA40 and Erv1, would mainly affect the import of complex I proteins that pass through Tim23. Are the authors suggesting a general failure of protein import in AIF mutants? The

differences in growth under respiring conditions seem relatively minor for a protein that would halt import under stress conditions. Perhaps AIF cleavage is more important for apoptosis than it is for stopping protein import.

Small points:

Fig. 1D. suggests that AK2 and Tom40 are also Oma1 substrates. Considering that Tom40 is in the outer membrane, is the cleavage site at least exposed to the IMS?

Fig. 1E. AIF cleavage is much stronger here than on other blots. Is this because of a longer incubation time than in other experiments?

Page 8. The 2nd paragraph does not show AIF cleavage is dispensable for fission and fusion. It is the other way around.

Fig. 3D suggests that AIF cleavage is accelerated by loss of Opa1, which could be an indirect effect of mitochondrial dysfunction. The authors should test causality with an uncleavable Opa1 mutant.

Fig. 5H. Rescue of a respiratory defect by uncleaved AIFM1 is convincing, but how does it compare with wild type cells? The authors should include that.

Fig. 5. If all these proteins colP, why is the complex only about 300kDa? Or is there a core complex with other proteins such as the import machinery and these are only present in small amounts? Does cycloheximide change the size of the BNGE complexes?

Also, it seems that CI and a number of carrier proteins stand out, but these are imported by different Tims (Tim22 vs Tim23).

Does that mean AIF affects both processes?

[Reviewer #1 was very favorable and asked us to address three specific comments]

“This exciting work identifies and characterizes a new substrate of the mitochondrial metalloprotease OMA1. Using chemical crosslinking followed by immunoprecipitation of OMA1 and mass spectrometry, the authors identified the inner membrane protein AIFM1 as an OMA1-binding protein. They also employed a proteomic approach that detects neo-amino-terminal peptides in response to FCCP treatment-which activates OMA1-and found that AIFM1 is a potential substrate. Previous studies have shown that AIFM1 is involved in apoptosis, protein import into the intermembrane space, and oxidative phosphorylation.

The authors demonstrated that AIFM1 is cleaved in an OMA1-dependent manner when cells are treated with FCCP or when Cox10 is knocked out in mouse hearts, both of which activate OMA1. They further mapped the cleavage site in AIFM1 and generated both a processing-defective mutant and a constitutively cleaved mutant. Using these mutants, the authors showed that cells expressing only the cleaved form of AIFM1 exhibit reduced complex I activity, impaired mitochondrial respiration, decreased cellular ATP levels, and diminished cell proliferation.

Through proteomic comparison of the interactomes between cleavage-resistant and cleaved forms, followed by confirmation via immunoprecipitation, the authors found that many electron transport chain components specifically interact with uncleaved AIFM1. These interactions are consistent with previous studies showing that AIFM1 functions as a chaperone for these complexes.

The authors also found that cleavage-resistant AIFM1 preferentially interacts with components of the TIM23 translocase, including TIM23, TIM17, and TIM50. The functional importance of this interaction was demonstrated through in vitro import assays, which showed partial impairment of TIM23 substrate import. Based on these findings, the authors propose that AIFM1 plays a vital role in protein import mediated by the TIM23 translocase, and that this function is inhibited by OMA1-mediated cleavage under stress conditions.

Overall, this study reveals a novel mechanism by which OMA1 regulates protein import and mitochondrial bioenergetics in response to stress through the cleavage of its newly identified substrate, AIFM1. The authors effectively integrate multiple cutting-edge proteomic approaches with comprehensive biochemical analyses. This work significantly and uniquely advances our understanding of mitochondrial stress responses.”

We are deeply grateful for the reviewer’s encouraging comments on the study.

My specific comments are:

1. “AIFM1 may function as a stable subunit, an assembly factor, or a regulator of the import activity of the TIM23 translocase. The authors should test whether AIFM1 is a component of the TIM23 complex using BN-PAGE. Is the integrity of the TIM23 translocase compromised in AIFM1 KO cells or AIFM1 KO cells expressing AIFM1(TCS/TEV)?”

We sincerely appreciate this comment. To address this comment, we performed size-exclusion chromatography to determine whether AIFM1 is endogenously associated with the TIM23 complex. Comparison of the elution profiles of the TIM23 translocase in mitochondrial extracts from wild-type (WT) or AIFM1 knockout (KO) cells revealed a significant shift in the TIM23 translocase to a lower molecular weight in the AIFM1 KO cells. As AIFM1 migration in WT cells is consistent with that of the TIM23 translocase, these results imply that AIFM1 is part of the TIM23 complex. We included this result in the revised manuscript (Fig EV5C).

We also performed a native antibody-based mobility shift assay (NAMOS: Swamy *et al*, *J Immunology Methods* 2007) to evaluate the TIM23–AIFM1 complex. Mitochondrial extracts from AIFM1 KO cells or cells expressing Myc-tagged AIFM1 were incubated with or without a Myc antibody and subjected to CN-PAGE, followed by immunodetection using a TIM23 antibody. The addition of the Myc antibody resulted in a mobility shift in the TIM23 complex (= AIFM1/Myc), providing support that AIFM1 is part of the TIM23 complex (see Fig C1, red asterisk).

Fig C1. NAMOS assay to detect the AIFM1-TIM23 interaction.

2. “Related to the above point, does FCCP treatment affect the integrity of the TIM23 complex in an OMA1-dependent manner?”

Thank you for this comment. We performed size-exclusion chromatography to investigate whether mitochondrial depolarization (i.e., stress-induced activation of OMA1) affects the integrity of the TIM23 complex. Treatment of WT cells with FCCP induced AIFM1 processing by activated OMA1, leading to a slight shift in the AIFM1 peak to a lower molecular weight. This change was not observed in the fraction from the OMA1 KO cells (see Fig C2). Similar trends were observed in the TIM23 translocase. Overall, however, the mobility shifts of both AIFM1 and TIM23 were minimal under FCCP conditions.

Fig C2. Size-exclusion chromatography of the endogenous AIFM1 and TIM23 extracted from *OMA1*^{+/+} and *OMA1*^{-/-} MEFs.

3. “Previous studies have reported that AIFM1 is cleaved near the OMA1 cleavage site during apoptosis. Are the functional consequences of these cleavages the same or different?”

We sincerely appreciate this comment. Using *OMA1*^{+/+} and *OMA1*^{-/-} MEFs, we tested for AIFM1 cleavage under apoptotic stimuli (staurosporine and actinomycin D), but did not observe any processing. We included this result in the revised manuscript (Fig EV2E). According to the literature (Susan *et al*, *Nature* 1999; Polster *et al*, *J Biol Chem* 2005), the μ -calpain cleavage site in AIFM1 is located between amino acid positions 101 and 102. This differs from our observation and is downstream of the OMA1 recognition site (Fig EV2E, bottom sequence). Interestingly, our *in vitro* experiment showed that μ -calpain-mediated AIFM1 cleavage is OMA1-independent, and that the cleavage sites are distinct, as revealed by our specific antibody against the N-terminus of AIFM1 (Fig EV2E). The experiment also showed that μ -calpain cleavage occurs only in the presence of Triton X-100, likely outside of the mitochondria.

【Referee #2 was favorable and as provided us very constructive comments with 5 specific and minor points】

“The authors identify OMA1 as a protease, which is involved in the cleavage of AIFM1. The manuscript demonstrates the nice set of data which identifies the AIFM1 recognition and cleavage site by OMA1 and the stress conditions that activate this cleavage. The data are convincing.

The most exciting finding of this paper is the identification of OMA1 as a protease processing AIFM1. Based on a large body of literature, this cleavage is needed for a canonical function related to the programmed cell death. A putative protease responsible for this action has been unknown. Is this OMA1? This notion is however not sufficiently addressed. This is the major weakness and the large gap left open.

Another aspect that requires strengthening are conclusions made on the basis of the experiments with the use of an artificial construct of AIFM1 released from the membrane. None of the functional conclusion can be drawn from such a system exclusively and without taking under consideration and addressing experimentally the following remarks below.”

We sincerely appreciate these comments and respond to them point-by-point below.

Major concerns essential to be addressed to support the conclusions:

1. *“The authors seem to avoid the most important question, still unanswered despite the large body of the literature. The AIFM1 to be translocated and induce the cell death must be proteolytically released from its TM anchored to the inner membrane. A putative protease responsible for this action has been unknown. Is it OMA1 that contributes to this process? The important and pending question which must be addressed whether this function of OMA1 is prerequisite for execution of the cell death and its lack contribute to cell death resistance.”*

Thank you for your comment. We carefully reviewed the literature on the role of AIFM1 in cell death and found that its contribution to programmed cell death is very controversial, varying by the cell type and apoptotic insult (Hangen *et al*, *Trends Biochem Sci* 2010; Wischhof *et al*, *eBioMedicine* 2022). As the reviewer noted, the putative protease responsible for this action remains unknown. Therefore, we tested whether staurosporine (STS)- or actinomycin D (ActD)-induced cell death leads to AIFM1 cleavage via the OMA1 pathway. We confirmed that treating cells with these drugs did not induce AIFM1 processing under our experimental conditions, although OMA1 acts as a pro-apoptotic factor (*Fig EV2E* in the revised manuscript) (Rivera-Mejías *et al*, *Cell Rep* 2023). Previous studies reported that the cysteine protease μ -calpain cleaves

AIFM1 *in vitro* (Polster *et al*, *J Biol Chem* 2005). We also observed the cysteine protease-cleaved AIFM1, but only in the presence of Triton X-100, indicating that processing occurred outside of the mitochondria (*Fig EV2E*). We confirmed that the cleavage site via μ -calpain differs from that of OMA1 (see our specific antibody against the N-terminus of AIFM1).

2. “Another important and pending question is the interaction with MIA40, described originally in Salscheider *et al.*, 2022, and structurally explored further by Brosey *et al* 2025 and via AlphaFold by Mussulini *et al.* 2025. How this interaction with CHCHD4/MIA40 modulates the cleavage of AIFM1 by OMA1?”

We sincerely appreciate this comment. In response, we generated an AIFM1 mutant that cannot bind to MIA40 (AIFM1^{T503D/V504K/G505P} mutant; Fagnani *et al*, *Structure* 2024). Using cells expressing the AIFM1^{DKP} variant, we examined how the interaction between AIFM1 and MIA40 modulates AIFM1 cleavage by OMA1. We found that the AIFM1^{DKP} variant markedly destabilizes the protein, making it more susceptible to OMA1 attack under mitochondrial stress conditions. This result is described in the revised manuscript (*Appendix Fig S3*). We hypothesize that the physical interaction between AIFM1 and MIA40 affects AIFM1's susceptibility to OMA1-dependent proteolytic processing by stabilizing its dimeric structure in mitochondria (Mussulini *et al*, *EMBO Rep* 2025), increasing its folding stability and reducing flexibility around the OMA1 cleavage site.

3. “Salscheider *et al* and Brosey *et al* report the role of AIFM1 in organizing the MIA40 pathway. How the cleavage by OMA1 influences this function?”

We sincerely appreciate this comment. To address this comment, we used non-cleavable and soluble AIFM1 variant-expressing cell lines and explored the whole cell/mitochondrial proteome, focusing on MIA40 substrates (Reinhardt *et al*, *Biochim Biophys Acta* 2020). The result described in the revised manuscript (*Fig EV4I*). We confirmed that MIA40 substrate protein levels were significantly decreased in cells expressing the soluble AIFM1 variant (AIFM1^{TCS/TEV}) relative to those expressing the non-cleavable version, indicating that spatial organization of AIFM1 influences the MIA40 pathway.

The reviewer also raised a critical point about our previous conclusion, which was based on experiments using an artificial construct of AIFM1 released from the membrane (= AIFM1^{TCS/TEV}). Therefore, we next attempted to address this issue using a physiologic approach. Notably, we found that influenza A viral infection stress activates OMA1 and increases AIFM1 processing in mouse lung (*Fig 7G* in the revised manuscript), demonstrating its physiologic effect *in vivo*. We therefore again explored the physiologic role of

AIFM1 topology in biogenesis relating to the MIA40 pathway in tissue using the proteome approach. We found that the lungs of virus-infected mice had significantly higher levels of proteins from immune response pathways (*Fig EV6C*), whereas the levels of mitochondrial proteins corresponding to the MIA40 pathway (*Fig EV6D*) and OXPHOS complex subunits (*Fig 7H*) were much lower than in uninfected mice. Due to the significance of these results, we included a new section in the *Results* section (pages 17-18) titled "*Physiologic impact of AIFM1 cleavage in vivo*" and added these new data (*Dataset EV6*).

4. "Mussulini *et al.* reports an interesting functional role for interaction of AIFM1 with MIA40 in regulating the cell death resistance likely by inhibiting the AIFM1 release from mitochondria. Does the cleavage by OMA1 influences this function?"

Thank you for this comment. As previously reported by Mussulini *et al* (*EMBO Rep* 2025), we confirmed that the functional role of the interaction between AIFM1 and MIA40 is to stabilize AIFM1 protein folding and confer resistance to OMA1 attack (related to *point #2*). To address the reviewer's comment, we tested whether the spatial organization of AIFM1 in mitochondria would affect drug-induced cell death (i.e., STS). By evaluating cell death through monitoring LDH release, we confirmed that AIFM1 certainly regulates cell death resistance, as previously reported (*Appendix S4A* in the revised manuscript; WT vs AIFM1^{KO}). However, we observed no difference in cell death between AIFM1 variants (AIFM1^{TCS} vs AIFM1^{TCS/TEV}). Consistent with the LDH assay results, PARP cleavage levels in cells expressing the AIFM1 variant that were treated with either STS or overexpressed the pro-apoptotic protein Bax did not change (*Appendix Fig S4B*). Based on these results, we hypothesize that AIFM1 cleavage by OMA1 does not correlate with apoptotic insult. However, it is conceivable that the apoptosis-related attenuation of $\Delta\Psi_m$ activates OMA1, which would subsequently process AIFM1 under certain circumstances. We discuss this possibility in the revised *Discussion* (*p.18, line 4 from bottom* to *p.19, line 5*; "Therefore, the previously reported...required for AK2's role in apoptosis.").

5. "The part concerning TIM23 is based on the physical interaction with the TIM23 complex. This interaction is not sufficient to support the functional claims, which remind fully unjustified. In the mitochondrial membranes many various complexes are close to each other. Furthermore, the interaction can be the remnant of the interaction with OMA1, prohibitins or other players. There is no evidence that would justified the functional consequences of such interaction. Without further experimental development this part should be minimized. The conclusion of this part, mostly based on the artificial construct of AIFM1 should be minimal."

We sincerely appreciate this comment. As the reviewer noted, the mitochondrial inner membrane (MIM) contains many different protein complexes, and our previous data may not be sufficient to support our experimental claim. In response, we conducted extensive biochemical experiments that reinforced our conclusion.

i). First, we verified whether AIFM1 binds to the TIM23 translocase during import or after assembly in the MIM. To exclude the possibility that newly imported AIFM1 binds to the TIM23 complex in the mitochondria, we performed an endogenous immunoprecipitation (IP) assay in the presence of cycloheximide (CHX), which inhibits cytosolic protein synthesis. The results are described in the revised manuscript (*Fig 6C*). We confirmed that CHX did not diminish the observed AIFM1–TIM23 interaction, thus excluding the possibility that the interaction occurred during protein import.

ii). Next, we performed a proximity ligation assay (PLA), a widely used method for detecting protein-protein interactions *in situ* (Alam, *Curr Protoc Immunol* 2018). The results of the PLA confirmed the interaction between AIFM1 and TIM23 (*Fig 6D*), as did the IP results.

iii). Size-exclusion chromatography was also performed to determine whether AIFM1 could migrate in the TIM23 complex. Monitoring the elution profiles of the TIM23 translocase in mitochondrial extracts from WT or *AIFM1* knockout (KO) cells revealed a significant shift in the TIM23 translocase to a lower molecular weight in the *AIFM1* KO cells. As AIFM1 migration in WT cells is consistent with that of the TIM23 translocase, these results imply that AIFM1 is part of the TIM23 complex. The results are described in the revised manuscript (*Fig EV5C*).

iv). Finally, we performed a native antibody-based mobility shift assay (NAMOS: Swamy *et al*, *J Immunology Methods* 2007) to evaluate the presence of AIFM1 in the TIM23 complex. Mitochondrial extracts from *AIFM1* KO cells or cells expressing Myc-tagged AIFM1 were incubated with or without a Myc antibody and subjected to CN-PAGE, followed by detection using a TIM23 antibody. Adding the Myc antibody resulted in a mobility shift in the TIM23 complex (= AIFM1/Myc), which also indicates that AIFM1 is part of the TIM23 complex (see *Fig C1*, red asterisk).

Together, these results affirm that AIFM1 endogenously associates with TIM23 in the MIM.

Fig C1. NAMOS assay to detect the AIFM1-TIM23 interaction.

“The above mentioned papers should be cited”

We appreciate this comment and have cited the suggested papers (Salscheider *et al*, *EMBO J* 2022; Brosey *et al*, *EMBO J* 2025; Mussulini *et al*, *EMBO Rep* 2025) in the revised manuscript, which has significantly improved the quality of our discussion.

Minor concerns that should be addressed:

“The authors focus the introduction on the mechanisms involved in morphology of mitochondria. However the paper is about OMA and AIFM1 and the introduction should fine-tuned in one of the directions, or both. The intro seems to deal with another topic that the experiments.”

We appreciate this comment and have revised the text as suggested. In the revised manuscript, we provided more detailed notes on AIFM1 in the *Introduction* section as follows:

Page 5, lines 7-18: “AIFM1 is a mitochondrial flavoprotein initially identified as... altering the topology of one of its substrates.”

“There is a typo in the abstract: AIFM1 is...the membrane-anchored inter membrane space (IMS) protein, not “inner membrane space”

Thank you for this comment. We corrected the typo as follows:

Page 2, lines 6-7: “the membrane-anchored intermembrane space protein AIFM1”

【Referee #3 acknowledged the significance of the study and provided us with 10 specific points】

“Nishigori et al. used co-IP with myc-tagged Oma1 to identify binding partners. Interestingly, they pulled down AIFM1, along with Opa1, prohibitins and a series of MICOS proteins. A MassSpec assay for Oma1-dependent changes in the N-termini of peptides further suggested that AIFM1 is a substrate for Oma1. Cleavage was induced by FCCP as well as COX10 deletion. The authors also confirmed that other mitochondrial proteases were not involved, and they found that the cleavage of AIFM1 is much slower than that of Opa1. The authors identified the Oma1 cleavage site in AIFM1, mutated it, and replaced it with a TEV cleavage site to study the

consequences of cleavage events without stress-inducing conditions. The authors then demonstrate that uncleaved AIFM1 is part of a larger protein complex. Other proteins in this complex include ETC complex I proteins and inner membrane carrier proteins. They also found that the uncleaved form is needed for respiration, consistent with a role in promoting complex I function. The authors then demonstrated that the levels of a large number of other proteins imported by TIM23 are also influenced by the cleavage state of AIF. In addition, AIFM1 co-IPs components of the TIM23 complex, consistent with earlier studies showing that it is an integral part of the import machinery.

The identification of AIF as a substrate for Oma1 and the observation of slow cleavage kinetics are potentially significant findings. However, the discussion of their functional consequences is unclear. For instance, the heading "AIFM1 ensures cell proliferation by controlling OXPHOS activity" suggests a direct control of OXPHOS, but the data are more consistent with a general role as an oxidoreductase that aids protein import during mitochondrial biogenesis, as also noted toward the end of the discussion. Stress-induced cleavage by Oma1 would, therefore, hinder import in a response that is considerably slower than Opa1 cleavage, but aligns with the kinetics of Dele1 cleavage by Oma1."

We appreciate these constructive comments and respond to each point below.

Some comments:

1. *"FCCP is probably not the best inducer for studying the physiological relevance of AIFM1 cleavage, because it will also block protein import during biogenesis, which seems to be one of the main functions of AIF"*

Thank you for this comment. We agree that it is difficult to evaluate the general relevance of AIFM1 cleavage based solely on results obtained using FCCP. To address this issue, we used several inducers to activate OMA1, including valinomycin (a potassium-selective ionophore), H₂O₂ (an oxidizer), and oligomycin A and antimycin A (respiratory inhibitors). These same treatments also induced AIFM1 cleavage (*Fig EV1B* and *EV2D* in the revised manuscript).

However, these chemicals are not the most effective means for studying the physiologic relevance of AIFM1 cleavage. Therefore, we explored the impact of AIFM1 cleavage *in vivo*. Notably, we found that influenza A viral infection stress activates OMA1 and increases AIFM1 processing in mouse lung (*Fig 7G* in the revised manuscript), demonstrating its physiologic impact *in vivo*. Given the significance of these findings, we added a new section in the *Results* section (pages 17-18) titled "*Physiologic impact of AIFM1 cleavage in vivo*" to the revised manuscript.

2. “It is unclear why AIF, which is reported to assist the disulfide relay, including MIA40 and Erv1, would mainly affect the import of complex I proteins that pass through Tim23. Are the authors suggesting a general failure of protein import in AIF mutants?”

We sincerely appreciate this comment. To address this comment, we first examined the mitochondrial proteome, focusing on MIA40 substrates (Reinhardt *et al*, *Biochim Biophys Acta* 2020) using non-cleavable and soluble AIFM1-expressing cell lines. The results are included in *Fig EV4I* of the revised manuscript. We confirmed that MIA40 substrate protein levels were significantly lower in cells expressing the soluble AIFM1 variant (AIFM1^{TCS/TEV}) than in those expressing the non-cleavable version (AIFM1^{TCS}). These results suggest that the spatial organization of AIFM1 in mitochondria influences not only the import of complex I subunits via the TIM23 pathway, but also the MIA40 pathway. These data further support the idea that the MIA40 pathway is indispensable for complex I biogenesis.

Therefore, we again explored the physiologic role of AIFM1 topology in MIA40-related biogenesis in tissue using the proteome approach. We found that the lungs of virus-infected mice had significantly higher levels of immune response proteins (*Fig EV6C* in the revised manuscript), whereas the levels of MIA40 pathway substrates (*Fig. EV6D*) and OXPHOS complex subunits (*Fig. 7H*) were much lower than in uninfected mice. Given the significance of these *in vitro* and *in vivo* findings, we added the new data to the revised manuscript (*Dataset EV6*).

3. “The differences in growth under respiring conditions seem relatively minor for a protein that would halt import under stress conditions. Perhaps AIF cleavage is more important for apoptosis than it is for stopping protein import.”

Thank you for this comment. In response to the comment, we tested whether the spatial organization of cleaved or non-cleaved AIFM1 in mitochondria affects the staurosporine (STS)-induced apoptotic process. Through monitoring the release of lactate dehydrogenase (LDH), we confirmed that AIFM1 certainly regulates cell death resistance, as previously reported (*Appendix S4A* in the revised manuscript; WT vs AIFM1^{KO}). However, we observed no difference in cell death between AIFM1 variants (AIFM1^{TCS} vs AIFM1^{TCS/TEV}). Consistent with the results of LDH assay, PARP cleavage levels in cells expressing the AIFM1 variant that were treated with STS or that overexpressed the proapoptotic protein Bax did not change significantly (*Appendix S4B*). Based on these results, we hypothesize that AIFM1 cleavage by OMA1 is not correlated with apoptotic insult.

However, it is conceivable that the apoptosis-related attenuation of $\Delta\Psi_m$ activates OMA1, which would subsequently process AIFM1 under certain circumstances. As the reviewer suggested, we discuss this possibility in the revised *Discussion* section (*p.18, line 4 from bottom* to *p.19, line 5*; “Therefore, the previously reported...required for AK2’s role in apoptosis.”).

Small points:

4. “Fig. 1D. suggests that AK2 and Tom40 are also OMA1 substrates. Considering that Tom40 is in the outer membrane, is the cleavage site at least exposed to the IMS?”

Yes, it is. Our neo-amino-terminal MS approach identified two nearly identical peptides from TOM40 (*Appendix Table S1* in the revised manuscript). A recent structural study of the TOM complex revealed that the N-terminal region of TOM40 exposes the peptide sequence recognized by OMA1 in the IMS (Araiso *et al*, *Nature* 2019; Araiso *et al*, *FEBS J* 2021). See *Fig C3*, indicated by the red dashed line.

Fig C3. Topologic model of the putative cleavage site of TOM40 targeted by OMA1.

5. “Fig. 1E. AIF cleavage is much stronger here than on other blots. Is this because of a longer incubation time than in other experiments?”

Fig C4. The Tryp-N digested AIFM1 peptide from the WT MEFs ± FCCP treatment was quantified by targeted MS using the PRM method.

Thank you for this comment. We believe this is likely due to the fact that the OMA1-rescued cells used in the study exhibit slightly different protein expression levels compared with endogenous protein levels (i.e., *Fig EVIA & EVID* in the revised manuscript). However, the observed OMA1-dependent processing of AIFM1 was not merely a cellular artifact. We confirmed that WT MEFs exhibited the same peptide enrichment at the endogenous level in response to FCCP (see *Fig C4*).

6. “Page 8. The 2nd paragraph does not show AIF cleavage is dispensable for fission and fusion. It is the other way around.”

We sincerely appreciate this comment. We revised the text as follows: "These results suggest that mitochondrial fusion and fission processes are likely dispensable for OMA1-mediated AIFM1 cleavage" (page 9, lines 2–3).

7. “Fig. 3D suggests that AIF cleavage is accelerated by loss of Opa1, which could be an indirect effect of mitochondrial dysfunction. The authors should test causality with an uncleavable Opa1 mutant.”

We appreciate this comment and performed the suggested experiment. Using a cell line that expresses a non-cleavable OPA1 mutant (denoted OPA1V1^{Δ4}), we confirmed that loss of OPA1 accelerates AIFM1 processing, an effect that is canceled by the presence of either OPA1V1^{Δ4} or cleavable L-OPA1V1. These results are described in the revised manuscript (*Fig EV3A*).

Next, we investigated whether the loss of mitofusins (Mfn1 and Mfn2) would also accelerate AIFM1 processing by OMA1 because cells with double knockout of mitofusins are also affected by mitochondrial dysfunction resulting from respiratory defects (Chen *et al*, *J Biol Chem* 2005). We found that the AIFM1 cleavage kinetics were not altered relative to WT cells (*Fig EV3D* in the revised manuscript), suggesting that the observation in *OPA1*^{-/-} cells was not simply due to indirect effects of mitochondrial dysfunction.

8. “Fig. 5H. Rescue of a respiratory defect by uncleaved AIFM1 is convincing, but how does it compare with wild type cells? The authors should include that.”

Thank you for this comment. We included the WT OCR dataset in the revised manuscript (*Fig 5H* and *Fig EV4D*).

9. “Fig. 5. If all these proteins coIP, why is the complex only about 300kDa? Or is there a core complex with other proteins such as the import machinery and these are only present in small amounts? Does cycloheximide

change the size of the BNGE complexes?"

We sincerely appreciate this comment. As the reviewer noted, it is possible that the AIFM1 interactome has a much larger size than we observed in Native-PAGE or size-exclusion chromatography if all interacting proteins stably associate with AIFM1. The conditions used in the IP-MS (Fig 5A and 5E), however, were performed on formalin-fixed samples. Therefore, we assume that the interactions between AIFM1 and OXPHOS subunit components detected in IP-MS are relatively weaker than those of the TIM23 complex. This may explain why some of the weaker OXPHOS subunits were undetectable in Native-PAGE or size-exclusion chromatography. Instead, we observed a substantial interaction between AIFM1 and the TIM23 complex in Native-PAGE (see **Fig C1**) and size-exclusion chromatography (**Fig EV5C** in the revised manuscript), as well as in IPs performed under non-fixed conditions (**Fig 6C** and **EV5A**), related to *point #10* described below.

To investigate whether cycloheximide (CHX) alters the size of AIFM1 interactomes, we performed size-exclusion chromatography on endogenous AIFM1 extracted from mitochondrial fractions of untreated or CHX-treated cells. We confirmed that inhibiting protein synthesis did not affect the size of the AIFM1 complex (see **Fig C5**). These results are consistent with data showing that inhibiting protein synthesis with CHX does not diminish the AIFM1–TIM23 interaction, as revealed by an IP and *in situ* proximity ligation assay (PLA) (related to *point #10*, described below).

10. “Also, it seems that CI and a number of carrier proteins stand out, but these are imported by different Tims (Tim22 vs Tim23). Does that mean AIF affects both processes?”

Thank you for this comment. First, due to data limitations, we toned down the finding of interactions between AIFM1 and mitochondrial carrier proteins, such as the SLC25A family. In addition, we excluded related data from the revised manuscript (Fig 5A).

We hypothesize that AIFM1 specifically binds to the TIM23 complex for the following reasons. First, IP data show that AIFM1 co-precipitates with TIM23 complex subunits (TIM23, TIM50, and TIM17), but not with the TIM22 translocase (Fig EV5A in the revised manuscript). Second, PLA also supports the specificity of the AIFM1–TIM23 interaction (Fig. 6D). In both experiments, we believe that this interaction is not part of transient associations that occur during protein import because inhibiting protein synthesis with CHX did not diminish the observed AIFM1–TIM23 interaction (Fig 6C-6D, + CHX). Finally, size-exclusion chromatography shows that AIFM1 and TIM23 have a similar elution profile when migrating from mitochondrial extracts of WT cells (also see Fig C5). The elution profile of the TIM23 complex, however, shifts to a lower molecular weight when AIFM1 is omitted from the cells (Fig EV5C, see *AIFM1* KO), suggesting that AIFM1 is part of the TIM23 complex. In the revised manuscript, we also discuss the role of AIFM1 in the MIA40 pathway (Fig EV4I and EV6D).

Taken together, we hypothesize that AIFM1 anchoring to the MIM results in a functional interaction with the TIM23 translocase that coordinately regulates OXPHOS biogenesis.

Dear Dr Koshiba,

Thank you for submitting your revised manuscript (EMBOJ-2025-121411R) to The EMBO Journal, as well for your patience with our feedback. Your amended study was sent back to the referees for their scientific reassessment, and we have received re-reports from all of them, which I enclose below. As you will see, the reviewers state that the work has been substantially enhanced by the revisions and they are now broadly in favour of publication.

Thus, we are pleased to inform you that your manuscript has been accepted in principle for publication in The EMBO Journal.

We now need you to take care of a number of issues related to formatting and data presentation as detailed below, which should be addressed at re-submission.

Please contact me at any time if you have additional questions related to below points.

Thank you for giving us the chance to consider your manuscript for The EMBO Journal. I look forward to your final revision.

Again, please contact me at any time if you need any help or have further questions.

Best regards,

Daniel Klimmeck

>> Author Contributions: Remove the author contributions information from the manuscript text. Note that CRediT has replaced the traditional author contributions section as of now because it offers a systematic machine-readable author contributions format that allows for more effective research assessment. and use the free text boxes beneath each contributing author's name to add specific details on the author's contribution.

More information is available in our guide to authors.
<https://www.embopress.org/page/journal/14602075/authorguide>

>> Correct the order of the manuscript sections as follows: Abstract / Keywords / Introduction / Results / Discussion / Methods / Data Availability / Acknowledgements / Disclosure and Competing Interests Statement / References / Main Figure Legends / Tables / Expanded View Figure Legends.

>> Please remove the "Expanded View" section before the Acknowledgements.

>> Funding: please add the following information to the list of funders in our online system: 'KAKENHI Grant 24K220, the Medical Research Center Initiative for High Depth Omics, the Joint Usage and Joint Research Programs, the Institute of Advanced Medical Sciences, Tokushima University, and the Grant from International Joint Usage/Research Center, the Institute of Medical Science, the University of Tokyo'.

>>Appendix file with ToC: please add page numbers to table of contents.

>> Provide comprehensive source data for the study as to the separate request e-mail by our office team.

>> Reagents and Tools table: Please upload as a separate file using the existing template in the Guide For Authors, listing key reagents, experimental models, software and relevant equipment.

>> Data availability section: make sure that all global datasets at GEO, PRIDE and proteomeXchange are publicly accessible.

>> Dataset EV legends: Please add a legend with the title and short description to each file in a separate tab or worksheet.

>> Consider additional changes and comments from our production team as indicated below:

- DAS:

1. Please note that the specific URLs for ProteomeXchange Consortium (PXD063014, PXD063017, PXD063094, PXD069450, PXD063602) and Gene Expression Omnibus datasets (GSE296062) are not provided in the data availability statement.

2. Please note that reviewer access codes for "ProteomeXchange Consortium (PXD063014, PXD063017, PXD063094, PXD069450, PXD063602) and Gene Expression Omnibus datasets (GSE296062)" datasets are not provided in the data availability statement.

- Figure legends

1. Please define the annotated p values ****/****/**/* as well as provide the exact p-values for the same in the legend of figure EV 5d as appropriate.

2 Please note that the exact p values are not provided in the legends of figures 1b; 5f, g, i; 6g; 7i; EV 4d, g; EV 6e

3 lease indicate the statistical test used for data analysis in the legends of figures 1a, b; 5a, b, g, i; 6a, b, e, g, 7i; EV 4d, e, g, h; EV 5d; EV 6c, e

4 Please note that information related to n is missing in the legends of figures EV 5d

5. Please note that the error bars are not defined in the legends of figures 3b, e; EV 3c, d; EV 5d

6. Please indicate the statistical test used for data analysis in the legends of figures 1a, b; 5a, b, g, i; 6a, b, e, g, 7i; EV 4d, e, g, h; EV 5d; EV 6c, ePlease indicate the statistical test used for data analysis in the legends of figures 1a, b; 5a, b, g, i; 6a, b, e, g, 7i; EV 4d, e, g, h; EV 5d; EV 6c, e

Referee #1:

The authors have addressed all my comments and I highly recommend acceptance of this exciting work.

Referee #2:

The authors have adequately responded to suggestions and recommendations for improvement.

This is a highly interesting story that identifies a new stress sensing mechanism mediated by AIFM1 and OMA1 impacting protein import and shaping the proteome of mitochondria.

Referee #3:

In this newly revised version of the manuscript, the authors have gone out of their way to address my comments, as well

as

those of the other reviewers. The new addition describing the effects of viral infections is quite interesting. (super small comment: next time, please put figure numbers under the figures to make it easier for reviewers).

The authors addressed the remaining editorial issues.

Dear Dr Koshiba,

Thank you for submitting the revised version of your manuscript. I have now evaluated your amended manuscript and concluded that the remaining minor concerns have been sufficiently addressed.

I am thus pleased to inform you that your manuscript has been accepted for publication in the EMBO Journal.

Best regards,

Daniel Klimmeck

Daniel Klimmeck, PhD
Senior Editor
The EMBO Journal
EMBO
Postfach 1022-40
Meyerhofstrasse 1
D-69117 Heidelberg
contact@embojournal.org

Please note that it is The EMBO Journal policy for the transcript of the editorial process (containing referee reports and your response letters) to be published as an online supplement to each paper. If you should prefer removal of any referee-only figures included in the point-by-point response(s), e.g. because they may still be used for future publication or because they have been reproduced from published work by others, please do let us know immediately via response email. More information is available here: <https://link.springer.com/partners/embo-press/editorial-policies#Peer%20review>